# Rift-induced disruption of cratonic keels drives kimberlite volcanism

Thomas M. Gernon[1 ✉], Stephen M. Jones[2], Sascha Brune[3,4], Thea K. Hincks[1], Martin R. Palmer[1], John C. Schumacher[5], Rebecca M. Primiceri[1], Matthew Field[6], William L. Griffin[7], Suzanne Y. O'Reilly[7], Derek Keir[1,8], Christopher J. Spencer[9], Andrew S. Merdith[10] & Anne Glerum[3]

Kimberlites are volatile-rich, occasionally diamond-bearing magmas that have erupted explosively at Earth's surface in the geologic past[1–3]. These enigmatic magmas, originating from depths exceeding 150 km in Earth's mantle[1], occur in stable cratons and in pulses broadly synchronous with supercontinent cyclicity[4]. Whether their mobilization is driven by mantle plumes[5] or by mechanical weakening of cratonic lithosphere[4,6] remains unclear. Here we show that most kimberlites spanning the past billion years erupted about 30 million years (Myr) after continental breakup, suggesting an association with rifting processes. Our dynamical and analytical models show that physically steep lithosphere–asthenosphere boundaries (LABs) formed during rifting generate convective instabilities in the asthenosphere that slowly migrate many hundreds to thousands of kilometres inboard of rift zones. These instabilities endure many tens of millions of years after continental breakup and destabilize the basal tens of kilometres of the cratonic lithosphere, or keel. Displaced keel is replaced by a hot, upwelling mixture of asthenosphere and recycled volatile-rich keel in the return flow, causing decompressional partial melting. Our calculations show that this process can generate small-volume, low-degree, volatile-rich melts, closely matching the characteristics expected of kimberlites[1–3]. Together, these results provide a quantitative and mechanistic link between kimberlite episodicity and supercontinent cycles through progressive disruption of cratonic keels.

Over geologic time, pulses of kimberlite magmatism correspond to episodes of global plate reorganization, with comparatively few kimberlite eruptions occurring during periods of supercontinent stability[4] (Fig. 1a). This synchronicity may suggest that kimberlites are triggered by tectonic disturbances in cratonic lithosphere[4] or by abrupt changes in plate movement[6]. However, such hypotheses do not adequately explain what process stimulates melt generation and raises a paradox: cratons are defined by their mechanical strength and long-term stability[7–10], so they should resist tectonic deformation[11]. An alternative model is that kimberlite distributions are controlled by mantle (super)plumes, possibly linked to large low-shear-velocity provinces[5], which might fertilize, hybridize and even destabilize the cratonic root[9,12]. This model is, however, hard to reconcile with geochemical characteristics of kimberlites, many of which are inconsistent with plume sources[13–15] and instead necessitate partial melting of convecting mantle coupled with assimilation of cratonic lithosphere[13,15,16]. Testing these and other models (see Methods) requires both a fully integrated consideration of geodynamics and geochemistry as well as an assessment of spatiotemporal dependencies in the global tectonic cycle.

## Kimberlites and global tectonics

We begin by assessing the global link between kimberlites and tectonics through time and across all continents using a database of radiometrically dated kimberlites[6] and a measure of the degree of fragmentation of continental plates from tectonic reconstructions[17]. We calculate the rate of change of fragmentation ($\Delta F$) as a proxy for dynamic plate reorganization (Extended Data Fig. 1) and then calculate multi-million-year time lags[18] between $\Delta F$ and the kimberlite count, $K$, at 1-Myr intervals (Methods; Fig. 1b and Supplementary Dataset 1). Our cross-correlation analysis reveals a statistically significant association between fragmentation and kimberlites over the past 500 Myr (Fig. 1b), which persists when we account for autocorrelation in the time series (Extended Data Fig. 2a). The strongest correlation coefficient ($\rho = 0.41$) prevails at lags of $-26 \pm 4$ Myr, indicating that continental fragmentation typically leads kimberlite magmatism by 22–30 Myr, whereas a proportion of kimberlites erupt during the interval between rift onset and breakup (Fig. 1b). When we extend our data compilation to 1 billion years ago (Ga), using more uncertain data but capturing two supercontinent cycles (Fig. 1a and Extended Data Fig. 1), the strongest correlation remains at the same

[1]School of Ocean and Earth Science, University of Southampton, Southampton, UK. [2]School of Geography, Earth and Environmental Sciences, University of Birmingham, Birmingham, UK. [3]Helmholtz Centre Potsdam – GFZ German Research Centre for Geosciences, Potsdam, Germany. [4]University of Potsdam, Potsdam-Golm, Germany. [5]Department of Geology, Portland State University, Portland, OR, USA. [6]Mayfield, Wookey Hole, Wells, UK. [7]GEMOC ARC National Key Centre, Department of Earth and Environmental Sciences, Macquarie University, Sydney, New South Wales, Australia. [8]Dipartimento di Scienze della Terra, Universita degli Studi di Firenze, Florence, Italy. [9]Department of Geological Sciences and Geological Engineering, Queen's University, Kingston, Ontario, Canada. [10]School of Earth and Environment, University of Leeds, Leeds, UK. ✉e-mail: T.M.Gernon@soton.ac.uk

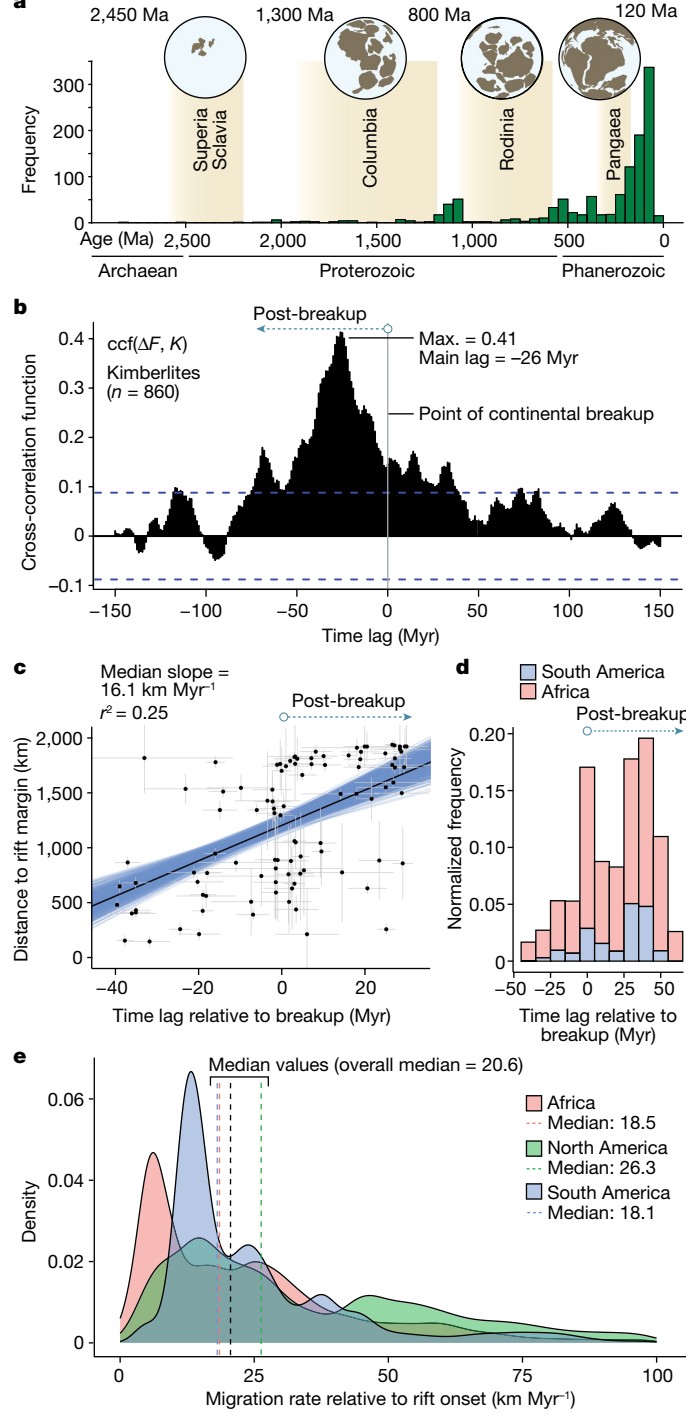

**a**

2,450 Ma  1,300 Ma  800 Ma  120 Ma

Superia Sclavia  Columbia  Rodinia  Pangaea

**b**

$ccf(\Delta F, K)$
Kimberlites
($n = 860$)

Post-breakup

Max. = 0.41
Main lag = −26 Myr

Point of continental breakup

**c**

Median slope =
16.1 km Myr⁻¹
$r^2 = 0.25$

Post-breakup

**d**

South America
Africa

Post-breakup

**e**

Median values (overall median = 20.6)

Africa
Median: 18.5
North America
Median: 26.3
South America
Median: 18.1

**Fig. 1 | Temporal relationships between tectonics and kimberlites.**
**a**, Frequency distribution of kimberlites through geologic time ($n = 1,133$; data from ref. 6), showing peaks coinciding with the breakup phase of supercontinent cycles (modified from ref. 52). **b**, Cross-correlations between $\Delta F$ and global kimberlites[6] spanning 500–0 Ma ($n = 860$; Methods). Note $ccf(\Delta F, K)$ gives correlations between $\Delta F(t+l)$ and $K(t)$ at lags $l$. Here, positive lags indicate fragmentation lagging eruption and negative lags indicate fragmentation leading eruption. The peak correlation at lag −26 ± 4 Myr shows that fragmentation leads kimberlite eruption. Positive correlations show that kimberlites are linked to continental fragmentation, not assembly; dashed blue lines show 95% confidence intervals. **c**, Scatter plots for kimberlite clusters in Africa and South America showing time lags for kimberlite eruption ($n = 388$) relative to continental breakup (at $t = 0$), versus distance from the rifted margins of Gondwana. The individual points are median values for clusters ($n = 175$) accounting for age and distance uncertainties, and error bars show the standard deviation. The best-fit regression line (black) indicates a migration rate of approximately 16.1 km Myr⁻¹; blue lines show individual regressions for 5,000 simulations incorporating uncertainties (Methods), yielding rate estimates from 9.3 to 22.3 km Myr⁻¹. **d**, Time lags for eruption of kimberlite clusters (shown in **c**) relative to continental breakup (Methods; Extended Data Fig. 3d). **e**, Probability density plots showing migration rates of kimberlites ($n = 623$; clusters $n = 253$), here relative to rift initiation, for continental margins of Africa, South America and North America; the global median migration rate is 20.6 km Myr⁻¹ for clustered data (dashed black line), accounting for age, distance and model uncertainties (Methods; Supplementary Dataset 2).

perfectly suited for our purposes because the rift kinematics along adjacent plate boundaries are well constrained at this time[19] and the abundant kimberlites are well understood in terms of their age distributions[6] and the structure of lithosphere they sample[20–22]. We perform spatiotemporal cluster analysis (Methods) to avoid spatial biases that may result from oversampling within kimberlite fields. We account for uncertainty in kimberlite age, breakup age and rift distance by Monte Carlo simulation (Methods; Extended Data Figs. 3–5). Using palaeogeographic reconstructions from the plate tectonics software GPlates[23] (https://www.gplates.org/), we measure distances between kimberlite clusters and adjacent rift boundaries and calculate time lags between continental breakup and eruption (Extended Data Figs. 3 and 4), allowing for cases in which eruption occurred during the interval between rift onset and breakup (Methods). This analysis confirms that the main peak in kimberlite emplacement occurs 25–45 Myr after the onset of regional breakup (Fig. 1d), consistent with our global analysis (Fig. 1b). In particular, we find that kimberlites tend to erupt closer to rifted margins earlier in the rifting cycle—as qualitatively noted previously in the São Francisco and Kaapvaal cratons[9]—and migrate towards the cratonic interior as fragmentation progresses, initially at a rate of about 10–25 km Myr⁻¹, but most likely on the order of 16–20 km Myr⁻¹ (Fig. 1c,e and Extended Data Fig. 3c; Methods).

This finding prompts us to ask whether similar migration patterns can be identified in the many kimberlite fields that erupted across North America during the Phanerozoic, with this activity escalating in the Jurassic (Fig. 1a and Extended Data Fig. 3a). The Cretaceous kimberlite 'corridor' (encompassing Somerset Island, Kansas and Saskatchewan) tracks the edge of the North American Craton, far from any known mantle-plume influence[15]. We perform a similar distance–lag analysis for these kimberlites relative to the major Pangaea rift system that initiated along the Atlantic continental margin of the United States in the Middle Triassic, at around 240 million years ago (Ma) (ref. 24). In addition to the Central Atlantic, this also encompasses the later rifts of Greenland–North America and the Arctic/Amerasian Basin (Methods). As with the African and South American fields (Fig. 1d), we can identify two peaks in volcanism—one occurring between rifting and breakup and the other lagging breakup by 30–60 Myr (Extended Data Fig. 5). Collectively, these data reveal median migration rates of 26.3 km Myr⁻¹ relative to rifting onset (Fig. 1e). The variability in

lag (Extended Data Fig. 2b). Further, when we account for preservation bias by weighting the number of kimberlites inversely according to surface preservation, the same lag persists (Extended Data Fig. 2c). The signal strengthens ($\rho = 0.52$) when we isolate kimberlites <200 Myr old, a period when the level of certainty in global tectonic models is highest (Extended Data Fig. 2d; Methods). The consistent delay between rifting onset and kimberlite volcanism strongly seems to be a genuine feature of the global geodynamic cycle.

We scrutinize this linkage further by analysing the temporal and spatial relationships between kimberlites and continental plate margins, first targeting the Mesozoic kimberlite fields of Africa and South America that formed during breakup of the southern part of the Pangaea supercontinent, Gondwana (Fig. 1a). These regions are

migration-rate estimates (Fig. 1e) can be explained by the complex spatial and temporal evolution of rifting (and subsequent breakup) in relation to kimberlite clusters. To address this, we quantify the uncertainty associated with attributing a kimberlite eruption to a specific initiating rifting event (Fig. 1e; Methods). Median migration-rate estimates for the endmember cases are 11 and 30 km Myr$^{-1}$ for kimberlite clusters globally (Extended Data Fig. 4), with an overall median of 20.6 km Myr$^{-1}$ (Fig. 1e; including $n = 623$ kimberlites, or 87.5% of the catalogue from 240 Ma). Notably, these rates are consistent across all three continents, irrespective of continental length scale (Fig. 1e), suggesting that some fundamental mechanism gives rise to migration of kimberlite volcanism far inboard of rift zones in the tens of millions of years after rifting begins.

## Kimberlites, rifting and mantle plumes

Both our global (Fig. 1b) and regional (Fig. 1c–e) analyses indicate that kimberlite magmatism is strongly related to continental breakup. However, both breakup[25] and kimberlite magmatism[5] are commonly attributed to mantle plumes. A crucial question, therefore, is whether rifting itself is the primary driver of kimberlites or whether mantle plumes drive both rifting and kimberlite magmatism. To address this question, we quantify the relationship between rifting, plumes and kimberlites globally (Methods). We find that the strongest statistically significant correlation between large igneous provinces (LIPs, the accepted main surface expression of mantle plumes[25]) and continental fragmentation occurs at $+7 \pm 4$ Myr, that is, plumes/LIPs lead breakup (Extended Data Fig. 6). Recalling our observation that most kimberlites lag breakup by around 30 Myr, it at first seems possible that a plume could trigger breakup roughly 7 Myr after impingement and then the main peak in kimberlite generation approximately 30 Myr later. However, the lagged correlation between breakup and kimberlites is considerably stronger than that with LIPs, suggesting that rifting is the first-order control. Further, the plume model cannot explain our observation that kimberlites tend to erupt closer to the rift boundary earlier in the rifting cycle and migrate inboard of the rift over time. There is no clear mechanism by which plumes could explain this pattern. Thus, we conclude that mantle plumes may (or may not) be a primary driver of continental breakup and may locally warm cratonic keels[12], impregnating them with super-deep diamonds (potentially long before eruption[14]), but it is the rifting process itself that controls most kimberlite magmatism.

We propose that rifting triggers the migrating pattern of kimberlite eruptions hundreds of kilometres inboard of the rift over time (Fig. 1c). We review the proposed mechanisms for tectonic and magmatic rejuvenation of cratons to assess which have the potential to explain both kimberlite melting and migration, as well as an association with rifting (Methods). Cratonic thinning by surface uplift and exhumation can be triggered by rifting[26] but occurs too slowly to generate melting. In cases in which rifting occurs on the edge of the craton, the cratonic lithosphere does not seem to stretch and thin mechanically. This interpretation is based on geologic observations of a lack of horizontal tectonic motions within cratons[7–11] and on dynamical models that show that lithosphere >300 km inboard of rifted margins is not thinned appreciably by extension[27,28]. The remaining potential cratonic rejuvenation mechanisms involve removal of the basal lithosphere[29], or keel, and the subset that can potentially explain both kimberlite melting and an association with rifting all involve mantle convective removal of the keel (Methods). The open questions are: how does rifting trigger convective instability in the adjacent cratonic lithospheric keel (hereafter, keel); and can removal of this keel cause melting of appropriate volume and composition?

Kimberlites are generally found in, or marginal to, thick (150–250 km) cratons[1,3,7,8,11,13,30] (Extended Data Fig. 3a,b). An inevitable consequence of fragmenting cratons is the generation of a physically steep-sided lithosphere–asthenosphere boundary (LAB)[12]. The steep edge of the LAB prompts edge-driven convection[31], in which convective downwelling of keel occurs on the side of the edge further from the rift, as demonstrated by numerical modelling[26–28,32] and seismic tomography beneath modern passive margins[31]. Although edge-driven convection is triggered by rifting and can remove keel near the rift, it is at present unclear whether it can explain convective removal of keel further away (>300 km) from the rift. One possible mechanism is Rayleigh–Taylor instability, a well-described mantle convective process that can potentially occur beneath mature continental or oceanic lithosphere[26,29,32–35]. This instability is driven primarily by the density contrast between colder lithosphere and hotter asthenosphere. In the case of cratons, this negative buoyancy driver can be augmented by metasomatism, or refertilization, of keels[21,29,36,37]. For instance, pre-eruptive melt metasomatism[36] can impart compositional changes (for example, Extended Data Fig. 7) that increase the bulk density by several percent[36,37], destabilizing the lowermost tens of kilometres of keel[21,36–38]. Therefore, Rayleigh–Taylor instability could cause convective removal of keel and, subsequently, lithospheric thinning. Further, convective downwelling is balanced by upwelling of asthenosphere that can feasibly cause melting, particularly if detached keel veined by hydrous and carbonate-rich metasomatic phases[12,20,39] is entrained in the upwelling. Petrological observations[12,21,36,40] and dynamic models[34] indicate that substantial removal of cratonic keel (tens of kilometres thick) can occur abruptly (over several million years)[9,12], suggestive of a convective process. The question is then whether rift-related edge-driven convection can destabilize, and partially melt, adjacent keel hundreds to even thousands of kilometres inboard of rifted margins through Rayleigh–Taylor instabilities.

## Geodynamic and analytical models

To address this question, we quantify the mechanical and thermal influence of breakup on lithospheric stability using two independent and complementary methods (Figs. 2 and 3). To investigate the essential physics of the process, we perform a scaling analysis based on analytical models of viscous instabilities representing the thermal boundary layer between mechanically rigid lithosphere and convecting asthenospheric mantle (Fig. 3a and Extended Data Table 1; Methods). To assess the influence of the more complex natural geometry and rheology of rifting cratonic lithosphere, we also carry out more sophisticated numerical thermomechanical simulations (Fig. 2 and Extended Data Table 2; Methods). Both methods predict that rift onset and subsequent necking should trigger initial instabilities in the basal lithosphere beneath the rift shoulder, which crucially then trigger a chain of further instabilities that propagate away from the rift towards the cratonic interior (Fig. 2). The scaling analysis predicts that the horizontal propagation velocity for the chain of instabilities scales as the ratio of the characteristic horizontal wavelength, $\lambda_d$, and the characteristic e-folding growth time, $\tau_d$, of individual instabilities. Specifically, the propagation velocity ($U$) is given by

$$U \sim \frac{\lambda_d}{\tau_d} = \lambda_d^* q_d^* \frac{g' b^2}{\nu} \qquad (1)$$

in which $\lambda_d^*$ and $q_d^*$ are analytically determined scales for the characteristic horizontal wavelength and growth rate, respectively, $g' = g\Delta\rho/\rho$ is reduced gravity, $\Delta\rho$ is the density difference that drives the instability relative to asthenospheric density $\rho$, $b$ is the mean starting thickness of the unstable lithospheric keel (that is, thermal boundary layer) and $\nu$ is the kinematic viscosity. Using equation (1), and considering thermal-boundary-layer thicknesses from xenolith pressure–temperature ($P$–$T$) estimates (Extended Data Fig. 8), the propagation velocities are on the order 14–26 km Myr$^{-1}$ (Fig. 3a and Extended Data Table 1). The simulations confirm how migrating instability leads to a sequence of progressive convective removal events (Fig. 2 and Supplementary Video 1), which initiate during rifting and migrate

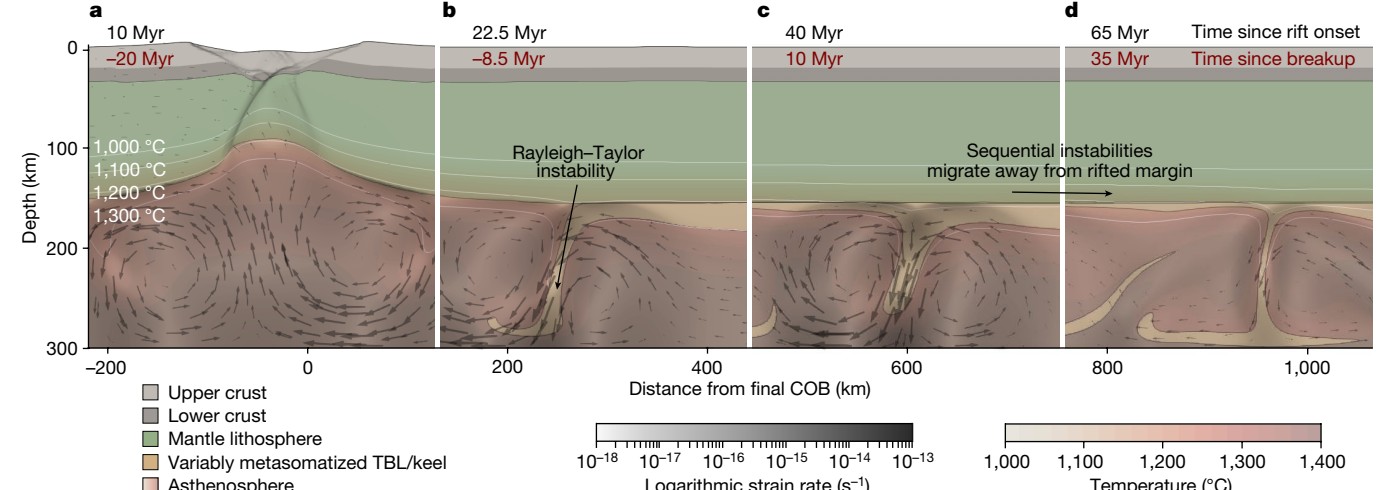

**Fig. 2 | Thermomechanical simulations of continental breakup showing generation and migration of Rayleigh–Taylor instabilities. a**, Initial rifting causes mantle upwelling (black arrows) and development of steep lithospheric gradients; distance from the continent–ocean boundary (COB) is shown along the base and time since rift onset (black) and breakup (red) along the top (TBL, thermal boundary layer). **b**, Rifting and necking leads to Rayleigh–Taylor instabilities in the convective mantle, which—over time (**c**,**d**)—propagate along the base of cratonic keel, progressively removing the TBL. The instability shown migrates at a rate of approximately 15–20 km Myr⁻¹ (Fig. 3a, Extended Data Fig. 9 and Supplementary Video 1). The models use an activation energy of 480 kJ mol⁻¹ that is considered most reasonable for the asthenospheric mantle[53], but the effects of varying these values on the migration of instabilities and convective removal are explored in Methods. Details of the model setup are provided in Methods.

beneath the craton at similar rates of 15–20 km Myr⁻¹ (Extended Data Fig. 9). Both approaches yield propagation (or migration) rates that are very closely consistent with those estimated for kimberlites (Figs. 1e and 3a and Extended Data Fig. 3c). Further, the characteristic scale of instabilities (wavelengths: 40–65 km; Extended Data Table 1) broadly

match those of proposed kimberlite melt sources (10–100 km diameter[41,42]) and kimberlite fields at the surface (30–50 km diameter[42]). The simulations (Fig. 2 and Extended Data Fig. 9) corroborate our observation that kimberlite migration can initiate during rifting, tens of millions of years before breakup (Fig. 1c). This suggests that

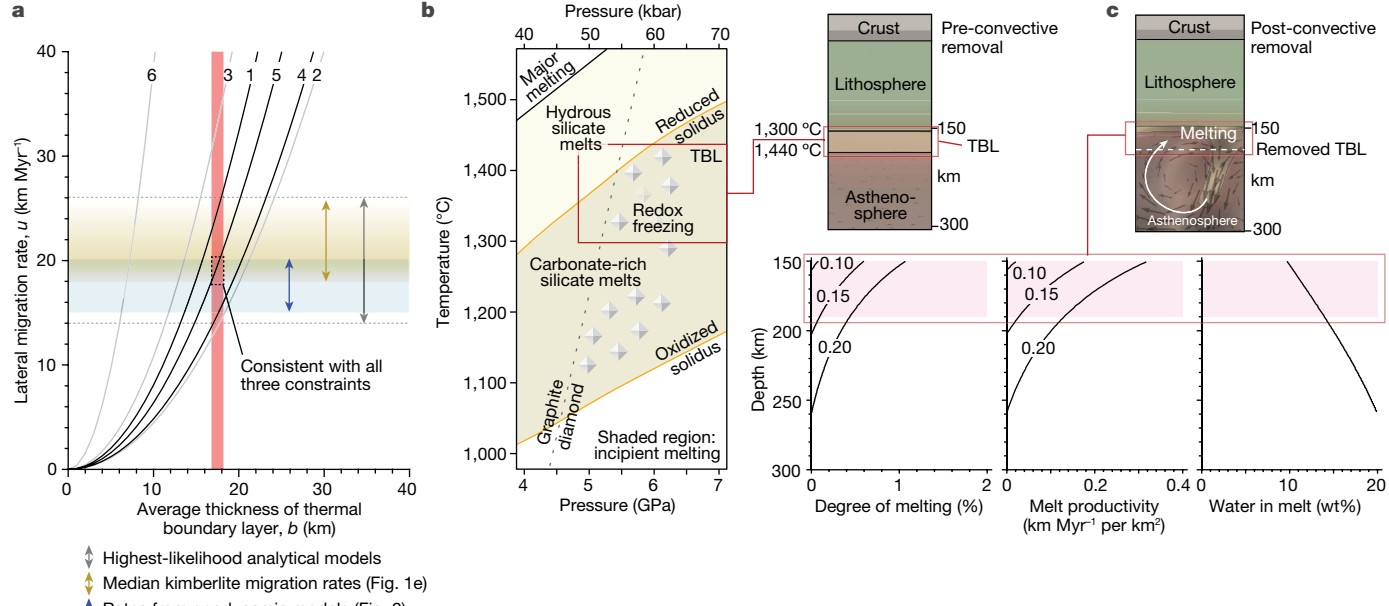

**Fig. 3 | Rates of instability migration and conditions of melt generation. a**, Lateral migration rates for Rayleigh–Taylor instabilities (equation (1)) for six analytical fluid-dynamical models (Extended Data Table 1). Models 1, 4 and 5 (black lines) are most applicable and describe a dense, viscous layer representing the lithospheric keel that overlies a less dense, viscous half-space representing asthenosphere. A kinematic viscosity of $4 \times 10^{15}$ m² s⁻¹ gives good agreement with constraints from kimberlite migration (Fig. 1e), numerical simulations of basal lithospheric instabilities (Fig. 2) and thermal boundary layer (TBL) thicknesses inferred from modelling xenolith P–T data (Methods). **b**, Phase

diagram for peridotite showing the oxidized solidus ($CO_2 + H_2O$) and reduced solidus ($CH_4 + H_2O$) from ref. 30 (see Methods for further details) and graphite–diamond phase boundary from ref. 54; red box shows P–T conditions in the TBL (Extended Data Fig. 8). **c**, Degree of melting as a function of depth; also shown is the associated melt productivity per unit area of convective upwelling, assuming a mean upwelling rate of 30 km Myr⁻¹ determined from numerical models (Fig. 2) and the wt% water in the resulting melts for bulk water contents of 0.1, 0.15 and 0.2 wt%; these calculations use the hydrous decompressional melting parameterization of ref. 47 (Methods).

kimberlites can feasibly be associated with 'failed' rifts as well as those that progress to breakup.

## Hybridization of mantle melts

Hybridization of asthenospheric and lithospheric melts is required to explain kimberlite compositions[16]. At pressures conducive to diamond formation ($\geq$5 GPa), the LAB—a thermal boundary with a temperature range of 1,300–1,400 °C—closely coincides with the solidus of carbonated mantle[2,43]. Hence, although cratonic roots are anomalously thick, cool and stable over billions of years[7,9,30], only small changes in pressure and temperature are required to generate the small-volume, low-degree partial melts thought to be characteristic of kimberlites[2,42–44].

Our models suggest that entrained, variably metasomatized keel (Fig. 2) could be an important contributor of carbonate and hydrous phases to the kimberlite mantle source[2,3,41,45]. The question is whether these phases coexist in the thermal boundary layer of the lithosphere, which we propose is detached by means of convective erosion before rapidly recirculating upward and melting (Fig. 2). To investigate this, we can examine phase equilibria models of peridotite rocks[30] that dominate the cratonic keels[7,10,11]. In our model, migrating convective instabilities detach peridotite from the keel and kimberlite magmas result from partial melting of the convectively circulating peridotite. Considering peridotite melting in the presence of $CO_2 + H_2O$, the thermal boundary layer (as defined by our geotherm analysis; Extended Data Fig. 8) is expected to have an important reduced solidus that juxtaposes carbonate-rich and hydrous incipient silicate melts[6] (Fig. 3b). The shallowest keel is metasomatized, containing diamonds that probably formed through redox freezing[30]. Melting experiments show that hybridization of these reduced, depleted peridotites with oxidized, hydrous $CO_2$-rich melts should drive strong melting reactions[30]. However, until now, a mechanism to decouple and hybridize these oxidized and reduced domains was lacking.

We propose that convective removal of the thermal boundary layer can hybridize such compositionally heterogeneous domains in the asthenosphere, promoting interaction between these reactive, incipient melts (Fig. 2 and Supplementary Video 1). In our analytical models, the thermal Péclet number for return flow confirms that asthenosphere will well up adiabatically (to shallower depths of 150–170 km and lower pressures of 5–5.5 GPa) to replace the removed part of keel (Fig. 2, Fig. 3c and Extended Data Table 1). Our simulations then advance on this footing to show that detached keel is entrained in the upwelling asthenospheric return flow (Fig. 2 and Supplementary Video 1), but can this process reconcile the attributes of kimberlite melting? Assuming that $H_2O$ and $CO_2$ are present in the source—as is generally accepted[2,3,15,41,43–46] (Fig. 3b)—we apply a hydrous decompressional melting parameterization[47] (Methods) to estimate that up to approximately 1% partial decompressional melting can occur within the upwelling limbs of convective instabilities beneath thick lithosphere (Fig. 3c). Our prediction closely matches melt degrees inferred from kimberlite petrology (<1%)[42]. Upwelling within each convective instability can potentially generate tens to hundreds of cubic kilometres of magma over its lifetime (Fig. 3c), which is sufficient to explain estimated volumes of eruptions in kimberlite clusters[41]. According to our calculations, the resulting small-volume melts will have pre-eruptive $H_2O$ contents on the order of 9–14 wt% (Fig. 3c), overlapping with estimated compositions[46]. These melts will ascend rapidly and adiabatically[45], reacting with mantle lithosphere and evolving in composition during ascent[3,46].

## Evolution of kimberlite magmatism

If several tens of kilometres of cratonic keel are removed, this should be detectable using geochemical and geophysical constraints. Taking southern Africa as an example, geochemical studies of peridotitic

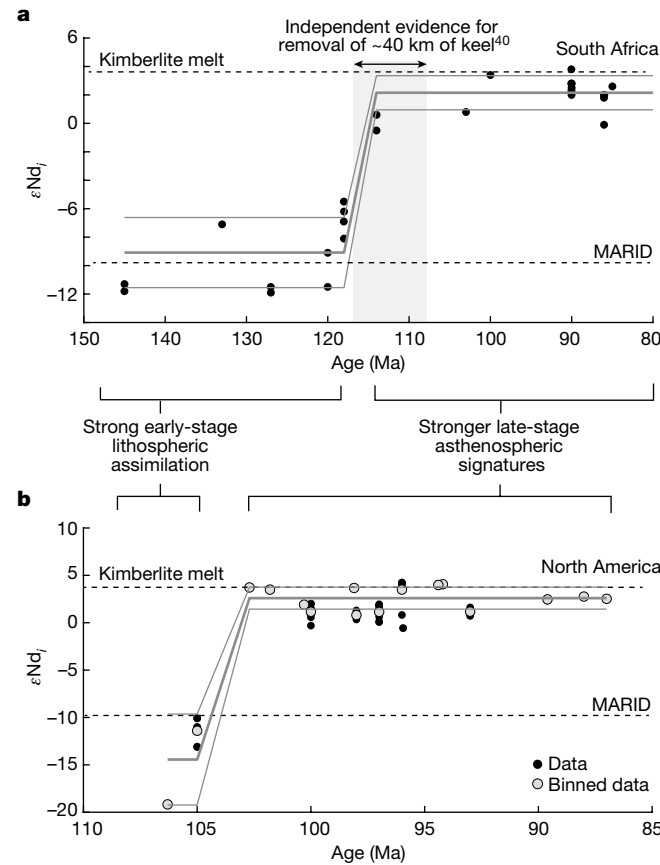

**Fig. 4 | Geochemical evidence for removal of cratonic keel.** Major kimberlite fields in South Africa (**a**) and North America (**b**) show abrupt shifts, identified using conjugate partitioned recursion (grey lines; Methods), from lithospheric ($\epsilon Nd_i$ −6 to −20) to mantle ($\epsilon Nd_i$ 0 to +5) Nd isotope compositions. Note that MARID denotes the mica–amphibole–rutile–ilmenite–clinopyroxene endmember representing metasomatized lithospheric mantle (Methods). Data sources provided in Extended Data Figs. 10 and 11; shaded band in **a** denotes independent evidence for removal of about 40 km of keel[40].

xenoliths and garnet xenocrysts independently invoke loss of 30–40 km of lithosphere, which was coincident with massive kimberlite volcanism during the mid-Cretaceous[40,48] (Figs. 1a and 4). This removal is consistent with evidence for coeval, substantial exhumation (2–4 km) across the cratonic regions of southern Africa and South America[9,49]. The removed thickness closely matches both our empirically derived thickness of the lithospheric thermal boundary layer (about 35 km; Extended Data Fig. 8) and the intensely melt-metasomatized root of the Kaapvaal Craton inferred from the petrology of deep xenoliths brought up by the kimberlites[20–22].

Sublithospheric convective instabilities may also drive melting in the oceanic realm, in which complex (or absent) age–distance patterns of volcanism have been linked to prolonged interaction of asthenospheric mantle with thinned lithosphere[35]. By contrast, a simple, systematic age–distance pattern emerges from our statistical analysis of kimberlites (Fig. 1c,e), although there is clearly variability about this general trend. Our analysis indicates that kimberlite melting beneath cratons occurs through convective removal of keel that has been enriched by volatiles over a prolonged period (in comparison with oceanic lithosphere). This process drives explosive eruptions above the migrating locus of melting (Fig. 5). Indeed, melting may occur in several phases, reflecting the dynamic evolution of rifts. It is tempting to interpret the bimodal peak in kimberlite lag times—evident in Africa, South America and North America (Fig. 1d and Extended Data Figs. 3d and 5)—to relate to two

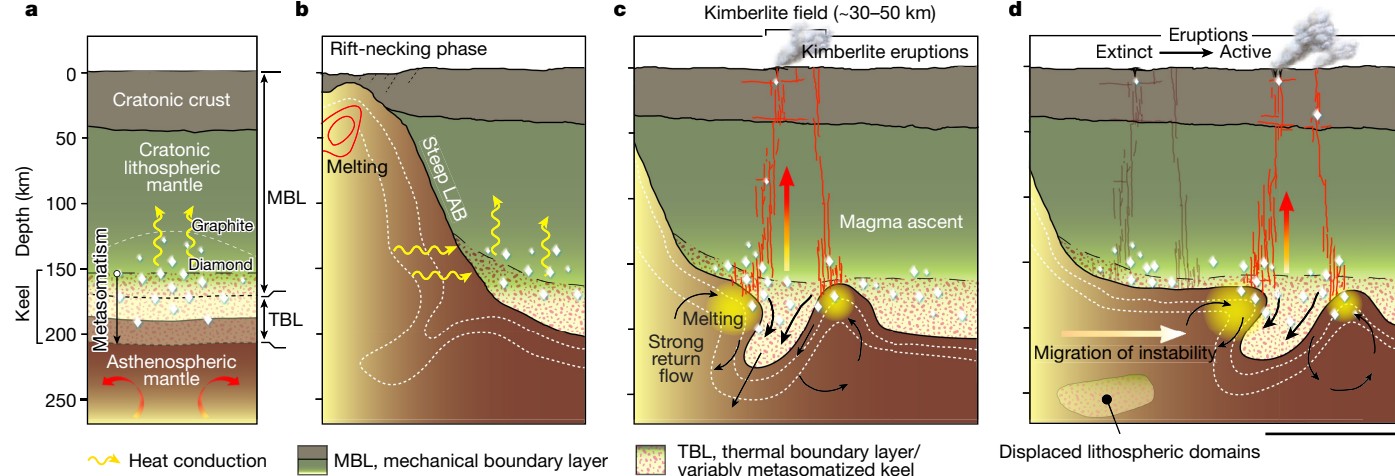

**Fig. 5 | Far-field effects of rifting on cratonic mantle keel stability through time. a**, Simplified craton structure showing mechanical and thermal boundary layers. **b**, Rifting generates a steep lithospheric gradient that gives rise to Rayleigh–Taylor instability (dashed white lines depict schematic isotherms). **c**, Migration, continued growth and detachment of the instability (Fig. 2). Low-density asthenosphere wells up adiabatically (black arrows) to replace removed keel and may entrain fragments of the displaced keel, causing low-degree (<1%) partial melting of a mixture of wet asthenosphere and metasomatized lithosphere. Mixing of the resultant melts produces kimberlitic magmas that ascend rapidly to erupt at Earth's surface. In regions where the return flow

upwelling generates very low-degree (<<1%) melts, these may infiltrate, freeze within and refertilize the keel, thereby promoting further decoupling and magma generation. **d**, The process repeats: destabilization and convective removal of cratonic keel propagates inboard of the rift, leading to migration of kimberlite volcanism towards the cratonic interior (Fig. 1c,e). The simulations predict that the intensity of lithospheric assimilation declines over time: late-stage melting is dominated by asthenospheric mantle once the cratonic keel has foundered and/or been exhausted during early melting (Supplementary Video 1). Scale bar, 100 km.

main phases of instability growth and migration: the first linked to rift onset with kimberlite magmatism peaking 20–40 Myr later (preceding or coinciding with breakup) and the second linked to rift necking with kimberlites peaking 25–50 Myr after breakup (Fig. 1d). Because convective instabilities can persist beneath cratons for many tens of millions of years (Fig. 2 and Supplementary Video 1), the longevity of kimberlite magmatism will be limited by the availability of metasomatized keel. However, this may vary depending on its entrainment rate in convective upwellings and exhaustion during melting, which can result in either prolonged or short-lived magmatic activity, leading to considerable natural variability in the spatiotemporal data, as we observe (Fig. 1c).

Furthermore, kimberlite compositions should evolve from exhibiting mainly lithospheric signatures, which reflect a relatively high proportion of entrained keel, to primarily asthenospheric mantle signatures. We can test this hypothesis by studying time-integrated variations in kimberlite isotope geochemistry at the craton scale. Notably, there is compelling evidence for a pronounced temporal shift in Mesozoic kimberlite compositions in southern Africa that signals abrupt lithospheric disruption: a step change in compositions from those initially exhibiting strong lithospheric and metasomatic enrichment (orangeites, previously termed 'group II' kimberlites)[50] to those exhibiting predominantly asthenospheric mantle signatures (archetypal 'group I' kimberlites)[50] (Fig. 4 and Extended Data Fig. 10). The petrogenesis of later kimberlites implies shallower, hotter melting conditions at 150–160 km depth, in many cases only several million years after melting dominated by lithospheric assimilation at roughly 200 km depth[40]. This compositional change is not peculiar to southern Africa: we identify a remarkably similar shift from lithospheric to mantle kimberlite isotope compositions in North America (Fig. 4 and Extended Data Fig. 11). We attribute these geochemical changes to the progressive convective erosion and eventual removal of the lithospheric keel. Accordingly, the formation of orangeites is linked to the disturbance and upward recirculation of lithospheric material caused by early instabilities. These instabilities are not strong enough to fully remove the keel (Figs. 3c,

4 and 5), resulting in the eruption of small-volume ultrapotassic magmas enriched in metasomatized lithospheric material. In the complete cycle, over tens of millions of years, the lithospheric keel undergoes more vigorous convective erosion, primarily caused by instabilities formed during continental breakup, leading to its progressive removal over distances of $10^2$–$10^3$ km (Supplementary Video 1). This removal is naturally then followed by eruption of kimberlites with progressively stronger asthenospheric signatures (Fig. 5), as evidenced by changes in isotope chemistry[50] and coeval shifts in the Ti contents of garnet xenocrysts[40] (Extended Data Figs. 10 and 11).

In summary, previous models cannot satisfactorily explain how kimberlite melts are generated and mobilized from the mantle source, nor their apparent linkage to the fundamental reorganization of Earth's tectonic plates (Fig. 1). Our analytical and geodynamic models (Fig. 2) demonstrate that rifting and continental breakup can generate a chain of Rayleigh–Taylor instabilities in the asthenosphere that progressively migrate inboard of rift zones, eroding cratonic keel—a process capable of generating kimberlites far from the parent rift (Figs. 3 and 5). Our findings demonstrate that kimberlite volcanism migrates into cratonic interiors at remarkably similar rates to those expected of such instabilities (Figs. 2 and 3a)—an observation spanning several continents (Fig. 1 and Extended Data Fig. 3). Our model reconciles diagnostic kimberlite features, such as association with cratons and geochemical characteristics that implicate a common asthenospheric mantle source contaminated by cratonic lithosphere[13,14,16,51]. Taken together, our results indicate that kimberlite magmas are generated, and their eruptions are triggered, by the far-field effects of rift tectonics during the breakup phase of supercontinent cycles.

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

## Methods

### Relationship between kimberlite formation and craton rejuvenation

The observed relationship between pulses of kimberlite magmatism and episodes of global plate reorganization (Fig. 1) strongly suggests a tectonic mechanism for the generation and mobilization of kimberlite magma. Furthermore, observations that kimberlites are only emplaced on cratons[1–3], and that kimberlite emplacement is commonly associated with uplift and erosion[49,55,56], suggest that kimberlite magma generation may be related to thinning and rejuvenation of cratonic lithosphere. We therefore review mechanisms proposed for thinning the cratonic lithosphere, to assess which mechanisms have the potential to explain both generation of kimberlite magma by mantle melting and an association with global plate reorganization.

Lithospheric thinning can be accomplished by erosion of Earth's surface. Cratonic erosion is often associated with kimberlite emplacement[55,56], but the erosion rates are too slow to drive adiabatic upwelling and consequent melting of the sublithospheric convecting mantle. Lithospheric thinning in response to lithospheric extension is a well-understood mechanism of inducing mantle melting by decompression, but the lack of evidence for tectonic extension of cratons means that this mechanism cannot explain kimberlite melting.

The remaining mechanisms for lithospheric thinning involve removal of the lowermost lithosphere, or keel. The keel is inherently quasi-unstable because it is denser than the underlying asthenosphere. Lee et al.[29] identified five processes that might critically destabilize and remove cratonic keel, which we now review. Cratonic keel is constantly cooled by conduction through the overlying lithosphere, leading to thickening, destabilization and convective removal of the keel[29]. Rayleigh–Taylor instabilities and lithospheric edge-driven convection are special cases of this type of convective removal. The planform and vigour of convective instabilities within the keel are known to be influenced by nearby rifting[26,27,32]. Until now, it was unclear whether rifting can cause convective removal of keel at large horizontal distances (>200 km). Furthermore, although convective instabilities near rifts have been argued to cause melting[26,28], it is unclear whether convective instabilities beneath thick cratonic lithosphere can cause kimberlite melting. Cratonic keel is constantly warmed by the underlying asthenosphere. A mantle plume head and/or smaller-scale asthenospheric convection cells might locally enhance this warming, leading to weakening and convective removal of keel[33]. Thermal weakening might be augmented by advection of heat into the keel if sufficiently large degrees of mantle melting occur. These plume models provide a well-established hypothesis for kimberlite melting, but they do not explain a global association with rifting (Fig. 1). Cratonic keel overlying subduction zones might be destabilized by infiltration of subduction-related fluids and melts, leading to rheological weakening, followed by convective removal[33]. Although this mechanism specifically involves melting, it does not explain the lack of arc geochemical signatures in most kimberlites or the co-location of kimberlites and rifted margins (as opposed to subduction zones). Cratonic keel might be removed by basal traction in response to relative motion between the lithosphere and the asthenosphere[33]. This mechanism does not provide an obvious cause of melting, nor can it explain a global association with rifting; for example, motion of the southern African lithosphere relative to the underlying asthenosphere was low throughout the Mesozoic when many of the southern African kimberlites were emplaced (Fig. 1a and Extended Data Fig. 3a). Cratonic keel might undergo sub-horizontal viscous flow that acts to smooth out basal lithospheric topography[33]. This speculative mechanism has received little attention and it does not provide an obvious cause of melting.

This discussion shows that all the suggested processes that can potentially explain both generation of kimberlite magma by mantle melting and an association with global plate reorganization involving convective removal of keel. There are two main open questions. The first relates to the process that causes destabilization, which might involve rifting and/or mantle plumes. The second relates to the relative role of small-scale convection cells and mantle plumes in kimberlite melting. This paper addresses these questions through a combination of observations and modelling.

### Kimberlite database

We use the kimberlite geochronology database of Tappe et al.[6], which—comprising 1,133 unique occurrences—is one of the most accurate and comprehensive compilations available. Notably, this database shows the same broad temporal distribution (Fig. 1a) as other databases such as that shown in Stern et al.[57], adapted from Faure[58]. It must be noted, however, that the latter compilation[58] contains some highly uncertain ages (that is, those in which kimberlites are assigned only to a geologic period and/or there is only 'loose' stratigraphic age control). We thus prefer to use the database of Tappe et al.[6], which specifies the geochronology method (targeted mineral and isotope system), and for those kimberlites <500 Ma in age, in which radiometric age uncertainties have mean, median and mode values of 6, 4 and 2 Myr, respectively. Further, this compilation represents every important known kimberlite cluster from each continent, avoiding "over-representing economic clusters that host diamond mines and are therefore more intensively studied"[6]. Our regional case studies apply further caution by using a spatiotemporal clustering approach (detailed below) that ensures statistically consistent criteria are applied to kimberlite distributions on each continent. For the regional study, we appended several kimberlite ages (see the section 'Spatial and temporal links between breakup and kimberlites'), some made available since the publication of Tappe et al.[6] (see 'Data Availability'). Thus, we are confident that our analysis is as complete as is possible at present and can be revised as further age constraints become available.

### Statistical analysis of geotectonics and kimberlites

To quantitatively understand the link between kimberlites and continental breakup, we used the Tappe et al. kimberlite database[6] and a quantitative measure of the degree of fragmentation of the continental-plate system over geologic time[59] derived from plate-tectonic reconstructions[60]. This method involves the calculation of a continental perimeter-to-area ratio, whereby—during supercontinent stability—the ratio should be low, whereas during assembly and dispersal, it should be high[59] (Extended Data Fig. 1). We applied this method[59] using a revised set of continental polygons and rotations[17]. The model uses a unified set of cratonic polygons for the Neoproterozoic and Phanerozoic and was found to be internally consistent, drawing on a large synthesis of palaeomagnetic and geologic data and kinematic constraints[17]. This fragmentation index is not sensitive to active breakup processes, that is, the value remains high even after supercontinent breakup is complete. To account for this, we next calculated the rate of change of fragmentation ($\Delta F$), which quantitatively captures dynamic breakup processes. We did this by calculating the slope of the regression line using a symmetric, moving window of ±4 Myr through the fragmentation time series (Extended Data Fig. 1c,d). We next calculated the statistical relationship between the kimberlite distribution (Extended Data Fig. 1a,b) and $\Delta F$. Figure 1b shows the cross-correlation function between these two data series computed in R (ref. 61) (using the function acf in the stats package) for lags up to ±150 Myr. ccf($x, y$) computes the empirical (Pearson) correlations between two time series $x$ and $y$ at different lags (offsets), in which the lag $k$ value is the correlation between $x[t + k]$ and $y[t]$. The dashed blue lines in Fig. 1b show the approximate 95% confidence interval $t = 0.088$ ($n = 500$ observations), calculated using

$$t = \text{qnorm}\left(\frac{1 + C}{2}\right)\frac{1}{\sqrt{n}} \tag{2}$$

in which qnorm is the quantile function for the normal distribution, $C = 0.95$ and $n = 500$.

To further test the estimated approximately 30-Myr lag between changes in fragmentation and kimberlite eruption (Fig. 1b), we incorporated these measures into a Bayesian network using Uninet[62], a software package for high-dimensional-dependence modelling, previously used for analysis of tectonic and geochemical processes across different spatial and temporal scales[18]. We constructed a Bayesian network to calculate conditional rank correlations at increasing lags from 0 to 50 Myr. Unlike the standard cross-correlation function, this accounts for the prior effects of shorter lags, similar to the partial autocorrelation or cross-correlation. Inputs are a 1-Myr time series for the kimberlite count ($K_t$) from 500 Ma to present (that is, number of recorded kimberlite events per million years) and $\Delta F$ at lead times 0–50 Myr before kimberlite eruption (in increments of 5 Myr). The average rate of change at a given point in time is the slope of the regression line from a symmetric, moving window of ±4 Myr.

As we are interested in the correlation of (lagged) $\Delta F$ with $K_t$, and wish to remove the effects of shorter lags at each step (see ref. 18), we define the Bayesian network node hierarchy as: $\Delta F_t, \Delta F_{t-5}, \Delta F_{t-10}, \ldots, \Delta F_{t-50}, K_t$, in which $\Delta F_{t-5}$ is the time series for $\Delta F$ offset by $t - 5$ (that is, $\Delta F$ leading $K_t$). These nodes are used to construct a saturated Bayesian network, a network in which each node is connected by an arc to every other node in the network. Using Uninet, we can then compute the rank correlation of $\Delta F_t$ and $K_t$ (lag 0), the rank correlation of $\Delta F_{t-5}$ and $K_t$ conditional on $\Delta F_t$, the rank correlation of $\Delta F_{t-10}$ and $K_t$ conditional on $\Delta F_t$ and $\Delta F_{t-5}$ etc. The output from Uninet (Extended Data Fig. 2a) is the unconditional (Spearman) rank correlation for the first pair of nodes (with no lag) and conditional rank correlations for subsequent nodes (that is, accounting for the conditional dependence on nodes higher up in the network hierarchy; see ref. 18). As Kurowicka and Cooke[63] state (p. 33): "The conditional correlation of $Y$ and $Z$ given $X$ is the product moment correlation computed with the conditional distribution of $Y$ and $Z$ given $X$", or

$$\rho_{YZ|X} = \rho(Y, Z|X) = \frac{E(YZ|X) - E(Y|X)E(Z|X)}{\rho(Y|X)\rho(Z|X)} \quad (3)$$

This method gives a peak conditional rank correlation (0.51) at about 25 Myr (with an uncertainty of approximately ±4 Myr owing to the moving window used to calculate $\Delta F$), in keeping with the results of modelling and observations (Extended Data Fig. 2a). We repeated this procedure to analyse the relationship between $\Delta F$ and LIPs using the well-established ages of LIP magmatism[25] (Extended Data Fig. 6). Here the input is a 5-Myr-resolution series, in which LIPstart is the total number of LIP events with a start date falling in each 5-Myr interval and $\Delta F$ is the slope of the regression line for continental fragmentation (over a 9-Myr window), again estimated every 5 Myr. Using a simple saturated Bayesian network, we compute the correlation of $\Delta F$ and LIPstart, or corr($\Delta F$, LIP), that is, for which LIPstart precedes $\Delta F$ (Extended Data Fig. 6b).

## Spatial and temporal links between breakup and kimberlites

We next evaluated the spatial and temporal relationship between continental breakup and kimberlite magmatism, taking Mesozoic Africa and South America (affected by the same Gondwana breakup episode) as a case study in which data confidence (for example, kimberlite ages and rifting history) is high. We used the same kimberlite database[6] as in our global analysis above, again adding several radiometric ages for the Brazil region[64–68]. Palaeogeographic reconstructions were obtained from the open-source plate-tectonic modelling software GPlates[23,69].

To mitigate spatial biases in the data, we undertook a spatiotemporal cluster analysis of the kimberlite fields. Clusters are defined as groups of kimberlites that are close enough in space and time that they could plausibly be attributed to the same initiating event. By using a clustering approach, we mitigate the overrepresentation of similar ages within geographically restricted areas. This eliminates some of the reporting bias in the catalogue, as eruptive products are more likely to be identified and radiometrically dated close to previously discovered kimberlites.

For the purposes of this analysis, a kimberlite is defined as being part of a cluster if there is another recorded event within ±5 Myr and a 25-km radius. Not all kimberlites are clustered (Extended Data Fig. 3a). We chose a 25-km radius as this is considered to represent a lower-diameter length scale of kimberlite fields at the surface[42]. Because kimberlite eruptions are generally considered to be rapid and short-lived (with inferred durations of hours to months[2]), our time window of 5 Myr is principally intended to capture uncertainties in radiometric ages (±2.5 Myr), which is considered representative for the studied kimberlites[6,70].

To perform spatiotemporal clustering, we used the R packages sf[71] (for handling spatial vector data), sp (further classes and methods for spatial data) and geosphere[72] (for spherical trigonometry). We first converted the longitude/latitude point coordinates of each kimberlite observation to an sf object, then convert from sf to a spatial points object (sp). We used the distm function (from geosphere) to calculate a matrix of distances between all pairs of kimberlites in the catalogue (that is, the shortest distance on the WGS84 ellipsoid). We also generated a matrix of time differences (dt) between each pair of kimberlites (using their radiometric ages[6]). The method to allocate clusters is defined as follows:

1. Step through each kimberlite in the catalogue $K_i$, for $i = 1:n$ (in which $n$ is the total number of kimberlites).
2. Identify any further kimberlites $K_j, j = (i + 1):n$, in which the distance between $K_i$ and $K_j$ is ≤25 km and the time lag abs(dt) is ≤5 Myr. If any $K_j$ meets these criteria, they are added to the current cluster. If $K_i$ has no 'neighbours' meeting both conditions, it is not clustered.
3. For each kimberlite $K_j$ added to the cluster, we search for any subsequent kimberlites, $K_m, m = (j + 1):n$, within a distance ≤25 km and absolute time lag abs(dt) ≤5 Myr relative to $K_j$. This allows the cluster to expand beyond the original radius but eliminates any potential for several/overlapping clusters.

This procedure ensures that every kimberlite is counted precisely once, either as part of a cluster or alone (if it is greater than 25 km and 5 Myr from any neighbours).

Next we measured the shortest distance and time lags between kimberlites and the margins of neighbouring, coeval rift systems—using the mapped extent of continent–ocean boundaries (COBs) through time[60] (Extended Data Fig. 3a) and their associated breakup ages[23,69]. Using the clusters defined above, we can then calculate the average distance and average time lag for each cluster.

As there are several boundary sections with breakup occurring at different times, we applied some simple rules to determine distance and lag. The COB shapefile comprises polylines with an attribute RiftAge denoting the age of continental separation of each rift section. We focused our attention on the major rift systems adjacent to the kimberlites: the Africa–Madagascar/Africa–Antarctica, the South Atlantic and the Central Atlantic, with the last two relating to both African and South American kimberlite fields (Extended Data Fig. 3a). We also consider those rift systems bounding the North American kimberlite fields: the Central Atlantic, Greenland–North America and Arctic region (Alaska–northern Canadian margin)[24].

For each kimberlite, we calculated: (1) the shortest distance $d$ from the kimberlite to each individual COB line segment and (2) the time lag $l$ between breakup time (for each COB line segment) and the kimberlite catalogue age. Lags can be negative, with eruption occurring before breakup, or positive, for which eruption occurs after breakup (Fig. 1c,d). Distances were calculated using the dist2Line function from the R geosphere package, which gives the shortest distance (in metres)

between points and polylines or polygons with longitude/latitude coordinates on the WGS84 ellipsoid.

This approach leads to several COB line segment associations $\{d_{ij}, l_{ij}\}$ defined as a distance $d$ and lag $l$ for all kimberlites $i$ and COB segments $j$ within a region. However, we are interested in identifying the most likely association for each kimberlite—the closest in time and distance. We first eliminate any cases of kimberlite eruption occurring more than 40 Myr before continental breakup (removing any associations with $l_{ij} < -40$). This focuses the analysis on kimberlites that formed during rift-related thinning of lithosphere and initiation of convective instabilities, as well as after breakup itself. We chose 40 Myr as an appropriate cut-off based on (1) inspection of the cross-correlation between global kimberlite occurrence and continental fragmentation (Fig. 1b) and (2) the observation that rifting tends to lead breakup by 30–40 Myr (refs. 19,24).

Second, we eliminated any cases with lags >60 Myr ($l_{ij} > 60$) after breakup, as we are primarily interested in the approximately 100-Myr interval after rift onset. For Africa and South America, breakup is estimated to occur between 6 and 40 Myr after rift onset[24]. The global cross-correlation between fragmentation and kimberlite eruption falls below the 95% confidence interval ≈ 60–70 Myr after continental fragmentation, supporting this as a cut-off point. Third, we impose a maximum distance of 2,000 km (eliminating $d_{ij} \geq 2 \times 10^6$ m), capturing the craton scale for Africa and South America (Extended Data Fig. 3a).

We finally eliminated any duplicate cases (in which a kimberlite is associated with more than one rift section) by assigning priority to shorter absolute lags. This means that each individual kimberlite is only counted once and associated with a single rift section. This procedure yields a dataset of kimberlite locations (point coordinates), eruption age, lag and distance to a single (most likely) associated COB. We then use this information to calculate the average age, time lag to breakup and COB distance for each cluster. Data for the kimberlite clusters are shown in Fig. 1c,d and Extended Data Fig. 3c,d.

### Kimberlite migration rates

To estimate the initial rate of kimberlite migration into the cratonic interior, we initially used the general apparent increase in distance between rifted margins and kimberlites in the tens of millions of years following rift onset (Fig. 1c and Extended Data Fig. 3c). Our data show that kimberlites typically first appear >200 km inboard of rifted margins (Fig. 1c). This is possibly because they can only form where the craton is sufficiently thick (≥150 km; Extended Data Fig. 3a,b) for diamonds to be stable[54] and chemical conditions in the basal lithosphere to be optimal for kimberlite generation (Fig. 3b). For instance, the surface boundary of the Kaapvaal Craton is located about 400 km inboard of the South Atlantic rift[73] and thickens towards the east. Kimberlite magmatism therefore initiates some distance (that is, typically several hundred kilometres) inboard of rifted margins.

### Uncertainty analysis

To account for uncertainties in kimberlite age, estimated rift location and rifting age, we repeat the above analysis as a Monte Carlo simulation of 5,000 runs. Using ages and uncertainty estimates[6], kimberlite age is sampled from a uniform distribution on the interval $[a_i - e_i, a_i + e_i]$, in which $a_i$ is the age and $e_i$ is the estimated uncertainty of each kimberlite $i$. Age uncertainties are typically assumed to be more centrally distributed (for example, normally distributed); however, there are not sufficient data to robustly parameterize a distribution for each individual kimberlite/dating method in the database. By choosing a uniform distribution, we take a conservative approach to error estimation.

Breakup age (for each individual rift section) is sampled from a uniform distribution $[b_m - 0.025b_m, b_m + 0.025b_m]$, in which $b_m$ is the rift breakup age (that is, an uncertainty of ±2.5% of the original age estimate to reflect uncertainties in tectonic reconstructions[18]).

For a set of regional kimberlites, the uncertainty in estimated distance to a rift section is spatially correlated. This is because rift sections form/propagate as a connected, broadly linear structure, and the relative positions of the kimberlites are not affected by uncertainty in the rift location. To preserve these spatial properties, we model uncertainty in rift location by sampling an offset distance (in km) from a uniform distribution on the interval [−90, 90] km for each rift section in turn. This value was chosen because COBs are zones rather than precise linear boundaries and the global mean half-width of the COB 'transition' zone is about 90 km (ref. 74). For each iteration of the Monte Carlo simulation, the sampled offset is applied to the estimated distances from the set of kimberlites in the region to that rift section. This effectively moves the entire rift section closer to or further away from the continental landmass by a random amount at each iteration. Within a single iteration, the spatial relationship between kimberlite points is preserved, giving a more robust representation of the true uncertainty.

### Incorporation of North American kimberlite data and uncertainty in rift association

To expand the analysis to include kimberlites in North America, it was necessary to modify our methods to accommodate longer timescales and distances. For North America, typical distances between rift margins and kimberlites in the continental interior are greater. This can lead to longer migration times from the rift boundary (COB) to the point of eruption and longer lags. In North America, breakup—which started in the Lower Jurassic—is estimated to have occurred approximately 10–76 Myr after rift onset[24], longer than for South America and Africa (in which breakup occurred about 6–40 Myr after rift onset[24]). We therefore need to consider a wider window of time, both pre-breakup and post-breakup.

To implement this, we define an extra measure, lag relative to rifting ($l_r$). This lag, in conjunction with the computed rift–kimberlite distances ($d$), can more easily accommodate the variation in timescales between rifting and breakup and, more importantly, can be used to directly estimate migration rates $r = d/l_r$, in which $r$ is the rate (km Myr$^{-1}$).

We set some basic constraints to focus on the range of plausible rift–kimberlite associations within any given region. The maximum distance for association was increased to 3,000 km. We extended the time window, using lag relative to rifting as the base measure, and limit associations to kimberlites erupted at or after the onset of rifting (but not before rifting), within the bounds of known age uncertainties. We also limit migration rate ($r$) to 100 km Myr$^{-1}$, to avoid making highly unlikely associations between rift activity and very distant kimberlites within a short time window. These conditions, applied jointly, enable us to capture the full range of plausible rift–kimberlite associations.

For each rift and kimberlite pair, we calculate (1) the distance $d$, (2) lag relative to breakup $l_b$ and lag relative to rifting $l_r$ and (3) migration rate $r$. As before, this yields several potential rift associations for each kimberlite. After applying the lag and distance constraints outlined above, we then identify, for each kimberlite, (1) the rift section with the shortest absolute lag relative to breakup $l_b$, 'min lag', and (2) the rift section with the shortest distance, 'min dist'. By considering these two 'endmember' models of association, we can provide a reasonable estimate of the model uncertainty in our estimates for distance, lags and migration rate; that is, we are considering the range of possible initiating rifting events for each kimberlite in the analysis.

As before, we apply Monte Carlo simulation to capture uncertainty in kimberlite age, rift age and rift–COB distance (see method outlined above).

For each region (Africa, North America and South America), we generate an ensemble of 5,000 runs, generating a set of distance, lag and migration rate for each kimberlite, using both 'min lag' and 'min dist' models of association. Results are presented as a probability density

plot (Fig. 1e), capturing both model and data uncertainty for migration rate (Supplementary Dataset 2). Extended Data Fig. 4 presents probability density plots for the two individual endmember association models ('min lag' and 'min dist') for (1) the kimberlite clusters (Supplementary Dataset 2) and (2) individual kimberlites, for all three continents combined.

This analysis, incorporating age, distance and model (kimberlite–rift association) uncertainty, gives the following migration rate estimates for North America, South America and Africa:

- Clustered data: median 20.7 km Myr$^{-1}$, 10th–90th percentile range of 5.6–58.8 km Myr$^{-1}$;
- Individual kimberlites: median 25.0 km Myr$^{-1}$, 10th–90th percentile range of 6.3–60.4 km Myr$^{-1}$.

This variability is to be expected, because of the complex temporal and spatial distribution of rifts, and their relation to kimberlites. We take a conservative approach to model uncertainty by considering the two endmember cases for initiating kimberlite eruption: evaluating migration rates for rifting events closest in both space and time. This approach gives robust upper-bound and lower-bound estimates of migration rate for the three main global kimberlite regions.

Our analysis uses 87.5% of all kimberlites younger than 240 Myr old (that is, erupted since the onset of Pangaea breakup).

## Relationship between kimberlites and lithospheric thickness

We analysed the relationship between lithospheric thickness and kimberlite occurrence (Extended Data Fig. 3a,b) using the LITHO1.0 (ref. 75) and LithoRef18 (ref. 76) models of the crust and lithosphere. We performed this analysis using open-source GIS software QGIS (v. 3.16; https://qgis.org/), with further open-source GRASS GIS processing tools (https://grass.osgeo.org/). Starting with a vector point map of lithospheric thickness from LITHO1.0, we performed a surface interpolation with regularized splines under tension, using the GRASS function, v.surf.rst, to generate a 0.1° cell size raster map of lithospheric thickness. We then used the 'Point Sampling Tool' QGIS plugin to obtain lithospheric thickness (Extended Data Fig. 3b) at each kimberlite location from the interpolated raster map (Extended Data Fig. 3a). Using LITHO1.0, and focusing on the younger subset of kimberlites (<250 Ma; $n$ = 722), we obtain relatively high lithospheric thicknesses (mean = 207 km; Extended Data Fig. 3b).

As a sanity check, we tested another, more recent global reference model of LAB depth (LithoRef18) derived by joint inversion and analysis of several datasets[76]. This model shows the main peak in kimberlites corresponding to lithospheric thicknesses of 160–170 km (overall mean = 167 km)—clearly lower than those of LITHO1.0 (Extended Data Fig. 3b). Both models indicate that most kimberlites are associated with cratonic lithosphere 150–250 km thick, although we favour the lower end of this range for most cases (Extended Data Fig. 3b).

## Scaling analysis of Rayleigh–Taylor instability chains

Motivated by the observation that kimberlite magmatism typically migrates laterally over time (Fig. 1c,e), we investigated whether an initial convective instability in the basal lithosphere can trigger further instabilities in an organized manner. We first developed a simple physical model for spatial and temporal organization within a series of Rayleigh–Taylor instabilities based on analytical models for individual instabilities and derived a scaling law for the lateral propagation rate of an instability chain.

The thermal boundary between the lithosphere and asthenosphere can be represented by a simple model of a viscous fluid layer of thickness $b$ representing lithospheric keel that overlies a less dense viscous fluid representing asthenospheric mantle[33]. Analytical solutions for Rayleigh–Taylor instability are available for a starting situation in which the interface between the two fluids has a sinusoidal displacement of

horizontal wavelength $\lambda$ and amplitude $w_0$, and $w_0 \ll \lambda$. The amplitude of an instability grows as

$$w = w_0 e^{qt} \tag{4}$$

in which $q$ is the growth rate and $t$ is time. The solutions give the scaled horizontal wavelength of the fastest growing instability, $\lambda_d^*$, and the corresponding scaled growth rate, $q_d^*$. For the geologically relevant solutions of interest, $\lambda_d^*$ is 2–4 and $q_d^*$ is on the order $10^{-1}$ (Extended Data Table 1).

To apply these analytical solutions, note that length is scaled by the mean starting-layer thickness, $b$, and time is scaled by $v/g'b$, in which $v$ is the kinematic viscosity, $g' = g\Delta\rho/\rho$ is reduced gravity, $\Delta\rho$ is the difference in density between the upper and lower layers and $\rho$ is the reference density. Hence, the dominant wavelength of lithospheric root instabilities is

$$\lambda_d = \lambda_d^* b \tag{5}$$

and the characteristic time period for growth of the dominant instability is

$$\tau_d = \frac{v}{q_d^* g' b} \tag{6}$$

Now we consider a single instability of dominant wavelength. Because $\tau_d$ is an e-folding time, soon after $t = \tau_d$, the instability will achieve terminal downward velocity, detach from the upper layer and sink into the lower layer. Because fluid has been removed from the upper layer, the new thickness of the upper layer will be less than $b$ above the site of the detachment and the wavelength of the thinned patch of the upper layer will be approximately $\lambda_d$. This situation resembles the original initial condition, and equation (4) with $w_0 \approx b$ predicts growth of the resulting instabilities. Thus, new instabilities will develop at horizontal distances $\pm\lambda_d$ relative to the initial instability. If this process repeats, the second-generation instabilities will grow, detach and trigger third-generation instabilities. Because this is a convective process, second-generation instabilities grow and detach before the thinned patch of upper layer above the initial instability can regain its original thickness by conductive cooling and thickening. Hence, the topographic gradient on the base of the upper layer will be greater above the outer edges of the second-generation instabilities, relative to the initial instability. Thus, third-generation and later instabilities will be initiated at progressively greater distances from the initial instability.

This simple physical model predicts that an initial Rayleigh–Taylor instability should initiate a chain of further instabilities, in which successive instabilities occur at horizontal distances of about $\lambda_d$ outboard of the initial instability with a periodicity of about $\tau_d$. The lateral propagation rate of this chain of events is therefore expected to scale as

$$U \approx \frac{\lambda_d}{\tau_d} \tag{7}$$

Substituting equations (5) and (6) gives

$$U \approx \lambda_d^* q_d^* \frac{g' b^2}{v} \tag{8}$$

which shows that $U$ scales as $b^2$. Typical values for $U$ are plotted in Fig. 3a and shown in Extended Data Table 1. The values of $g'$ and $b$ were estimated by fitting geotherms to xenolith $P$–$T$ data (see below). A viscosity of $4 \times 10^{15}$ m$^2$ s$^{-1}$ has been used previously for modelling the basal lithosphere, including in xenolith geotherm modelling[77,78], and we find that this value gives a good match between our scaling

law, numerical models and observed migration rates of kimberlite magmatism (Figs. 1c,e and 3a and Extended Data Fig. 3c).

## Thermomechanical models

To advance on the above footing, we assessed the influence of the more complex natural geometry and rheology of rifting cratonic lithosphere, using numerical forward models conducted with the finite-element code ASPECT[79–81]. These models solve the conservation equations of mass, energy and momentum for materials with viscoplastic rheology[82]. In particular, we account for temperature, pressure and strain-rate dependent rheologies based on experimentally derived flow laws for dislocation and diffusion creep combined with plasticity (Extended Data Table 2). Our model is kinematically driven through velocity boundary conditions at lateral sides. The model reproduces the formation of a narrow rift that is bounded at depth by steep LAB gradients resulting in pronounced rotational flow patterns[27]. This flow destabilizes the base of the thermal lithosphere (cratonic keel), producing Rayleigh–Taylor instabilities that evolve self-consistently by sequential destabilization. In further agreement with previous models, breakup is delayed owing to rift migration[83].

We will now briefly describe the geometrical, mechanical and thermal setup of our geodynamic model, as well as its limitations. For a more detailed description of the functionalities of ASPECT and solution techniques, we refer the reader to refs. 79,80 and https://aspect.geodynamics.org/manual.pdf.

The model comprises a domain of 2,000 × 300 km and 800 × 120 elements in the $x$ (horizontal) and $y$ (vertical) directions, respectively. The second-order finite elements are visualized by dividing them into four squares, leading to an effective resolution of 1.25 km. The model comprises four layers: 20-km-thick upper crust, 15-km-thick lower crust, 125-km-thick mantle lithosphere and 140-km-thick asthenosphere (that is, beneath 160 km depth). We use a wet quartzite[84] and wet anorthite[85] flow law for the upper and lower crust, respectively, dry olivine rheology[53] for the mantle lithosphere and wet olivine[53] for the asthenosphere. The model involves frictional strain softening, for which we linearly reduce the friction coefficient with a factor of 0.25 for brittle strain between 0 and 1. For strains greater than 1, it remains constant. We also account for viscous strain softening by decreasing the viscosity derived from the ductile flow law with a factor of 0.25 between viscous strains 0 and 1. All rheological and mechanical model parameters are provided in Extended Data Table 2. For visualization purposes, we distinguish a 30-km-thick layer beneath some parts of the lithosphere as a simplified representation of variably metasomatized thermal boundary layer of the lower lithosphere. To initiate rifting in a predefined area, we use a 25-km-thick upper crust and 100-km-thick mantle lithosphere, representing typical mobile belt conditions[75]. The above layer thicknesses gradually transition to ambient lithosphere over a distance of around 200 km.

We use velocity boundary conditions with a total extension velocity of 10 mm per year. For simplicity, we keep the right-hand model side fixed, but we verified that the conclusions do not change if extension velocities are symmetrically distributed between the lateral boundaries. At the bottom boundary, we prescribe a constant vertical inflow of material that balances the outflow through the lateral model sides. The top boundary constitutes a free surface[81].

Temperature boundary conditions feature a constant surface temperature of 0 °C and a bottom temperature of 1,420 °C, that is, normal-temperature mantle with potential temperature of around 1,320 °C. Lateral boundaries are thermally isolated. The initial temperature field of each model column results from the 1D thermal equilibrium defined by the boundary conditions, the crustal radiogenic heat contribution and the initial depth of the thermal LAB, that is, the 1,350 °C temperature isotherm, which at first coincides with the compositional LAB. The sublithospheric temperature increases adiabatically with depth. To smooth the initial thermal gradient across the LAB, we equilibrate the entire thermal state of the model for 30 Myr before the onset of extension. All thermal parameters are listed in Extended Data Table 2.

The following model limitations have to be kept in mind when interpreting the results. In this model, we focus on first-order thermomechanical processes and do not explicitly account for chemical alterations, melt generation and magma ascent (that is, these are not ignored but are considered separately using different approaches). For simplicity, we assume that the initial depth of the LAB does not vary on the thousand-kilometre scale. Nonetheless, we conducted further model runs with gradual changes in LAB depth and the conclusions remained the same. Further effects such as large-scale flow patterns related to mantle convection, the impingement of mantle plumes and along-strike lithospheric heterogeneities may exert local effects on the formation of Rayleigh–Taylor instabilities that we have to neglect in our generic modelling strategy.

Sublithospheric viscosity and, in particular, activation energy constitute important parameters for the development of instabilities[86]. In agreement with observations of seismic anisotropy in the upper mantle, shallow-asthenosphere viscosity in our models is dominated by dislocation creep, for which we use experimentally derived wet olivine activation energy values of 480 ± 40 kJ mol⁻¹ (ref. 53) that are representative for the shallow asthenosphere. Notably, these values for dislocation creep lie within the independently derived activation energy range of 360–540 kJ mol⁻¹ (ref. 86). We conducted alternative modelling runs in which we varied the asthenospheric activation energy within experimental uncertainties while keeping all other parameters identical. When we lower the activation energy to 440 kJ mol⁻¹, the shallow asthenosphere and thermal boundary layer exhibit a viscosity that is about two times smaller than for the reference case. This model generates lateral propagation/migration rates for the instabilities that are roughly two times larger than for the reference model, which is in agreement with analytical model 3 that constitutes the low-viscosity endmember of the analytical modelling suite (Fig. 3a and Extended Data Table 1).

## Xenolith geotherms

To estimate the typical thickness, temperature and density of the thermal boundary layer between the lithosphere and asthenosphere—for application in the delamination and melting calculations—we applied the numerical geotherm calculation approach of Mather et al.[77], which uses peridotite $P–T$ estimates from various kimberlites. We used the FITPLOT program (refs. 78,87), which takes the thickness and thermal properties of the crust as inputs and determines the thicknesses of the rigid lithospheric mantle and the basal lithospheric thermal boundary layer by minimizing the misfit between the calculated geotherm and the xenolith $P–T$ data. Full details of this method and the xenolith datasets used are provided in ref. 77. Calculated geotherms and thermal-boundary-layer thicknesses are plotted in Extended Data Fig. 8. The total thickness of the thermal boundary layer is estimated to be 35 km. The corresponding thickness of the unstable viscous layer, $b$, that should be used in the analytical Rayleigh–Taylor instability models is approximately half this value, that is, 17.5 km, as the LAB sits near the middle of the thermal boundary layer (Extended Data Fig. 8). The temperature change across the thermal boundary layer, $\Delta T$, is about 150 °C. Thus, the mean density contrast that drives Rayleigh–Taylor instabilities is $\Delta\rho = \rho\alpha\Delta T/2 = 10$ kg m⁻³, in which $\alpha = 4 \times 10^{-5}$ C⁻¹ is the thermal expansivity[78], $\rho = 3,300$ kg m⁻³ is the reference density and $\Delta T/2$ is the mean temperature difference relative to the asthenosphere. The corresponding reduced gravity is $g' = 0.03$ m s⁻², which was used in the analytical Rayleigh–Taylor instability models with a constant-density upper layer (models 1, 2, 3 and 6 in Fig. 3a and Extended Data Table 1). A value of $g' = 0.06$ m s⁻² was used for the analytical models with linearly decreasing density (models 4 and 5), so that all the analytical model results in Fig. 3a and Extended Data Table 1 have the same mean driving density. This driving-density contrast is a lower bound because

it does not include possible metasomatic density increases within the upper thermal boundary layer, but even without compositional effects, it is sufficient to drive convective removal of lithospheric keel (Figs. 2 and 3a).

## Melting calculations

Neither the analytical modelling of Rayleigh–Taylor instability chains nor the numerical modelling carried out in ASPECT directly predict where and when melting occurs. However, both analytical and numerical models show that convectively driven lithospheric delamination will result in adiabatic upwelling that could potentially cause mantle melting. In the analytical models, the terminal velocity of an individual isolated downwelling Rayleigh–Taylor instability in the lithospheric keel is expected to approximate the Stokes velocity for a sphere of radius $\lambda_d$, which is $g'\lambda_d^2/3\nu$ (ref. 88). We assume that the upward return flow exactly balances the downwelling and occurs in a cylindrical annulus of width $2\lambda_d$ that surrounds the downwelling plume. With this geometry, the cross-sectional area of the upwelling annulus is eight times that of the downwelling plume, so the mean upwelling velocity will be one-eighth of the downwelling velocity. The mean upwelling velocity is therefore expected to be

$$V = \frac{g'\lambda_d^2}{24\nu} \qquad (9)$$

The upwelling velocities estimated for the Rayleigh–Taylor instability models considered here are a few tens of kilometres per year (Extended Data Table 1). The thermal Péclet number for the upwelling flow is estimated using

$$Pe = \frac{bV}{\kappa} \qquad (10)$$

in which $\kappa$ is the thermal diffusivity and $b = 17.5$ km is the net upward motion of the return flow, equivalent to half the thickness of the keel layer that detaches (see the previous section 'Xenolith geotherms'). The predicted Péclet numbers are approximately 10 (Extended Data Table 1), which suggests that upwelling will be adiabatic, meaning that the return flow should not lose substantial heat as it approaches the base of the thinned lithosphere. Thus, if part of the detached lithospheric keel becomes entrained in the upward return flow, it can move to a new depth beneath the thinned lithosphere that is shallower than its original depth, whilst maintaining its original potential temperature. Decompressional melting can potentially occur in this situation if sufficient volatiles are present. If rifting and continental breakup organizes the downwelling patches to resemble a detaching sheet with little along-strike variation, then the factor in the denominator of equation (9) would be 9 rather than 24. In this case, the return-flow upwelling would be stronger and still more likely to be adiabatic and to cause melting.

The ASPECT modelling confirms the expectation of adiabatic upwelling and indicates the relative locations and strengths of the convective upwellings through time (Extended Data Fig. 9). The strongest convective upwelling at any time is always associated with the leading convective instability in the propagating chain. The leading cell has two upwelling regions, one on either side of the central downwelling. The upwelling closer to the rift is the stronger of the two, although both upwelling regions are adiabatic and can potentially drive melting (subject to further conditions). Typical upwelling velocities in the leading cell are roughly 30 km year$^{-1}$, in agreement with the analytical model. Earlier convection cells in the propagating chain remain visible to the rift-ward side of the leading cell. Circulation in the older cells is weaker than in the leading cell but may still be adiabatic.

Another factor needed to enable decompressional melting of normal-temperature mantle beneath thick cratonic lithosphere is that the upwelling mantle is enriched in volatiles. We have already demonstrated that the thermal boundary layer, which we propose is convectively removed by propagating instabilities, probably contains both hydrous and carbonate-rich incipient silicate melts (Fig. 3b). The ASPECT modelling shows that this volatile-enriched, metasomatized keel detached by the downwelling limbs of the small-scale convection cells is partially recirculated in the upwelling limbs. The upwelling limb on the rift-ward side of the leading convection cell contains most recirculated keel, together with a notable component of background asthenospheric mantle. Older convection cells may also contain volatile-enriched material but with a greater proportion of asthenospheric material. We note that the detailed spatiotemporal pattern of recirculated keel material is difficult to interpret because the convection cells interact with the bottom of the computation box at 300 km. Nevertheless, the ASPECT modelling shows adiabatic upwelling of asthenosphere that is variably enriched with recently detached metasomatized keel material. Thus, it is feasible to infer that migration of the instability chain will potentially be accompanied by kimberlite melting. The bulk of melting is predicted to occur across the footprint of the leading convection cell in the instability chain, and smaller fluxes of melting in the wake of the leading instability may be expected in some cases.

To determine whether this inferred mechanism of melting can explain kimberlite occurrences, we estimated the characteristic volume of melting associated with convectively driven upwelling of mantle at the base of thick lithosphere. We assumed adiabatic upwelling of normal mantle temperature (potential temperature of 1,300 °C) and calculated the degree of melting as a function of depth to the base of the rigid lithosphere (that is, mechanical boundary layer) and also the $H_2O$ content of the primary melt, for adiabatic upwelling of mantle with bulk water contents of 0.1, 0.15 and 0.2 wt% (Fig. 3c). We used the hydrous decompressional melting parameterization of Katz et al.[47] for these calculations, as (1) it is a well-known parameterization underpinned by a large database of experimental petrology results and (2) it is a parameterization of volatile-rich melting, which is relevant to the kimberlite mantle source, which most agree is enriched in volatiles including $H_2O$ (refs. 1–3,15,41,45,46). The calculation and plotting scripts for all the calculations described below are freely available (see 'Code availability').

We first calculated the degree of melting to a given depth for an adiabatic melting column with various bulk water contents (Fig. 3c). We used the MORMEL package (https://github.com/smj75/mormel) as a convenient implementation of the chosen hydrous melting parameterization. Our calculations predict that convective removal of keel followed by adiabatic upwelling up to depths between 190 and 150 km of a keel/asthenosphere mixture with bulk water content of 0.1–0.2 wt% yields low-degree decompressional melting (<1%), which is consistent with expectations for kimberlites[42]. We also plotted the associated cumulative total melt productivity for such melting (Fig. 3c), to compare with kimberlite magma volume estimates based on surface observations. We assumed an upwelling rate of 30 km year$^{-1}$ within the adiabatic melting column, which is typical of the upwelling limbs of the convection cells in our geodynamic models. Melt productivities of up to 0.15 km year$^{-1}$ for a 1D vertical upwelling column are predicted when mantle source with bulk water contents of 0.1–0.2 wt% wells up to 150 km. Thus, we infer that, if decompression generates a mean 1D melt productivity of 0.1 km year$^{-1}$ across a footprint comparable with that of a typical Rayleigh–Taylor instability, with diameter 50 km and footprint area around 2,000 km$^2$, then a total magma volume flux of 200 km$^3$ year$^{-1}$ could result. This flux is broadly consistent with estimated volumes for kimberlite clusters[41] and demonstrates that the system is not limited by magma production at depth. The predicted melt flux could be higher if melt from convection cells in the wake of the leading cell is also considered. Our calculations also show that kimberlite melting can occur in the absence of plumes; yet, more melting would be expected in cases in which plumes are present.

Where hydrous, carbonate-bearing peridotites are involved in melting, $H_2O$ is expected to be present above the reduced solidus, within the thermal boundary layer of keel (Fig. 3b,c). We calculated the total water content of the melt for the decompressional melting scenarios above (Fig. 3c) using the equation in ref. 47. Our calculations predict between 9% and 14% $H_2O$ in the melt (Fig. 3c), following partial to full convective removal of lithospheric keel. This range overlaps the estimated compositions in Table 1 of Sparks et al.[46], but we would expect our calculation to provide an upper limit rather than an estimate for $H_2O$ content in the erupted melt. The composition of the primary melt is expected to evolve substantially during ascent and is not expected to match the composition of the erupted magma: some $H_2O$ will be lost from the melt by exsolution and by reaction with wall rocks and xenoliths as the melt ascends over 150 km through the lithosphere[46]. In general, the high volatile contents and low melting degrees predicted by our simple calculations are in line with more detailed compositional calculations and petrological experiments on model systems[45,46].

## Phase diagrams and phase equilibria

We investigated phase equilibria to determine whether the conditions expected in the thermal boundary layer (Fig. 3b,c) should give rise to melting, specifically in cases in which the layer is disturbed by a migrating convective instability (Fig. 2 and Extended Data Fig. 9). Because our principal motivation here is to establish the $P$–$T$/chemical conditions immediately before melting onset, we do not focus on melts of kimberlite-like composition; in our framework, these represent the net product of melting induced by convective instability. Rather, we consider phase equilibria for peridotites (Fig. 3b), which dominate the cratonic keels. We consider experimentally determined solidi for peridotites under oxidized and reduced conditions[30]. Because a large body of theoretical and petrological evidence supports high $CO_2$ and $H_2O$ contents for erupted kimberlites[1–3,45,46,89], we include the oxidized solidus for peridotite in the presence of $CO_2 + H_2O$, first defined by Wallace and Green[90] (see refs. 91,92 for further discussion) and subsequently extended to higher pressures by Foley et al.[93] and Pintér et al.[94]. We also incorporate the reduced solidus for peridotite in the presence of $CH_4 + H_2O$, which was experimentally defined by Taylor and Green[95] (see refs. 91,92). We plot the graphite–diamond phase boundary experimentally defined by Day[54]. The phase equilibria show that, at $P$–$T$ conditions considered typical of the thermal boundary layer, and under expected conditions, incipient melts containing both $CO_2$ and $H_2O$ are expected to be present. This is supported by fluid inclusions in diamonds, which show that some hydrous and carbonatitic fluids may be in equilibrium in the diamond stability field[96].

Although we are primarily concerned with melting of carbonated and hydrous mantle peridotite, for completeness, we also consider the $CaO$–$MgO$–$Al_2O_3$–$SiO_2$–$CO_2$ model system of Dalton and Presnall[97], as this system is commonly implicated in generating melts of carbonatitic, kimberlitic and melilitic composition[43]. Using several representative experimental melt compositions (that is, JADSCM-7 and JADSCM-14) from Gudfinnsson and Presnall[43], we can confirm that the carbonate-bearing lherzolite solidus falls within the thermal boundary layer of the lithosphere, demonstrating that, for the $CaO$–$MgO$–$Al_2O_3$–$SiO_2$–$CO_2$ system, generation of incipient carbonate melts is expected to occur in the thermal boundary layer.

We plotted estimated $P$–$T$ conditions of lherzolite/peridotite nodules from Baptiste et al.[98] in Extended Data Fig. 7; here we converted some depths from published compilations to pressures, using

$$P = \rho g h \qquad (11)$$

in which $\rho$ is the density of the lithosphere, $g$ is acceleration due to gravity and $h$ is the depth. In our calculations, we took $\rho = 3{,}200$ kg m$^{-3}$, based on an assumed lithospheric thickness of 180 km, comprising a crustal thickness of 45 km and crustal density of 2,750 kg m$^{-3}$ (ref. 99)

and a lithospheric thickness of 135 km and density of 3,300 kg m$^{-3}$ (using a middle value from ref. 100).

## Statistical analysis of isotope variations

We compiled existing data to assess whether any temporal changes in kimberlite geochemistry are detectable during the abrupt reduction in lithospheric thickness in the Kaapvaal Craton at around 117–108 Ma, suggested to be a consequence of lithospheric delamination[40]. Here we first analysed whole rock $(^{87}Sr/^{86}Sr)_i$, $(^{143}Nd/^{144}Nd)_i$ and $(^{206}Pb/^{204}Pb)_i$ from Smith et al.[50], updating the radiometric ages of the kimberlites where necessary (note that the subscript $i$ denotes the initial isotope ratio, that is, at the time of formation). It has already been noted that the group II kimberlites are restricted to the main phase of Pangaea breakup[101]. Because kimberlites are particularly susceptible to syn-emplacement and post-emplacement alteration by crustal fluids, whole-rock Sr and Pb isotope data may not provide a very accurate record of mantle-source compositions[102]. Therefore, we also studied the temporal variation in perovskite initial $^{87}Sr/^{86}Sr$ ratios[103] and whole-rock $^{143}Nd/^{144}Nd$ ratios[104] in Cretaceous southern African kimberlites (Extended Data Fig. 10). The plots show the Sr and Nd isotope compositions of the mica–amphibole–rutile–ilmenite–clinopyroxene (MARID) endmember defined from kimberlite xenoliths and is thought to derive from a lithospheric mantle source that has been variably contaminated by recycled crustal components, possibly during a subduction event at about 2.9–3.2 Ga (ref. 105). The plots also show the composition of the kimberlite melt endmember of ref. 105, largely defined from analyses of phlogopite–ilmenite–clinopyroxene (PIC) kimberlite xenoliths. This endmember is interpreted to reflect a mantle source that has been metasomatized by kimberlite melts[105] and it is interesting to note that this endmember has very similar Sr, Nd and Pb isotope compositions as average Cretaceous to Cenozoic African carbonatites[106].

We next used conjugate partitioned recursion (CPR) to evaluate the potential presence of step changes in the above isotopic datasets (Extended Data Fig. 10). This iterative algorithm uses binary partitioning by marginal likelihood and conjugate priors (CPR) to identify an unknown number of change points[107]. If the marginal likelihood favours a change-point model, then the algorithm defines a change point and two-sigma uncertainty bounds of the two averages before and after the change point[107]. In applying the CPR algorithm to the perovskite $(^{87}Sr/^{86}Sr)_i$ and whole-rock $(^{87}Sr/^{86}Sr)_i$, $(^{143}Nd/^{144}Nd)_i$ and $(^{206}Pb/^{204}Pb)_i$ datasets, we identified a prominent change point occurring at 114 Ma for all of the isotopic proxies, except $(^{143}Nd/^{144}Nd)_i$, which occurs between 114 and 100 Ma, and $\epsilon$Nd, which occurs between 118 and 114 Ma (Extended Data Fig. 10).

Finally, to test whether this step change in kimberlite composition can be identified on other continents, we repeated the above procedure using kimberlites from the North American Craton (including Greenland), which contains numerous kimberlite fields (Extended Data Fig. 3a). Here we compiled isotope data for age-constrained kimberlites from the GEOROC geochemistry database (https://georoc.eu/). For binned data (Fig. 4b and Extended Data Fig. 11), $\epsilon$Nd$_i$ and $\epsilon$Sr$_i$ from samples with the same emplacement age were averaged. We find that these data reveal a similar step change from predominantly enriched to predominantly depleted kimberlite compositions. The CPR analysis identifies a step change in both $\epsilon$Nd$_i$ and $\epsilon$Sr$_i$ for these kimberlites between 105 and 102.5 Ma (Extended Data Fig. 11)—very similar in both tempo and magnitude to the observed shift in southern Africa (Extended Data Fig. 10). The geochemistry of these Cretaceous kimberlites in North America has been explained by hydrous decompression melting of an OIB-type mantle source[15].

The step changes in isotope composition of kimberlite at about 114 Ma (South Africa) and about 105 Ma (North America) are compatible with initial melting of metasomatized lithospheric mantle before, and during, detachment of the cratonic mantle keel, as predicted by our geodynamic models (Fig. 2 and Supplementary Video 1). This is then

followed by eruption of melts exhibiting a stronger asthenospheric signature, but that nevertheless retain a sufficient carbonate burden to generate diamond-bearing kimberlites.

## Data availability

All data generated and analysed during this study are provided as Source Data files and as Supplementary Datasets 1 and 2, available in the online version of the paper. All associated files and georeferenced data are available from the Zenodo open repository (developed under the European OpenAIRE programme and operated by CERN) at https://doi.org/10.5281/zenodo.7849141. Source data are provided with this paper.

## Code availability

The input file, custom source code and ASPECT installation details for the thermomechanical simulations are available from the Zenodo repository at: https://doi.org/10.5281/zenodo.7825780. The software, calculation and plotting scripts for the decompressional hydrous melting calculations in Fig. 3c are freely available at https://github.com/smj75/mormel. The input files, output files and source code for the kimberlite tectonic analysis are available at https://doi.org/10.5281/zenodo.7849141.

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

**Acknowledgements** T.M.G. and T.K.H. were supported by The Alan Turing Institute under the EPSRC grant EP/N510129/1. T.M.G. gratefully acknowledges funding from the WoodNext Foundation, a component fund administered by the Greater Houston Community Foundation. T.M.G. and R.M.P. received support from the Web Science Institute Stimulus Fund. A.S.M. was supported by the MCSA Fellowship NEOEARTH, project 893615. W.L.G. and S.Y.O. acknowledge funds from Australian Research Council grant CE110001017 and AuScope NCRIS. This is contribution 1736 from the ARC Centre of Excellence for Core to Crust Fluid Systems (http://www.ccfs.mq.edu.au) and 1505 in the GEMOC ARC National Key Centre (http://www.gemoc.mq.edu.au). The authors gratefully acknowledge the computing time granted by the Resource Allocation Board and provided on the supercomputer Lise at NHR@ZIB as part of the NHR infrastructure. The calculations for this research were conducted with computing resources under the project bbp00039. We thank J. VanDecar for his invaluable editorial support. We are

very grateful to S. Tappe for providing the kimberlite database from ref. 6. We also appreciate helpful discussions with R. Huismans, S. Sparks, Z. Pintér, R. N. Mitchell and L. T. de Oliveira. T.M.G. would like to acknowledge the kimberlite research community for many lively and thought-provoking interactions over the past two decades. T.M.G. also wishes to acknowledge and pay tribute to the late M. de Wit, whose visionary and inspiring leadership on geodynamics and the evolution of the Gondwana supercontinent has left an indelible mark on this field. Maarten's contributions will always be remembered and appreciated.

**Author contributions** T.M.G. conceived the idea, interpreted data and prepared the manuscript. S.M.J. conceived the analytical model of Rayleigh–Taylor instability and performed geotherm and melting calculations. S.B. developed the thermomechanical simulations, with input from A.G., who contributed expertise in ASPECT modelling. T.K.H. carried out statistical and geospatial analysis (R, QGIS, Uninet) and modelling, with input from T.M.G. D.K. and R.M.P. contributed geotectonic interpretations and A.S.M. provided support with plate tectonic modelling software, GPlates and pyGPlates (https://www.gplates.org/). M.F. contributed expertise on kimberlites. S.Y.O. and W.L.G. helped analyse and interpret xenolith data and lithospheric processes. J.C.S. contributed expertise in phase diagrams, phase equilibria and thermodynamic data, using the open-source Perple_X software (https://www.perplex.ethz.ch/). M.R.P. and C.J.S. analysed and interpreted geochemical data. T.M.G. and S.M.J. wrote the manuscript, with input from all co-authors.

**Competing interests** The authors declare no competing interests.

## Additional information

**Correspondence and requests for materials** should be addressed to Thomas M. Gernon.

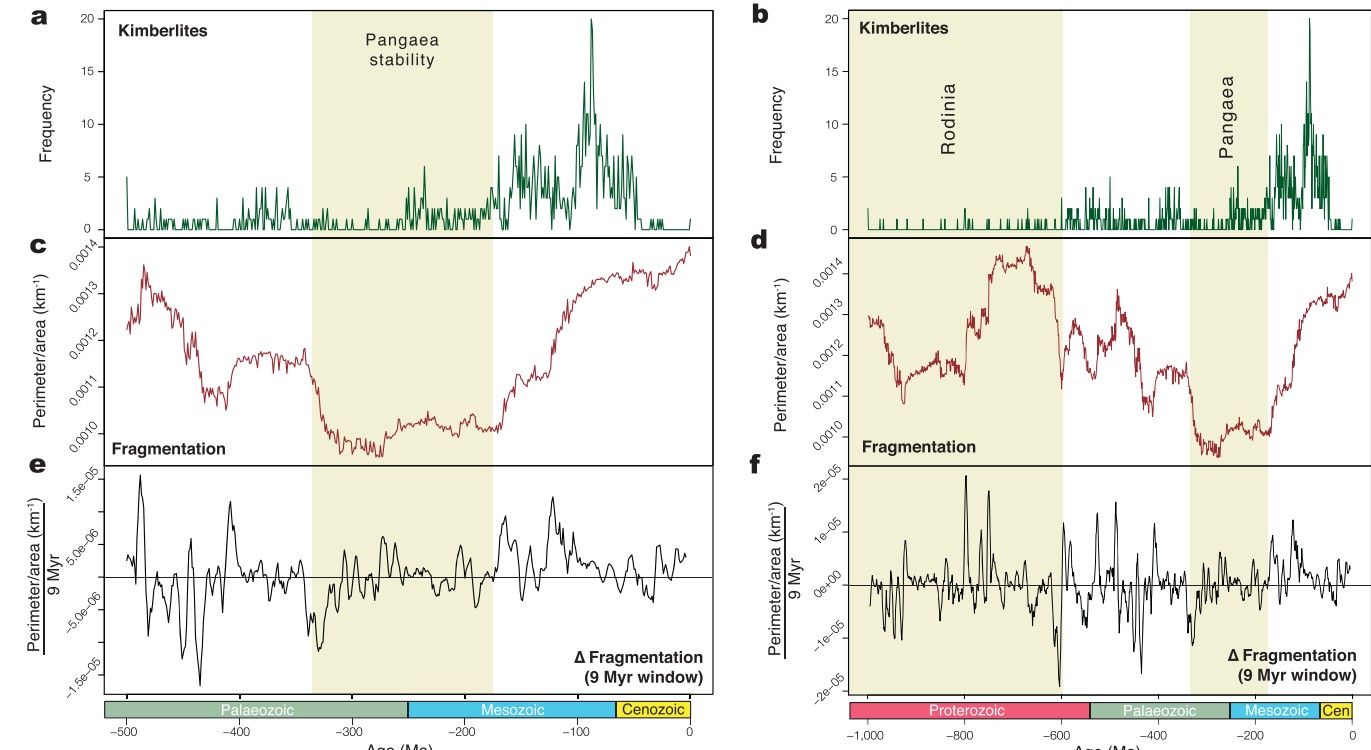

**Extended Data Fig. 1 | Relationship between continental fragmentation and global kimberlites.** Kimberlite distribution since 500 Ma (*n* = 860) (**a**) and since 1 Ga (*n* = 981) (**b**), using radiometrically dated kimberlites from the compilation of ref. 6 (Methods). Continental fragmentation (continental perimeter/area) derived from palaeogeographic reconstructions of ref. 17 for 500–0 Ma (**c**) and 1,000–0 Ma (**d**). Rate of change of continental fragmentation (Δ*F*; see Methods) using a 9-Myr window for 500–0 Ma (**e**) and 1,000–0 Ma (**f**). Data are available in Supplementary Dataset 1.

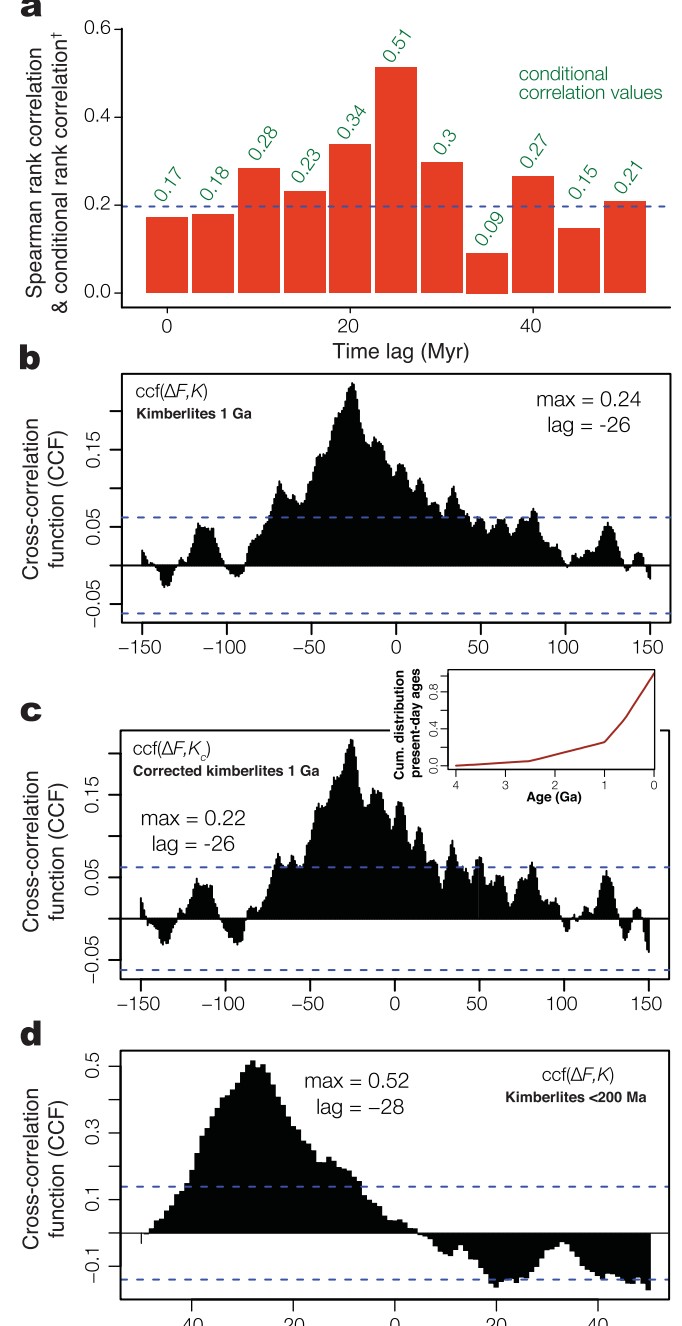

**Extended Data Fig. 2 | Relationship between fragmentation and kimberlites. a**, Spearman rank correlation and conditional rank correlation (†refer to equation (2) for definitions) for $\Delta F$ (slope over 9-Myr moving window) and kimberlites for 500–0 Ma, calculated using a Bayesian network[18] (Methods). Here the input is a 5-Myr-resolution series, in which kimberlite count is the total number of events in each 5-Myr interval and $\Delta F$ is the slope of the regression line for fragmentation estimated every 5 Myr. Using a simple saturated Bayesian network (in which each node is linked by an arc to every other node in the network), we computed the correlation of $\Delta F$ and kimberlite count corr($\Delta F, K$); then the correlation of $\Delta F$ and kimberlite with a lag of 5 Myr (in which $\Delta F$ precedes kimberlites) conditional on $\Delta F$ (unlagged), that is, corr($\Delta F_{t-5}, K|\Delta F$); then the correlation at lag 10 Myr, conditional on the lags at 0 and 5 Myr corr($\Delta F_{t-10}, K|\Delta F, \Delta F_{t-5}$) etc., up to a lag of 50 Myr. This removes the effect of shorter lags and thus the effects of autocorrelation. This test confirms that the maximum correlation between $\Delta F$ and kimberlites occurs roughly 25 Myr after fragmentation (with uncertainty of ±4 Myr). **b**, Cross-correlations between kimberlites[6] ($n = 981$) and $\Delta F$ (9-Myr window) spanning a billion years (Methods), showing dominant lags at −26 ± 4 Myr (that is, fragmentation preceding kimberlites); dashed blue lines show 95% confidence intervals. **c**, Cross-correlations between kimberlites and $\Delta F$ accounting for potential preservation bias by weighting the number of kimberlites inversely according to surface preservation (inset, from ref. 108). This analysis does not change the dominant lag (−26 Myr) relative to **b**. **d**, Cross-correlations between kimberlites[6] ($n = 665$) and $\Delta F$ (9-Myr window) from 200 to 0 Ma (Methods), showing the strongest correlation ($\rho = 0.52$) at a lag of −28 Myr; dashed blue lines show the 95% confidence intervals. Note the different scale on the $x$ axis relative to **a–c**.

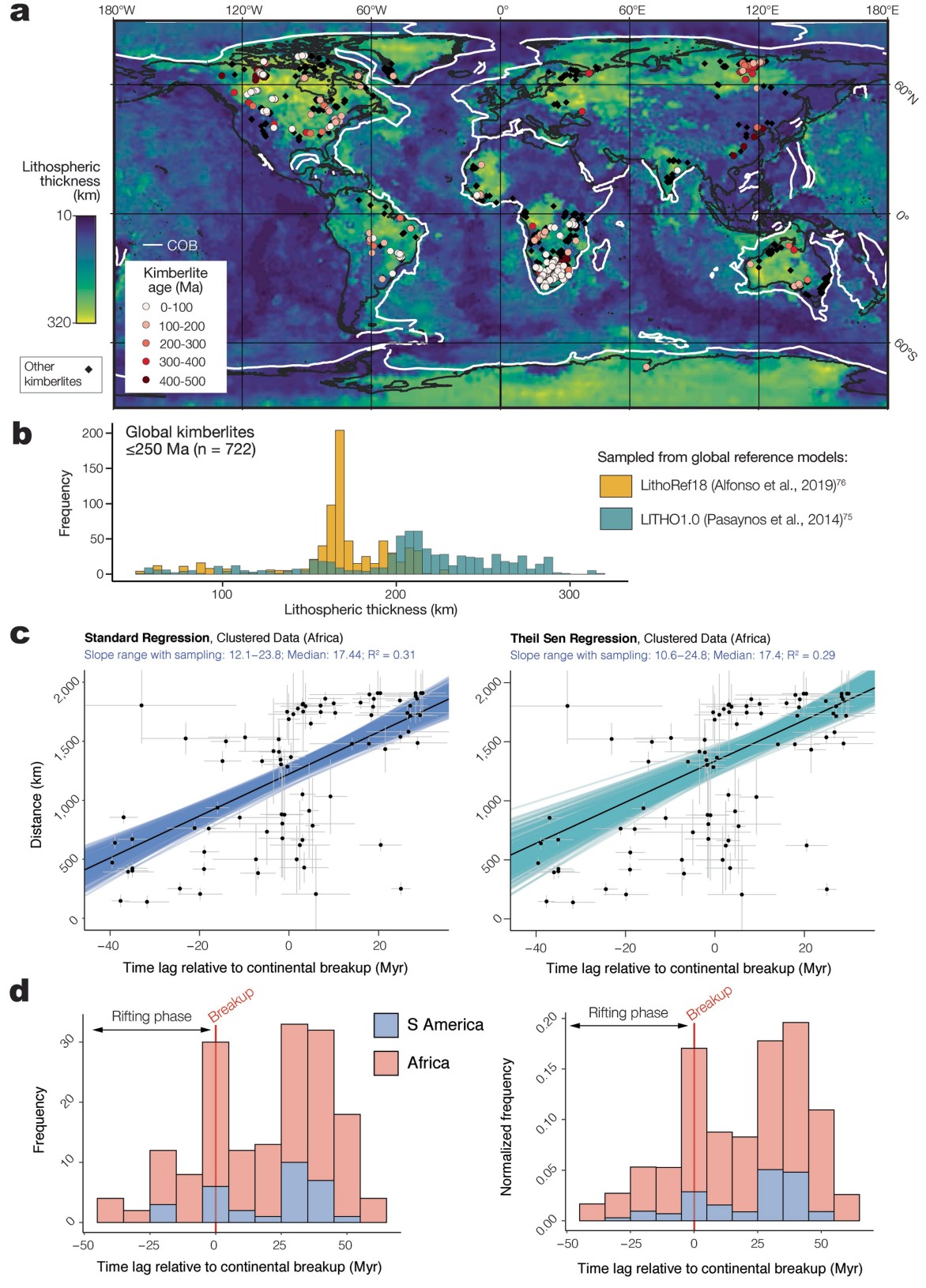

**Extended Data Fig. 3** | See next page for caption.

**Extended Data Fig. 3 | Kimberlite distributions, lithospheric thickness and migration characteristics. a**, Global map of kimberlites younger than 500 Ma from ref. 6 ($n = 860$) plotted on a map of lithospheric thickness interpolated using data from ref. 75 (COBs shown in white are from GPlates[23,69]). Kimberlites are coloured by their radiometric ages[6]; the global kimberlites ($n = 4,287$, black diamonds) are from ref. 58. **b**, Lithospheric thickness, sampled from two different global reference models[75,76], for kimberlites younger than 250 Ma; note that kimberlites predominantly occur on lithosphere >150 km thick. **c**, Estimation of migration rate (slope) using distance and time lag from continental breakup for African kimberlite clusters, using standard linear regression and Theil–Sen regression. A Monte Carlo simulation (5,000 runs) was performed to capture uncertainty in kimberlite age, breakup age and rift distance (Methods). Black circles denote the median distance and lag from all 5,000 simulations for each kimberlite cluster and bars denote the standard deviation. Regression lines for each individual simulation are shown in blue and the black line shows the regression for the original dataset. Results are broadly similar for both the standard and Theil–Sen regression models, and the estimated migration rate is consistent with analytical and geodynamic models (Fig. 3a). **d**, Histograms showing the distribution of lags (time in millions of years relative to breakup) for kimberlite clusters in Africa and South America. The first histogram uses the original dataset and the second (normalized) histogram incorporates age and distance uncertainties by Monte Carlo simulation. Note the peaks around breakup and approximately 25–55 Myr post-breakup.

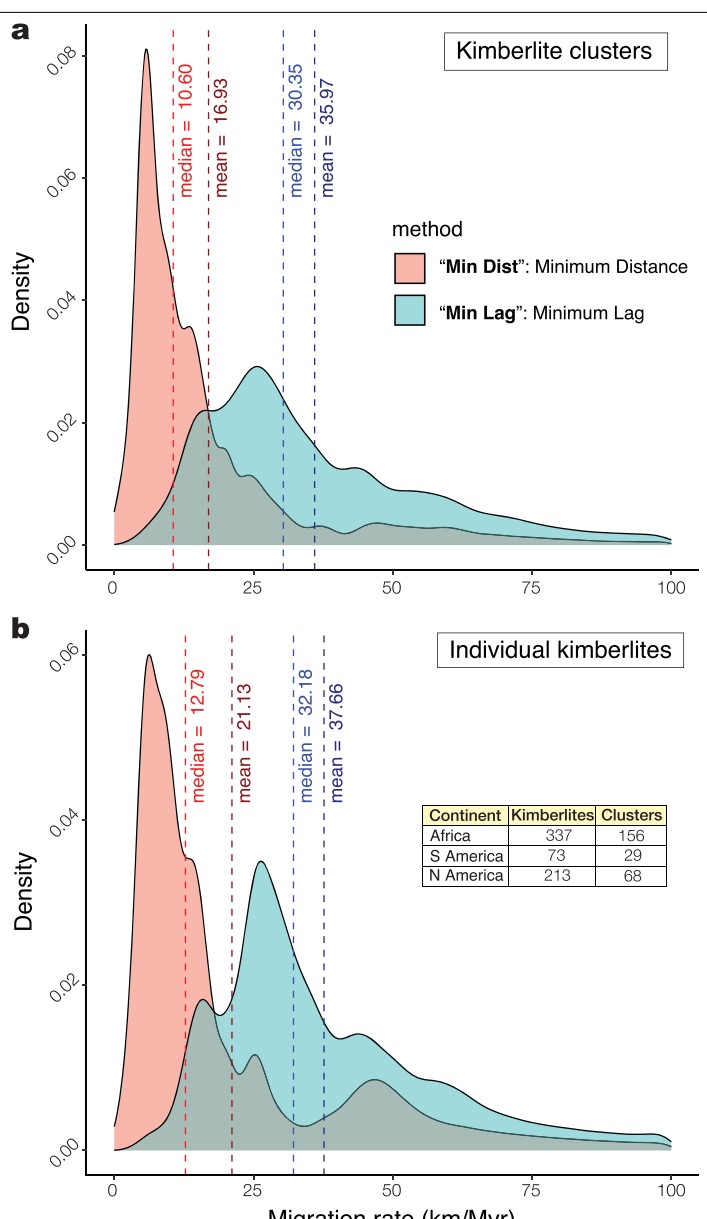

**Extended Data Fig. 4 | Minimum distances and time lags of kimberlite eruptions relative to continental boundaries.** Probability density plots for two endmember rift association models, associating kimberlites with the rift section with either the minimum lag (blue) or the minimum distance (red). This provides an estimate of the upper and lower bounds of the migration rate, for each kimberlite or cluster. Uncertainty in kimberlite age, breakup age and rift distance is accounted for by Monte Carlo simulation (see Methods). Results are presented for all three locations (North America, South America and Africa) combined. **a**, The density plot for kimberlite clusters, giving a median of 10.6 (shortest lag) to 30.4 (shortest distance); data are available in Supplementary Dataset 2. **b**, The density plot for individual kimberlites, giving medians of 12.8 (shortest lag) and 37.7 (shortest distance). Inset table summarizes the number of kimberlites and clusters by continent.

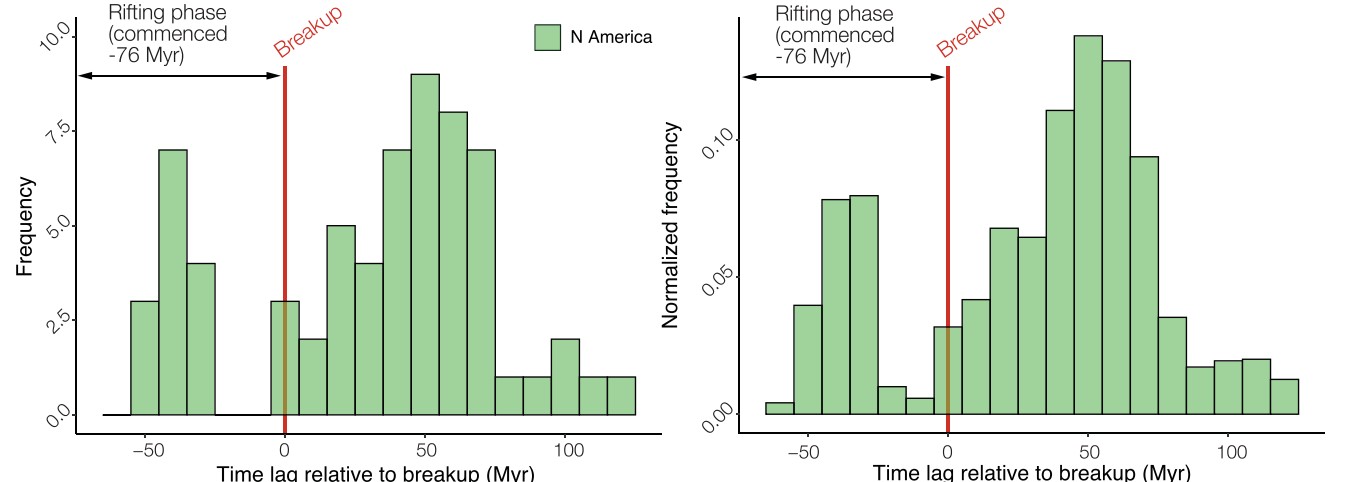

**Extended Data Fig. 5 | Temporal distribution of kimberlites in North America relative to continental breakup.** Histograms showing the distribution of lags (time in millions of years relative to breakup) for kimberlite clusters in North America, focusing on kimberlites closest to rift sections, using the original dataset (no uncertainty) (left) and shown as a normalized histogram incorporating uncertainty in kimberlite age, breakup age and rift distance (right). Using the clustering method (see Methods for details), we obtain a total of 65 clusters. Peaks occur around 55–25 Myr before breakup and 35–75 Myr post-breakup. Breakup in North America (that is, rifts of the Central Atlantic, Greenland–North America and Arctic region) is diachronous and estimated to occur between 10 and 76 Myr after rift onset[24].

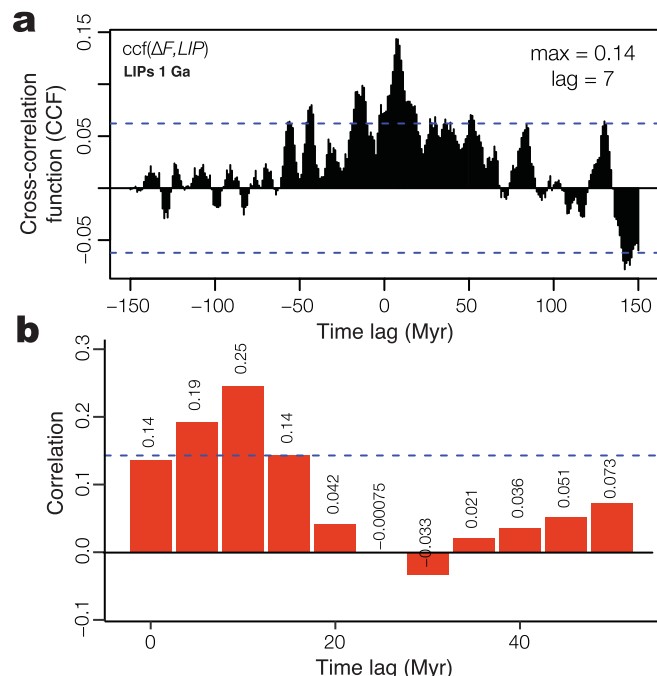

**a**

**b**

**Extended Data Fig. 6 | Relationship between continental breakup and plumes over 1 Ga. a**, Cross-correlations between $\Delta F$ (9-Myr window) and plumes over 1 Ga, using the well-established ages of their surface expression, LIPs[25] ($n = 104$). This analysis shows a strong peak at +7 ± 4 Myr lags, indicating that LIP magmatism most commonly initiates about 7 Myr before continental fragmentation. **b**, Results of a Bayesian network investigating the link between LIPs and $\Delta F$, and configured for LIPs leading $\Delta F$ (as shown in **a** to be dominant). The input is a 5-Myr-resolution series, in which LIP is the total number of LIP events with a start date falling in each 5-Myr interval and $\Delta F$ is the slope of the regression line for fragmentation (over a 9-Myr window) estimated every 5 Myr. Critically, this analysis removes the effect of shorter lags and thus the effects of autocorrelation. The maximum conditional correlation (0.25) occurs at a lag of approximately 10 ± 4 Myr (in which LIP leads $\Delta F$); dashed blue line show estimated 95% confidence intervals (threshold for the 95% confidence interval = 0.143 for the 5-Myr-resolution time series of length $n = 188$).

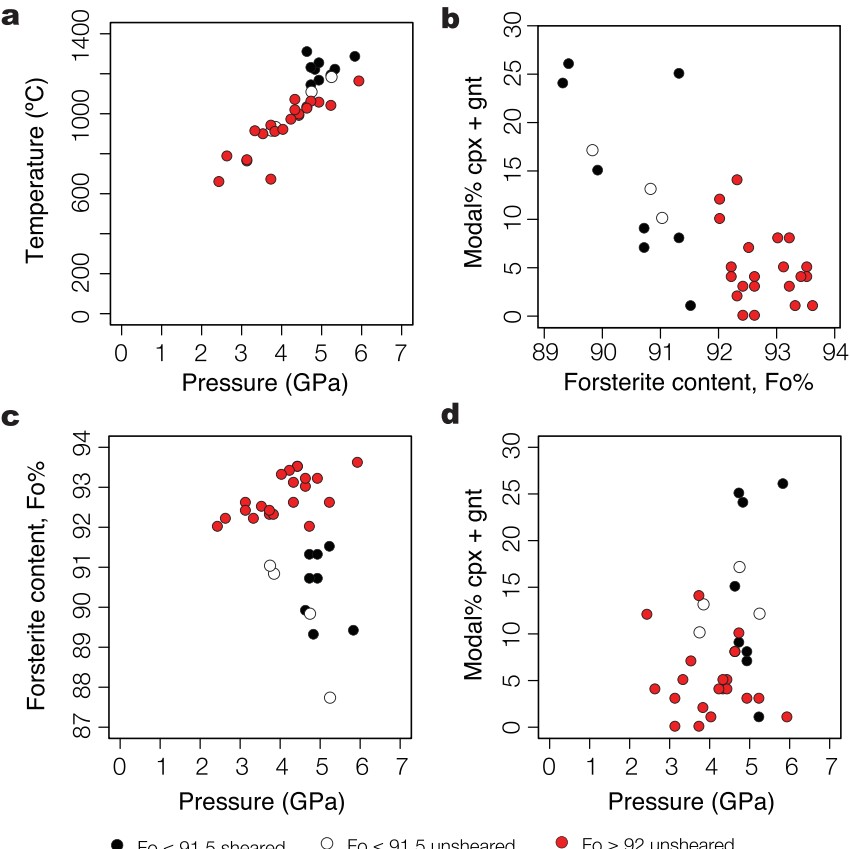

● Fo ≤ 91.5 sheared    ○ Fo ≤ 91.5 unsheared    ● Fo ≥ 92 unsheared

**Extended Data Fig. 7 | Compositional and *P–T* characteristics of Southern African xenoliths, showing the effects of refertilization. a**, Pressure versus temperature estimates (from thermometry/barometry) of peridotite xenoliths from the Kaapvaal Craton (data from various sources summarized in ref. 98); note that most low-Fo xenoliths lie above the geotherm defined by high-Fo xenoliths and are sheared, providing good evidence of a thermal effect.

**b**, Olivine forsterite content (Fo%) versus modal % clinopyroxene (cpx) + garnet (gnt) (as a measure of fertility); note the negative correlation of Fo with fertility. **c**, Pressure versus Fo; note the concentration of low-Fo xenoliths between 4.5 and 5.2 GPa (approximately 22 km thick layer), interpreted to represent a dense boundary layer. **d**, Pressure versus modal % cpx + gnt; note the general high fertility of low-Fo xenoliths, with two-thirds containing cpx + gnt >10.

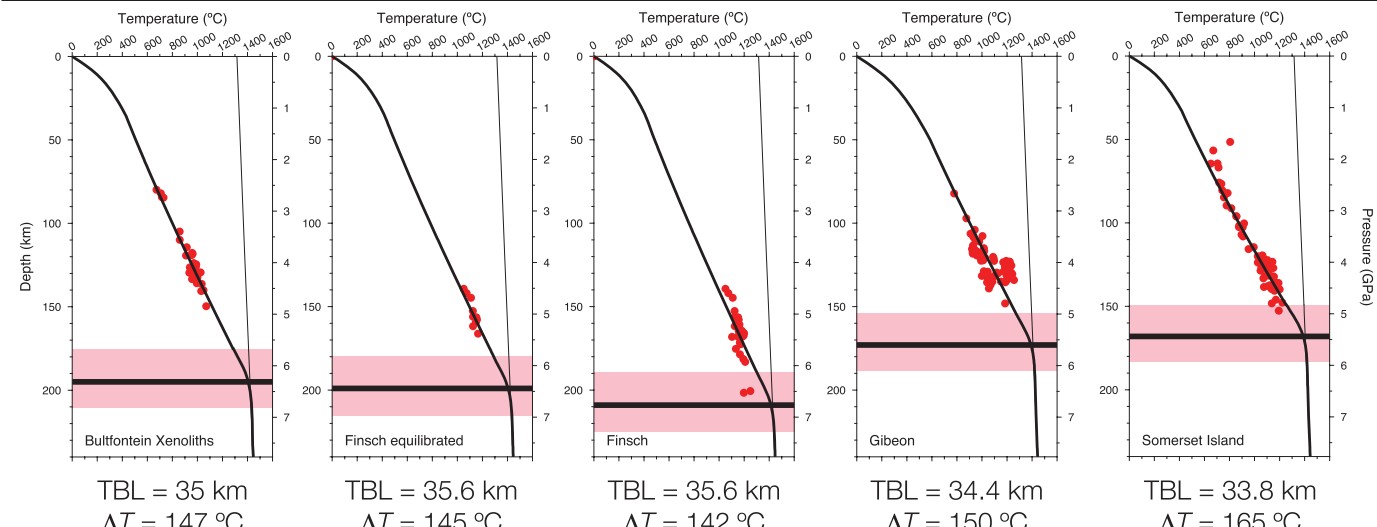

**Extended Data Fig. 8 | Thermal-boundary-layer properties derived from xenolith geotherms.** Thickness and temperature conditions of the lower lithospheric thermal boundary layer (TBL) derived using peridotite xenolith–$P$–$T$-based geotherms of Mather et al.[77] for four different kimberlites: Bultfontein, Finsch, Gibeon and Somerset Island (see Methods). Note that the TBL is consistently around 35 km thick and the temperature increase across it, $\Delta T$, is about 150 °C.

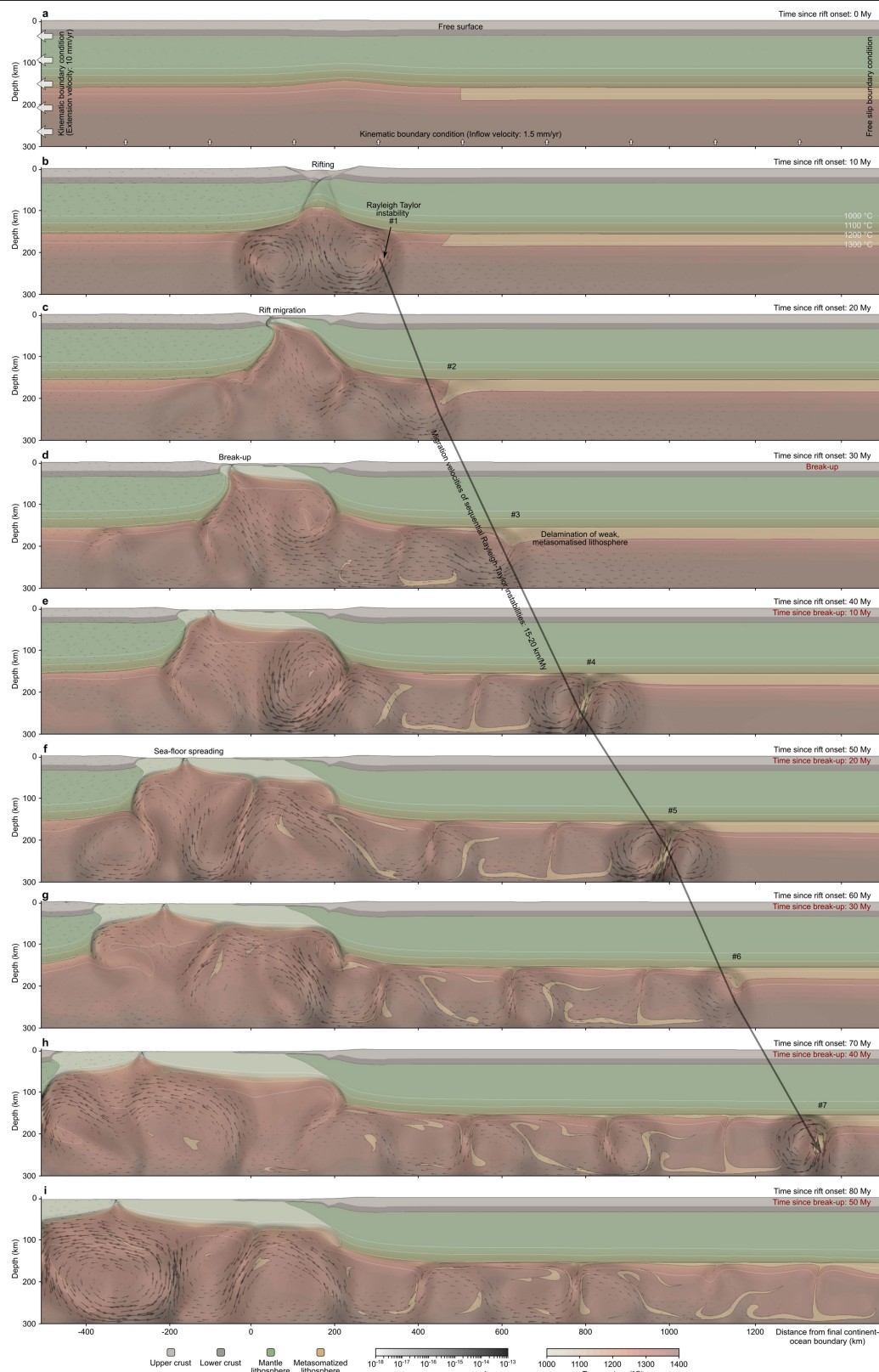

**Extended Data Fig. 9 | Thermomechanical simulations of continental breakup. a–i,** Generation and propagation/migration of sequential Rayleigh–Taylor instabilities (labelled 1–7 at different time slices), which migrate at velocities of 15–20 km Myr⁻¹.

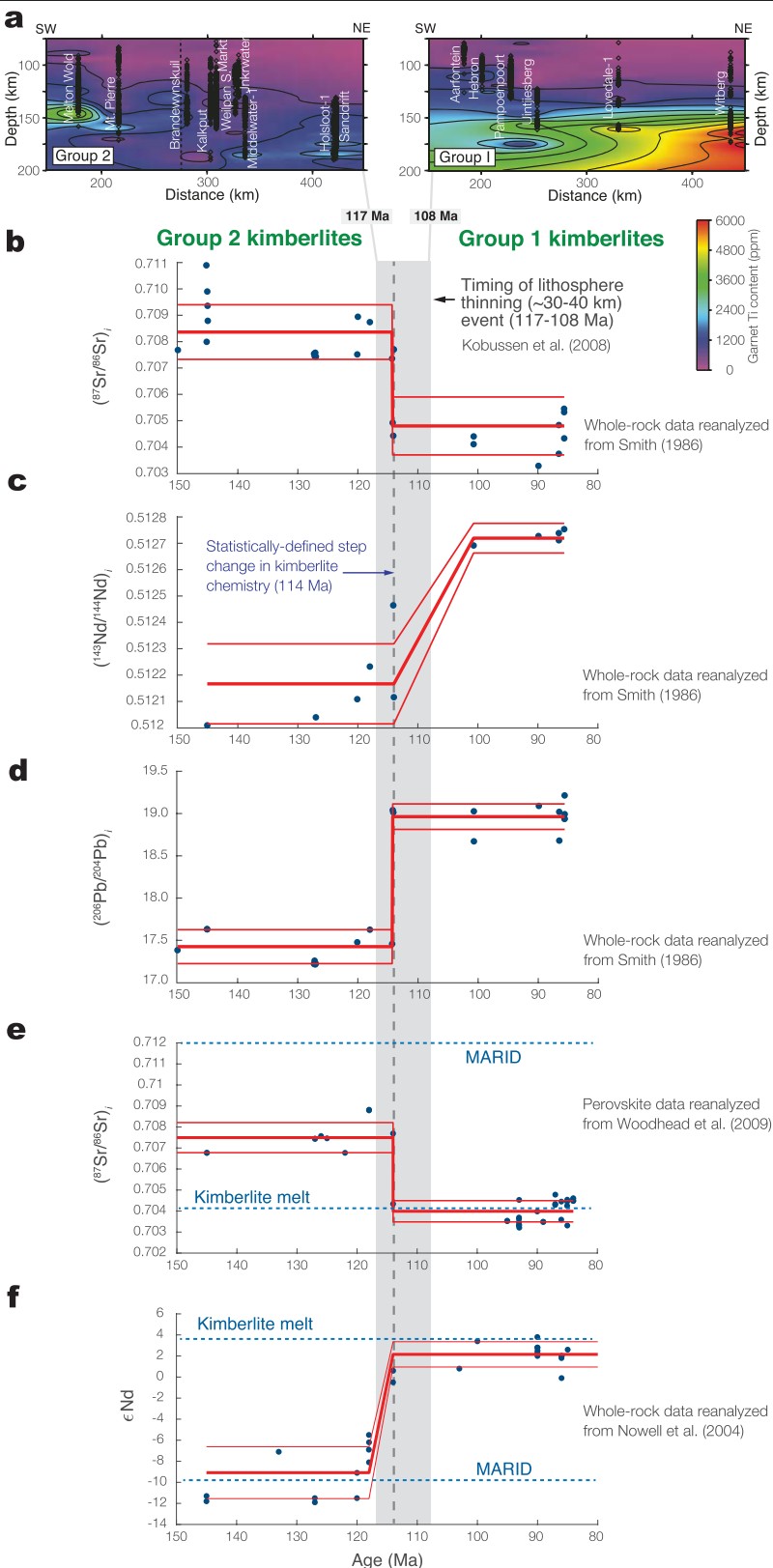

**Extended Data Fig. 10 |** See next page for caption.

**Extended Data Fig. 10 | Changing chemistry of kimberlites in the Kaapvaal Craton from 150 to 85 Ma. a**, Interpolated Ti contents of garnet xenocrysts (modified after ref. 40) at 117 and 108 Ma, showing the effects of heating and chemical refertilization of the lower lithosphere by asthenospheric melts, thinning the lithosphere by 30–40 km (vertical grey field). Below this are the chemical compositions of group II kimberlites (orangeites/lamproites) and group I kimberlites, specifically whole-rock $(^{87}Sr/^{86}Sr)_i$ (**b**), whole-rock $(^{143}Nd/^{144}Nd)_i$ (**c**) and whole-rock $(^{206}Pb/^{204}Pb)_i$ (**d**); these data are revised from Smith[50]. **e**, $(^{87}Sr/^{86}Sr)_i$ of kimberlitic perovskites from Woodhead et al.[103]. The plot shows the MARID endmember defined from kimberlite xenoliths and thought to derive from a lithospheric mantle source[105], and a kimberlite melt endmember[105] largely defined from analyses of PIC kimberlite xenoliths. **f**, Whole-rock $\epsilon Nd$ calculated from the data of Nowell et al.[104]. The lines on the plots show the statistically defined change points (using CPR; Methods) and two-sigma uncertainty bounds of the two averages (thin red lines) before and after the change point. Step changes occur at 114 Ma (dashed vertical line) for all variables, except $(^{143}Nd/^{144}Nd)_i$, which occurs between 114 and 100 Ma, and $\epsilon Nd$, which occurs between 118 and 114 Ma. Continent-scale metasomatism occurred before 114 Ma (ref. 109), raising the possibility that migrating chains of convective instabilities (Fig. 2) partially stripped and melted the lithospheric keels, driving infiltration (that is, melt metasomatism) of carbonate melts that caused further destabilization.

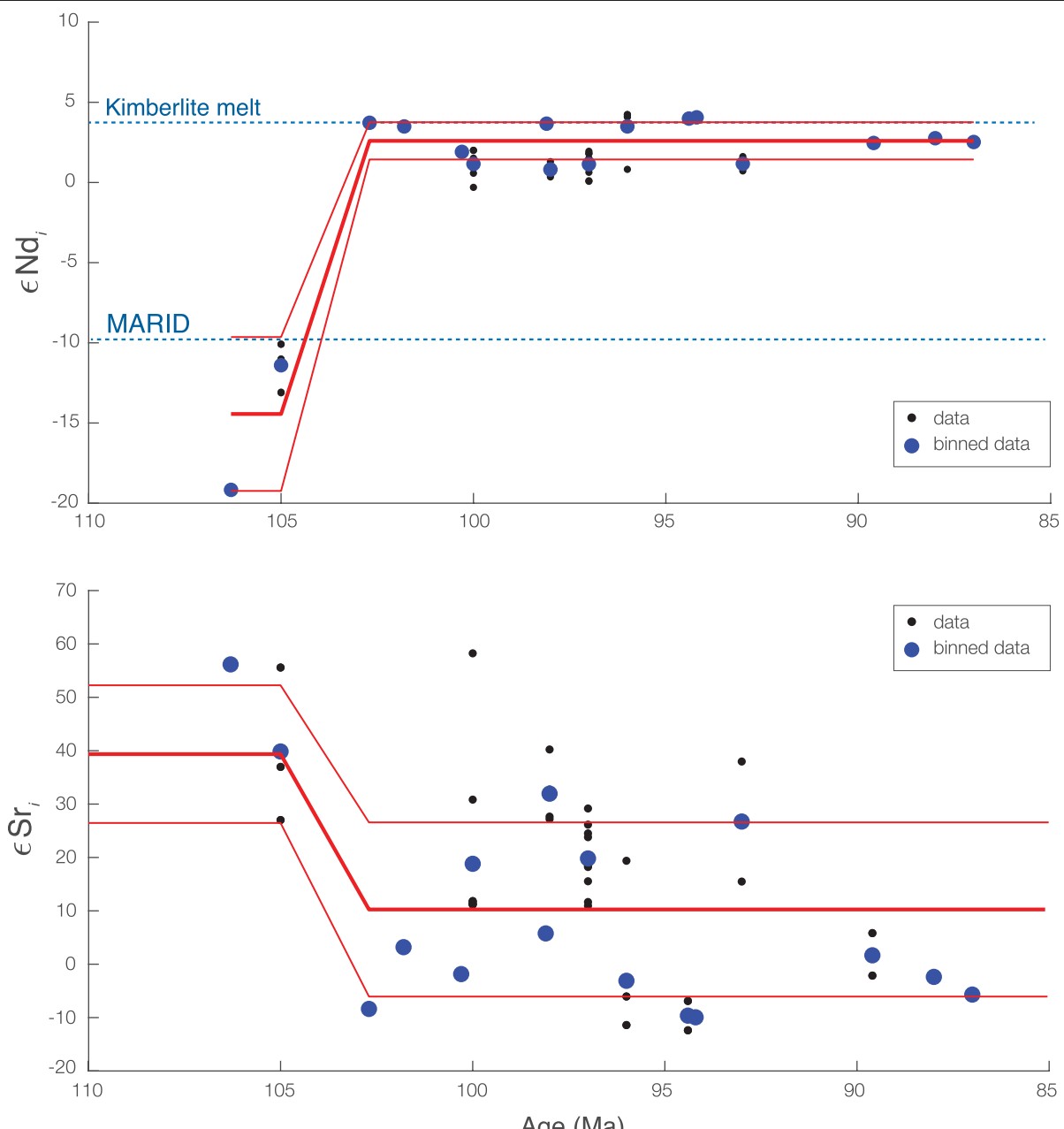

**Extended Data Fig. 11 | Changing chemistry of kimberlites in North America from 110 to 85 Ma.** Whole-rock kimberlite Nd and Sr isotope compositions in North America from the GEOROC repository. Note the abrupt shift from lithospheric ($\epsilon$Nd$_i$ −10 to −20) to mantle ($\epsilon$Nd$_i$ 0 to +5) kimberlite compositions, which are also reflected in a reduction of $\epsilon$Sr$_i$. This transition is nearly identical to that seen in Southern Africa (Extended Data Fig. 10) and is interpreted in similar terms to represent an early peak in lithospheric removal followed by upwelling of asthenospheric mantle.

**Extended Data Table 1 | Rayleigh–Taylor instability models applied to lithospheric keel delamination**

| Key | Model | Scaled Wavelength | Scaled Growth Rate | Instability Wavelength | Instability Growth Time | Lateral Propagation | Upwelling Rate | Péclet Number | Reference |
|---|---|---|---|---|---|---|---|---|---|
| | | $\lambda_d^*$ | $q_d^*$ | $\lambda_d$ | $\tau_d$ | $U$ | $V$ | $Pe$ | |
| | | | | km | Myr | km/Myr | km/Myr | | |
| 1 | Layer over half-space of same viscosity | 3.7 | 0.097 | 65 | 2.6 | 25 | 39 | 22 | 33 |
| 2 | Layer over layer of same thickness and viscosity | 2.6 | 0.077 | 45 | 3.3 | 14 | 19 | 11 | 88 |
| 3 | Layer over much less viscous half-space | 3.0 | 0.160 | 53 | 1.6 | 33 | 26 | 14 | 33 |
| 4 | Layer with linearly decreasing density over half-space of constant density and same viscosity | 2.9 | 0.037 | 51 | 3.4 | 15 | 24 | 13 | 33 |
| 5 | Layer with linearly decreasing density over half-space of constant density and much lower viscosity | 2.4 | 0.059 | 42 | 2.1 | 20 | 17 | 9 | 33 |
| 6 | Plastic layer over plastic half-space | 2.8 | 0.610 | 49 | 0.4 | 118 | 23 | 13 | 33 |

Six analytical models (see Fig. 3a) describe two fluid layers in which the upper layer has the higher density (Methods). The models differ in the relative layer thicknesses, viscosity and vertical density gradient. Each model is specified by a scaled dominant wavelength $\lambda_d^*$ and a scaled exponential growth rate $q_d^*$ for convective instabilities. Actual wavelengths and e-folding growth times (determined using equations (5) and (6)) are shown for the typical lithospheric thermal-boundary-layer properties we estimated by fitting geotherms to $P$–$T$ data from kimberlite xenoliths (see the section 'Xenolith geotherms' in Methods). The lateral propagation rate for a chain of instabilities is determined using equation (1), which we derived from scaling analysis (see the section 'Scaling analysis of Rayleigh–Taylor instability chains' in Methods). The characteristic rate and thermal Péclet number for vertical return flow confirms that asthenosphere will well up adiabatically to replace the removed part of cratonic keel (see the section 'Melting calculations' in Methods).

**Extended Data Table 2 | Geodynamic model parameters**

| Parameter | Symbol | Units | Upper crust | Lower crust | Lithospheric mantle | Asthenospheric mantle & thermal boundary layer |
|---|---|---|---|---|---|---|
| Reference density (at surface conditions) | $\rho_0$ | kg m$^{-3}$ | 2,700 | 2,850 | 3,280 | 3,300 |
| Thermal expansivity | $\alpha$ | K$^{-1}$ | $2.7 \cdot 10^{-5}$ | $2.7 \cdot 10^{-5}$ | $3.0 \cdot 10^{-5}$ | $3.0 \cdot 10^{-5}$ |
| Thermal diffusivity | $\kappa$ | m$^2$ s$^{-1}$ | $7.72 \cdot 10^{-7}$ | $7.31 \cdot 10^{-7}$ | $8.38 \cdot 10^{-7}$ | $8.33 \cdot 10^{-7}$ |
| Heat capacity | $C_p$ | J kg$^{-1}$ K$^{-1}$ | 1,200 | 1,200 | 1,200 | 1,200 |
| Heat production | $H$ | W m$^{-3}$ | $1.0 \cdot 10^{-6}$ | $0.1 \cdot 10^{-6}$ | 0 | 0 |
| | | | | | | |
| Cohesion | $C$ | Pa | $5 \cdot 10^6$ | $5 \cdot 10^6$ | $5 \cdot 10^6$ | $5 \cdot 10^6$ |
| Internal friction coefficient (unweakened) | $f$ | - | 0.5 | 0.5 | 0.5 | 0.5 |
| Strain weakening interval | - | - | [0,1] | [0,1] | [0,1] | [0,1] |
| Frictional weakening factor | $a_f$ | - | 0.25 | 0.25 | 0.25 | 0.25 |
| Viscous weakening factor | $a_v$ | - | 0.25 | 0.25 | 0.25 | 1.0 |
| Material | | | Wet quartzite[84] | Wet anorthite[85] | Dry olivine[53] | Wet olivine[53] |
| Stress exponent (dis) | $n$ | - | 4.0 | 3.0 | 3.5 | 3.5 |
| Prefactor (dis) | $A_{dis}$ | Pa$^{-n}$ s$^{-1}$ | $8.57 \cdot 10^{-28}$ | $7.13 \cdot 10^{-18}$ | $6.52 \cdot 10^{-16}$ | $2.12 \cdot 10^{-15}$ |
| Activation energy (dis) | $E_{dis}$ | J mol$^{-1}$ | $223 \cdot 10^3$ | $345 \cdot 10^3$ | $530 \cdot 10^3$ | $480 \cdot 10^3$ |
| Activation volume (dis) | $V_{dis}$ | m$^3$ mol$^{-1}$ | 0 | $38 \cdot 10^{-6}$ | $18 \cdot 10^{-6}$ | $11 \cdot 10^{-6}$ |
| Prefactor (diff) | $A_{diff}$ | Pa$^{-1}$ s$^{-1}$ | $5.97 \cdot 10^{-19}$ | $2.99 \cdot 10^{-25}$ | $2.25 \cdot 10^{-9}$ | $1.5 \cdot 10^{-9}$ |
| Activation energy (diff) | $E_{diff}$ | J mol$^{-1}$ | $223 \cdot 10^3$ | $159 \cdot 10^3$ | $375 \cdot 10^3$ | $335 \cdot 10^3$ |
| Activation volume (diff) | $V_{diff}$ | m$^3$ mol$^{-1}$ | 0 | $38 \cdot 10^{-6}$ | $6 \cdot 10^{-6}$ | $4 \cdot 10^{-6}$ |
| Grain size (diff) | $d$ | m | 0.001 | 0.001 | 0.001 | 0.001 |
| Grain size exponent (diff) | $m$ | - | 2.0 | 3.0 | 0 | 0 |

Model parameters used in ASPECT thermomechanical simulations (see Methods for details). diff, diffusion creep; dis, dislocation creep (creep property values from refs. 53,84,85).