## [Peer Review File · Nature]

Manuscript Title: Rift-induced disruption of cratonic keels drives kimberlite volcanism

Reviewer Comments & Author Rebuttals

Reviewer Reports on the Initial Version:

Referee #1:

This manuscript proposes a new conceptual model for kimberlite formation, supported by geochemical and geodynamical modeling analyses. In the model, kimberlitic magmas are formed as a result of melting that occurs below cratonic lithosphere as rifting removes the lowermost cratonic lithosphere through formation of instabilities that migrate inward the craton. Thus, kimberlites are closely related to plate tectonics. Statistical analysis suggests that continental breakup typically leads kimberlite magmatism by several tens of myrs (~25 Myr), but the lead time can be much longer, and kimberlites may also precede breakup.

I applaud the team's efforts to research whether kimberlitic magmas are related to plate tectonics. I like the research approach presented here: determining whether there is a statistical spatial/temporal relation between continental rupture and kimberlitic magmatism, and exploring a possible mechanism. At present, the manuscript has not been able to convince me that kimberlitic magmatism is related to continental breakup through the proposed mechanism of convective instabilities. Points that need attention are 1) that the mechanism presented here would only explain a portion of the kimberlitic magmatism, 2) that the ASPECT modeling results suggest that the volcanism would not occur at a particular point in time and space, but rather would linger over longer temporal (tens of millions of years) and spatial (800-1000 km) scales, 3) that the convective instabilities should not be directly related to the time of continental breakup because they can be initiated early on in the rifting process and the duration of rifting before breakup occurs varies, and 4) that the thickness of cratons varies strongly (also within one craton): if kimberlites are related to thin(ned)-craton locations, then a more general "craton rejuvenation event mechanism" model would better explain kimberlite occurrences than the model proposed here.

The model presented in this study does not explain kimberlites in other locations (for example, those not related to rifting, or far away from the breakup axis), or kimberlites that form at times not associated with continental breakup or plate tectonics (i.e., kimberlites lagging more than 30-50 Myr, or kimberlite ages preceding continental breakup with tens (or more) of millions of years). Based on this manuscript, I consider the rift-instability model presented as a possible mechanism for craton rejuvenation (and thinning) amongst a series of other mechanisms for craton rejuvenation.

1) Small-scale convection starts as soon as there is a significant lateral thermal gradient between the rift and surrounding regions as described in the manuscript (Buck, 1986, Boutilier and Keen, 1999, Gao and all, Baikal rift, 2003, Wilson et al. 2005 Rio Grande rift, Van Wijk et al. Rio Grande rift 2008, Simon, Huisman et al 2009), when the mantle lithosphere has been thinned below the rift zone. This may occur during the continental rift phase, as illustrated by the rift animation in the

supplemental material. This point in time may precede, however, continental breakup several millions of years, but also by many tens (or even more) millions of years; small scale convective instabilities may also be associated with rifts that never break up (see Buck, 1986, Van Wijk et al., 2008, Gao et al., 2003). What does this mean for the suggested relation between continental breakup and kimberlite ages/occurrences?

2) The geodynamic model predicts instabilities (and thus kimberlites) on both sides of the ocean basin. Africa is discussed in the manuscript; is this observed as well in South America?

3) Craton bases are not at a constant depth- dozens of seismic studies indicate that the base of a craton varies between locations, also within one craton. These variations in thickness may be up to tens of kilometers, or, according to some studies, maybe even as large as 100 km across a craton. Such thickness variations are of similar size (or larger) than those predicted by the rifting-instability model. It is therefore likely that in some locations, the base of the craton is at such shallow depths that melting would occur. If removal of the base of the craton would induce formation of kimberlitic magma, we'd expect that there is a relation between the depth of a craton-base and kimberlite occurrence. Does this relation exist? Would then a craton-rejuvenation model describe the occurrence of kimberlitic magmatism better, and the rift-instability model would be considered a rejuvenation mechanism?

4) I would suggest removing the mantle plume discussion from the manuscript because 1) it introduces another layer of uncertainty and 2) this discussion is, in my opinion, not crucial to the manuscript. I understand why the authors included the discussion, but the problem is that there are more conceivable explanations for kimberlites than only plumes or rifts. So, it is not so that if the plume-relation is disproven, the conclusion is that the rift-model must be correct.

A plume could be viewed as a possible craton-base rejuvenation event, just like the rift-instability model, and several other mechanisms.

Also, the relation between breakup and mantle plumes is more complex than the analysis in this manuscript indicates (and this would be a manuscript by itself). Complexities include the relation between dynamic uplift, onset of magmatism, and onset of rifting (all poorly constrained), plumes that do not cause continental breakup (Emeishan Large igneous province, Afar, and other examples), and a new body of work that suggests alternative explanations for plume breakup magmatism (see for example E. R. Lundin and A. G. Doré, Geological Society, London, Special Publications, 470, 375-392, 21 February 2018, <https://doi.org/10.1144/SP470.6>; and other work by Lundin and Dore).

5) The delamination event discussed in line 251 is puzzling. Is this the convective removal of the craton base (if so, please consider replacing the term delamination with convective removal)? The geodynamic models do not show that ~50 km of material is removed from the base of the craton (this is hard to see though); rather, the layer that is removed seems less than about 30 km. Does this matter? What controls the height of the layer that is removed by the instabilities? When a map is plotted of the depth of the base of cratons, how much variation is there? A quick literature search suggests that the craton base varies strongly, also within one craton. Why would 30-50 km thinning cause kimberlite formation, but other rejuvenation events (or craton thinning events) not?

6) The activation energy is an important parameter in the development of instabilities- whether they

form at all, and if they form, this parameter affects for example the size of the instabilities. See van Hunen, J., Zhong, S., Shapiro, N.M., and Ritzwoller, M.H., 2005, New evidence for dislocation creep from 3-D geodynamic modeling of the Pacific upper mantle structure: *Earth and Planetary Science Letters*, v. 238, p. 146–155, doi: 10.1016/j.epsl.2005.07.006. How do the models change when a different activation energy value is chosen for the mantle?

7) Where and when does ASPECT predict that melting occurs? This is important to know, as the instabilities permanently remove craton mantle, and melting may continue over large spatial and temporal spaces. Why would melting just be an isolated event as suggested by the conceptual figure, and not extend over longer timescales and spatial scales as suggested by the ASPECT modeling? In this respect, the conceptual model seems not supported by the geodynamic modeling results.

8) Related to the former point: The formation of convective instabilities in the asthenosphere is a three-dimensional process. This is nicely shown in for example work by Ballmer and van Hunen, with applications in oceanic and continental domains. Their work suggests that small-scale convective instabilities are not necessarily associated with time-progressive volcanism- age-distance patterns are complex (duration of melting may be prolonged below oceanic lithosphere). How does this apply to continental domains (and in three dimensions)?

Figure 1 caption: please replace rift separation with continent separation

Referee #2:

That is an interesting and truly interdisciplinary paper suggesting indeed new idea about the origin of kimberlites. I appreciate that authors use global database of kimberlite locations and ages, although they focus more on African kimberlites.

This work has several strengths, but also some weaknesses. There are however a number of important questions along with few smaller issues that authors do not answer.

I suggest that authors should answer the questions in the revised manuscript.

1. Is accuracy of reconstructions of timing of continental break-up for the age more than 200 Ma enough high to resolve lag between kimberlite eruptions and break-up of only about 20 - 30 My?

2. If kimberlite eruptions are related to continental break-up, why are they located only in few places along the continental margin and not everywhere where enough thick lithosphere is situated not far away from the margin?

3. While their model looks convincing for South African kimberlites, what about kimberlites of the Siberian craton? In particular, was there continental break-up in eastern Siberia prior to Triassic and Jurassic kimberlites? Some recent publications related to the Siberian craton are overlooked, for example: Ernst R.E. et al. (2016) *Russ. Geol. Geophys.*; Polyansky O.P., et al. (2018) *Russ. Geol. Geophys.*; Sun J. et al. (2018) *Chem. Geol.*

4. Authors convincingly show that mantle convection associated with the continental break-up can destabilize the lowermost layer of lithosphere, but why mantle plume can not do the same? Maybe in fact all processes that can destabilize lower metasomatized layer of continental roots can generate kimberlites including plumes and continental fragmentation?

Here are few smaller issues.

1. Authors use their fragmentation function (ratio between continental perimeter to continental area versus time) citing paper by Meredith et al., 2019. However, this function published in Meredith's paper (Fig. 3a) is significantly different from that used in the manuscript (Extended Data Fig.1d). Please explain.

2. Are correlation numbers on Extended Data Fig.2a, r or r-squared?

Author Rebuttals to Initial Comments:

We thank both Reviewers (R#1 & R#2) for their positive, thoughtful and constructive reviews, which have greatly improved our analysis and the strength and clarity of arguments presented in the paper. We outline our responses to each of their comments below. Where referring to figures included in the response, we use the convention “Fig. R1”, whereas those in the manuscript are referred to as “Fig. 1”. Full references are provided for papers not cited in the manuscript.

Referee #1:

This manuscript proposes a new conceptual model for kimberlite formation, supported by geochemical and geodynamical modeling analyses. In the model, kimberlitic magmas are formed as a result of melting that occurs below cratonic lithosphere as rifting removes the lowermost cratonic lithosphere through formation of instabilities that migrate inward the craton. Thus, kimberlites are closely related to plate tectonics. Statistical analysis suggests that continental breakup typically leads kimberlite magmatism by several tens of myrs (~25 Myr), but the lead time can be much longer, and kimberlites may also precede breakup.

I applaud the team’s efforts to research whether kimberlitic magmas are related to plate tectonics. I like the research approach presented here: determining whether there is a statistical spatial/temporal relation between continental rupture and kimberlitic magmatism, and exploring a possible mechanism. At present, the manuscript has not been able to convince me that kimberlitic magmatism is related to continental breakup through the proposed mechanism of convective instabilities.

Points that need attention are 1) that the mechanism presented here would only explain a portion of the kimberlitic magmatism

We are pleased that the reviewer likes the approach taken in our study. We also appreciate their point that some additional details are required to make our conceptual model more robust and compelling. To address this important point, we have undertaken additional analysis and made multiple changes throughout the manuscript, Methods and Extended Data, which we feel should hopefully convince the reviewer more fully of the viability of our model.

We agree that the mechanism described in our paper cannot explain all kimberlite occurrences. However, using our updated analysis (described below) it can feasibly explain the vast majority (approximately 90%) of kimberlite occurrences, at least over the past 500 million years. Of course, we cannot rule out the occurrence of kimberlite magmatism due to localized thermomechanical thinning of lithospheric root, but again the global geochemical characteristics of kimberlites (and their temporal relationships to LIPs that we describe; Extended Data Fig. 4) dominantly implicate mixing of asthenosphere (i.e., not deep-sourced plume melts) and metasomatized lithosphere (Giuliani et al., 2020; Tappe et al., 2011, 2017), which is completely consistent with the mechanism we are proposing.

2) that the ASPECT modeling results suggest that the volcanism would not occur at a particular point in time and space, but rather would linger over longer temporal (tens of millions of years) and spatial (800-1000 km) scales,

We agree with the reviewer that the ASPECT modelling results suggest that melting should linger over tens of Myr and horizontal scales of hundreds of kilometres. Indeed, our kimberlite dataset explicitly supports these predictions. We did not mean to imply that kimberlite volcanism is always tightly confined to a single space-time trajectory. A general spatio-temporal pattern does indeed emerge from our global statistical analysis (see revised Fig. 1c), on which we base our model, but there is also considerable variability about this general trend (see new Extended Data Fig. 3c), in line with the Reviewer's expectation and our observational dataset.

It must also be stated, however, that *most* kimberlites are expected to erupt during initial passage of the convective instability, which drives transient enhanced mantle melting and surface uplift/denudation. In effect, once the metasomatized lithospheric keel is stripped away, so too are most of the components required to generate kimberlite melts (principally, H₂O, CO₂). Thus, kimberlite frequency is expected to tail off over several Myr but may linger across the craton tens of Myr following instability migration. In fact, our geochemical analysis of Kaapvaal Craton kimberlites (Extended Data Fig. 9) supports this notion: here, we observe a step change from kimberlites exhibiting strong assimilation of metasomatized lithosphere to those exhibiting stronger asthenospheric signatures. Because this transition coincides with independent evidence for removal of ~30–50 km of lithosphere, collectively the data indicate that residual volatile-rich material is available for melting, but the asthenosphere is in closer contact with the mechanical lithosphere (i.e., most of the metasomatized thermal boundary layer has been removed). We also emphasise a distinct possibility that kimberlite volcanism occurring late in this cycle (i.e., post-convective removal) is less likely to be studied and/or radiometrically dated, because kimberlites that occur above thinned lithosphere (<160 km thick) are more likely to be barren in terms of their diamond contents (see Sun et al., 2018; now cited in the paper).

In the revised draft, we have provided a much more comprehensive description of the spatio-temporal pattern of kimberlite volcanism in the context of the ASPECT modelling (Methods section “*Melting calculations*”, lines 879-921); please see also our detailed response to Point 7 below.

3) that the convective instabilities should not be directly related to the time of continental breakup because they can be initiated early on in the rifting process and the duration of rifting before breakup occurs varies,

Again, we completely agree with the reviewer and our updated analysis shows this pattern much more clearly; we elaborate on this issue at length below.

and 4) that the thickness of cratons varies strongly (also within one craton): if kimberlites are related to thin(ned)-craton locations, then a more general “craton rejuvenation event mechanism” model would better explain kimberlite occurrences than the model proposed here. The model presented in this study does not explain kimberlites in other locations (for example, those not related to rifting, or far away from the breakup axis), or kimberlites that form at times not associated with continental breakup or plate tectonics (i.e., kimberlites laqqinq more than 30-50 Myr, or kimberlite aqes preceding continental breakup with tens (or more) of millions of years). Based on this manuscript, I consider the rift-instability model presented as a possible mechanism for craton

rejuvenation (and thinning) amongst a series of other mechanisms for craton rejuvenation.

Like the reviewer, we consider our rift-driven convective removal model to be one of several plausible mechanisms for craton rejuvenation and kimberlite genesis (we elaborate on this point below, and in detail in the revised manuscript/Methods). However, as we show, this model can explain most kimberlite occurrences, at least over the past 500 million years.

1) Small-scale convection starts as soon as there is a significant lateral thermal gradient between the rift and surrounding regions as described in the manuscript (Buck, 1986, Boutilier and Keen, 1999, Gao and all, Baikal rift, 2003, Wilson et al. 2005 Rio Grande rift, Van Wijk et al. Rio Grande rift 2008, Simon, Huismans et al 2009), when the mantle lithosphere has been thinned below the rift zone. This may occur during the continental rift phase, as illustrated by the rift animation in the supplemental material. This point in time may precede, however, continental breakup several millions of years, but also by many tens (or even more) millions of years; small scale convective instabilities may also be associated with rifts that never break up (see Buck, 1986, Van Wijk et al., 2008, Gao et al., 2003). What does this mean for the suggested relation between continental breakup and kimberlite ages/occurrences?

Because there is often considerable uncertainty around the timing of rift initiation (where onset is either very poorly constrained, unknown, or protracted over several tens of Myr), we deliberately performed our main statistical analysis using continental breakup time as a point of reference, as these are underpinned by multiple independent and internally consistent plate tectonic constraints. Using continental breakup times provides a more consistent datum with which to hang the temporal record of kimberlites.

We are therefore in complete agreement with the reviewer that rifting may precede continental break-up by a range of times from a few million years to many tens of million years (or even more). Our cross-correlation analysis of relative timing is compatible with this situation (Fig. R1b), showing $ccf(\Delta F, K)$, with negative lags indicating kimberlite eruption post breakup, and positive lags indicating eruptions pre-breakup. Fig. R1b (Fig. 1b) shows a distinct peak in the cross-correlation function (CCF) at -26 Myr (i.e., ΔF leads, and kimberlites lag local break-up by 26 Myr). However, the range of lags (h) where the CCF exceeds the 95% confidence interval spans from $h = -75$ to 40 Myr. Correlation begins to increase from -40 Myr before breakup (lag $h \approx 40$), peaking at around 26 Myr after breakup ($h = -26$), dropping sharply around $h \approx -50$, and falling below the 95% CI at $h \approx -75$, 75 Myr after breakup. Although the global data show a distinct trend, there is considerable scatter.

The sub-set of African and South American kimberlites shows a similar pattern, with a distinct peak at a lag of 30 Myr (i.e., post-breakup), again accompanied by considerable spread in observed lags (Fig. 1d). The natural variability in the kimberlite data arises from various processes, as noted by the reviewer. The variation in lags is consistent with the observation that rifting precedes continental break-up on timescales of approximately 20-50 Myr. For example, in a local situation where rifting is sufficient to trigger a convective instability 50 Myr before break-up, and a peak in kimberlite generation by our proposed mechanism occurred 26 Myr after the onset of this rifting, this kimberlite bloom would *lead* break-up by 24 Myr. Thus, natural variability in the duration of pre-breakup rifting can explain the broad zone to the right of the peak in the CCF shown in Fig. R1b, which shows some significant positive correlation from around 40 Myr before breakup ($h=40$) leading up to the peak at -26 Myr after breakup ($h=-26$). Although we acknowledge that kimberlites can precede breakup, our analysis, specifically the peak in rank correlation (Extended Data Fig 2a) which takes into account the effect of shorter lags, suggests that the majority of kimberlites post-date continental breakup by several tens of Myr.

We have adapted our analysis significantly to address the reviewer's point, and more comprehensively explicate the processes involved. In response to this reviewer's point 2 below we now include South America and have tested how earlier rift-related formation of convective instabilities may impact our regional case study. To do this, we have extended the study period to include the interval 40 Myr before breakup (i.e., a lag of -40 in Fig. R1c, guided by the point where the CCF for $(\Delta F, K)$ falls below the 95% CI at $\sim h=40$ pre-breakup in Fig. R1b; Fig. 1b). This captures kimberlites that erupted between rift onset and continental breakup. Importantly, this analysis (Fig. R1; updated Figs. 1c-d) confirms that the kimberlite generation and migration initiates prior to full continental breakup, consistent with our geodynamic simulations (and the reviewer's valuable point). We also find

Figure R1: Temporal relationships between tectonics and kimberlites | **a**, Frequency distribution of kimberlites through geologic time (data from ref. ⁶), showing peaks coinciding with the breakup phase of supercontinent cycles (from ref.¹⁸). **b**, Cross-correlations between ΔF and global kimberlites 6 spanning 500–0 Ma (Methods), showing dominant time lags at -26 ± 4 Myr (i.e., fragmentation leading kimberlites); positive correlations show that kimberlites are linked to continental fragmentation, not assembly; dashed blue lines show 95% confidence intervals. **c**, Box and whisker plot for Africa and South America showing time lags between continent separation and kimberlite eruption (negative number signifies rift-phase) versus distance from adjacent rifted margins of Gondwana ($n=169$; Methods); shaded pink field shows a general increase in distance between kimberlites and rift margins in the 20–30 Myr following breakup. Lower and upper hinges of boxes show the 25th and 75th percentiles; thick black line denotes the 50th percentile. Upper and lower whiskers extend to the largest and smallest values no further than 1.5 times the inter-quartile range (IQR) from the respective upper and lower hinge. **d**, Time lags between onset of local continent separation and eruption of those kimberlite clusters shown in (c) (Methods).

that there appear to be two distinct peaks of kimberlites, one (relatively minor) the result of convective instabilities that started to form from initial rifting, and the other, higher amplitude peak, the result of convective instabilities formed during rift necking. Our simulations predict that the latter pulse should produce larger and more powerful convective instabilities. We feel that this additional analysis has added immense value to the manuscript and thank the reviewer once again for highlighting this point. We have clarified the above points in the revised manuscript (lines 61-72, 277-285).

Figure R2: Kimberlite spatial distribution, lithospheric thickness and migration into cratonic interiors | **a**, Global map of kimberlites younger than 500 Ma (i.e., those used in our main analysis; Fig. 1b) from ref.⁶ ($n=860$) plotted on an interpolated map of lithospheric thickness from ref.⁶³ (continent-ocean boundaries (COB) shown in white are from GPlates⁵⁴; <https://www.gplates.org/>). The kimberlites are coloured by their radiometric ages⁶. **b**, Thickness distribution of lithosphere (point sampled from ref.⁶³; Methods) associated with those kimberlites younger than 250 Ma ($n=722$); note that most kimberlites are located in regions of lithosphere >200 km thick (mean=207 km and mode=209 km). **c**, Scatterplots showing distance from the adjacent rift margins versus time lag for clustered kimberlite eruptions (coloured by age), whereby positive values signify time after continental breakup (at $t=0$ Ma), and vice versa. The best fit regression line is shown in blue, and grey shaded region shows the 95% confidence interval for this line. Note that there is considerable scatter especially post-breakup (hence the low R^2 value), but the gradient of the line—indicative of the migration rate between 17 and 21 km Myr^{-1} —is consistent with analytical models and thermomechanical simulations (Fig. 3a).

A key implication of this result is that kimberlite volcanism is also likely to be associated with “failed” rift systems (those that do not progress to full continental breakup, e.g., Davis Strait & Baffin Bay to west of Greenland). Incidentally, this appears to be the case for many kimberlite fields of the Siberian Craton referred to by R#2 (see our response to their point 2 below). Critically, our cross-correlation analysis approach enables us to consider kimberlite distributions prior to breakup (positive lags for $ccf(\Delta F, K)$ in Fig. 1b), and supports the reviewer’s suggestion that convective instabilities can form much earlier. We have clarified in the paper that chains of convective instabilities commence during rift onset and may therefore be associated with “failed” rifts.

2) The geodynamic model predicts instabilities (and thus kimberlites) on both sides of the ocean basin. Africa is discussed in the manuscript; is this observed as well in South America?

The spatial distribution of kimberlites in our global database was not shown in our original manuscript, but we show it now in Extended Data Fig. 3a of the revised manuscript (Fig. R2). Within the period of interest (500–0 Ma), our database contains 82 kimberlites from South America (~10% of total), predominantly from the Brazil region, and largely age-equivalent to those kimberlite clusters in Africa (Fig. R2). These Brazilian kimberlites were originally not included in our regional analysis, which explicitly targeted southern Africa (for the reasons stated in the manuscript). We have now updated our analysis to include these South American kimberlites, and appreciate it is sensible to do so given that both groups are related to the same phase of Gondwana breakup, and sample similar cratons.

To perform this revised analysis, we updated the kimberlite database of Tappe et al. (2017) to include some additional recent radiometric ages of Brazilian kimberlites (Rosário do Sul, Canstrada-1, Limeira-1, Indaiá, Pântano, Três Ranchos and Lemes; relevant papers are now cited in the Methods). Almost all these kimberlites erupted after regional continental breakup. As we stated in the manuscript, it has been noted that kimberlites tend to erupt closer to the Brazilian rifted margins earlier in the rifting cycle (Hu et al., 2018). Owing to geographic features that lead to poor exposure, this region lacks the wide spatial distribution and intensity of exploration afforded by the South African case study. Fig. 1c-d (R1c-d) shows the results of including these additional South American kimberlites in our analysis. We note that these plots contain more data points since we extended the time limits of our analysis (in response to reviewer’s point 1). We have updated the main text (lines 50-59) and Methods (lines 504-654) to describe these updated analyses; and can conclude that the spatial pattern of kimberlites in South America is broadly consistent with that of southern Africa, and with our geodynamic model.

3) Craton bases are not at a constant depth- dozens of seismic studies indicate that the base of a craton varies between locations, also within one craton. These variations in thickness may be up to tens of kilometers, or, according to some studies, maybe even as large as 100 km across a craton. Such thickness variations are of similar size (or larger) than those predicted by the rifting-instability model. It is therefore likely that in some locations, the base of the craton is at such shallow depths that melting would occur. If removal of the base of the craton would induce formation of kimberlitic magma, we’d expect that there is a relation between the depth of a craton-base and kimberlite occurrence. Does this relation exist? Would then a craton-rejuvenation model

describe the occurrence of kimberlitic magmatism better, and the rift-instability model would be considered a rejuvenation mechanism?

We do consider that our rift-instability model is a craton rejuvenation mechanism. We elaborate on this topic in response to points 4 and 5, below, and hopefully you agree that our additions to the paper and Methods now make this point explicitly clear.

The reviewer asks if there is any relation between the depth of a craton-base and kimberlite occurrence. To address this question, we analysed the relationship between kimberlite occurrence and lithospheric thickness using the LITHO1.0 model of the crust and lithosphere (Pasyanos et al., 2014). We performed this analysis using open-source GIS software QGIS (<https://qgis.org/>) with additional open-source GRASS GIS processing tools (<https://grass.osgeo.org/>). Starting with the vector point map of lithospheric thickness from LITHO1.0, we performed a surface interpolation with regularized splines under tension, using the GRASS function, `v.surf.rst`, to generate a 0.1° cell size raster map of lithospheric thickness (shown in Extended Data Fig. 3a). We then point sample the lithospheric thickness from the interpolated raster map at each kimberlite location (Extended Data Fig. 3b).

Our analysis shows that kimberlites are most typically emplaced in regions of thick lithospheric mantle, with a mean thickness of 207 km and mode 209 km (n=722) (Fig. R2b; Extended Data Fig. 3b). Therefore, accepting that the bases of cratons are locally undulous, kimberlites are predominately found in regions of thicker lithosphere. This is likely because high-pressure regions of cratonic root can support stable carbonate phases (Fig. 3b) thought necessary to generate melts of kimberlite composition (e.g., Gudfinnsson & Presnall, 2005). We have noted this observation in the revised paper (lines 123 and 213). The associated methods have been added to the paper (lines 655-668).

The reviewer asks whether in some areas subject to substantial lithospheric thickness variations “the base of the craton is at such shallow depths that melting would occur”. We believe that melting of kimberlites at such shallow depths unlikely, first because any such regions will potentially have already lost—or indeed never had—the metasomatized (hydrous, carbonate-rich) keel thought necessary to enable kimberlite melting.

4) I would suggest removing the mantle plume discussion from the manuscript because 1) it introduces another layer of uncertainty and 2) this discussion is, in my opinion, not crucial to the manuscript. I understand why the authors included the discussion, but the problem is that there are more conceivable explanations for kimberlites than only plumes or rifts. So, it is not so that if the plume-relation is disproven, the conclusion is that the rift-model must be correct.

A plume could be viewed as a possible craton-base rejuvenation event, just like the rift-instability model, and several other mechanisms.

Also, the relation between breakup and mantle plumes is more complex than the analysis in this manuscript indicates (and this would be a manuscript by itself). Complexities include the relation between dynamic uplift, onset of magmatism, and onset of rifting (all poorly constrained), plumes that do not cause continental breakup (Emeishan Large igneous province, Afar, and other examples), and a new body of work that suggests alternative explanations for plume breakup magmatism (see for example E. R. Lundin and A. G. Doré,

We agree with the reviewer that it is not necessarily true that if the plume-relation is disproven, the conclusion is that our rift-model must be correct. At the same time, plumes represent an important model (and arguably the leading model) for kimberlitic magma genesis (Torsvik et al., *Nature*, 466, 352-355, 2010). If we were to remove the mantle plume discussion from the manuscript, we would generate considerably more questions among the wide section of the community that is familiar with and potentially supports this model. We note the complexities mentioned by the reviewer, and we do not deny any of these; indeed one of us (Jones) has a track record of documenting relationships between dynamic uplift, onset of magmatism and onset of rifting, which are in some cases constrained to sub-Myr resolution (e.g., Jones et al., *EPSL* (2003) 216, 271; Saunders, Jones et al., *Chem Geol* (2007) 241, 282; Jones et al., *J Geol Soc Lond* (2012) 169, 745; Parnell Turner, Jones et al., *Nature Geosci* (2014) 7, 914; Jones et al., *Nature Comms* (2019) 10, 1). As the reviewer recognises, a full discussion of the plume model would require a separate paper, and we agree that such a paper would be useful to write in future. However, the present manuscript is focussed on our new observations that most kimberlites spanning the past billion years erupted approximately 30 million years after the onset of continental fragmentation, and that kimberlites tend to erupt closer to the rift boundary earlier in the rifting cycle and migrate inboard of the rift over time, suggesting an association with rifting processes. We are therefore convinced that is appropriate and expected to discuss how well the plume model can explain these primary observations.

In the revised manuscript, we have therefore retained the discussion of the plume model, but we have taken multiple steps to avoid giving the impression that if the plume-relation is disproven then the rift-model must be correct, and to mention other potential models for kimberlite magma genesis.

We have changed "Testing these models..." to "Testing these and other models..." to emphasise that plume models are not the only alternative. We have added the following text to the Methods (lines 342-418) and refer to this in the main manuscript (lines 19 and 102-106):

"The observed relationship between pulses of kimberlite magmatism and episodes of global plate reorganization (Fig. 1) implies a tectonic mechanism for kimberlite magma generation. Furthermore, observations that kimberlites are only emplaced on cratons^{1,2,3}, and that kimberlite emplacement is often associated with uplift and erosion^{51,52}, suggest that kimberlite magma generation may be related to thinning and rejuvenation of cratonic lithosphere. We therefore review mechanisms proposed for thinning cratonic lithosphere, in order to assess which mechanisms have potential to explain both generation of kimberlite magma by mantle melting and an association with global plate reorganization.

Lithospheric thinning can be accomplished by erosion of Earth's surface. Cratonic erosion is often associated with kimberlite emplacement^{51,52} but the erosion rates are too slow to drive adiabatic upwelling and consequent melting of the sub-lithospheric convecting mantle. Lithospheric thinning in response to

lithospheric extension is a well-understood mechanism of generating mantle melting by decompression, but the lack of evidence for tectonic extension of cratons means that this mechanism cannot explain kimberlite melting.

The remaining mechanisms for lithospheric thinning involve removal of the lowermost lithosphere, or keel. The keel is inherently quasi-unstable because it is denser than the underlying asthenosphere. Lee et al. (2011)²⁸ identified five processes that might critically destabilize and remove cratonic keel, which we now review. Cratonic keel is constantly cooled by conduction through the overlying lithosphere, leading to thickening, destabilization and convective removal of the keel²⁸. Rayleigh-Taylor instabilities and lithospheric edge-driven convection are special cases of this type of convective removal. The planform and vigour of convective instabilities within the keel are known to be influenced by nearby rifting^{25,26,35}. Until now, it was unclear whether rifting can cause convective removal of keel at large horizontal distances (>200 km). Also, whilst convective instabilities near rifts have been argued to cause melting^{25,27}, it is unclear whether convective instabilities beneath thick cratonic lithosphere can cause kimberlite melting. Cratonic keel is constantly warmed by the underlying asthenosphere. A mantle plume head and/or smaller scale asthenospheric convection cells might locally enhance this warming, leading to weakening and convective removal of keel³⁶. Thermal weakening might be augmented by advection of heat into the keel if sufficiently large degrees of mantle melting occur. These plume models provide a well-established hypothesis for kimberlite melting, but they do not explain a global association with rifting. Cratonic keel overlying subduction zones might be destabilized by infiltration of subduction-related fluids and melts, leading to rheological weakening, followed by convective removal³⁶. Although this mechanism specifically involves melting, it does not explain the lack of arc geochemical signatures in most kimberlites, or the co-location of kimberlites and rifted margins (as opposed to subduction zones). Cratonic keel might be removed by basal traction in response to relative motion between the lithosphere and asthenosphere³⁶. This mechanism does not provide an obvious cause of melting, and nor can it explain a global association with rifting; for example, motion of the southern African lithosphere relative to the underlying asthenosphere was low throughout the Mesozoic when many of the southern African kimberlites were emplaced. Cratonic keel might undergo sub-horizontal viscous flow that acts to smooth out basal lithospheric topography³⁶. This speculative mechanism has received little attention, and it does not provide an obvious cause of melting.

This discussion shows that all the suggested processes that can potentially explain both generation of kimberlite magma by mantle melting and an association with global plate reorganization involve convective removal of the lowermost lithosphere (or keel). There are two main open questions. The first relates to the process that causes the destabilization, which might involve rifting and/or mantle plumes. The second relates to the relative role of small-scale convection cells and mantle plumes in kimberlite melting. The purpose of this paper is to address these questions through a combination of observations and modelling.”

5) The delamination event discussed in line 251 is puzzling. Is this the convective removal of the craton base (if so, please consider replacing the term delamination with convective removal)? The geodynamic models do not show that ~50 km of material is removed from the base of the craton (this is hard to see though); rather, the layer that is removed seems less than about 30 km.

Does this matter? What controls the height of the layer that is removed by the instabilities? When a map is plotted of the depth of the base of cratons, how much variation is there? A quick literature search suggests that the craton base varies strongly, also within one craton. Why would 30-50 km thinning cause kimberlite formation, but other rejuvenation events (or craton thinning events) not?

As the reviewer suggests, we have replaced the term “delamination” with “convective removal”.

The layer that is directly removed by the convective instabilities is the “thermal boundary layer” (TBL) between the mechanically rigid mantle lithosphere and the constantly convecting asthenosphere. We refer to the TBL as the lithospheric keel (or simply, keel) in our paper. The TBL is shown as “variably metasomatized TBL / keel” (beige colour) in Fig. 2. The exact nature of the TBL beneath cratons, and, how long the TBL can remain isolated from the underlying mantle convection, are open questions. Global geophysical studies consider that the TBL has broadly similar composition to the asthenosphere, but it is cooler, denser, and is likely intermittently involved in mantle convection on time periods of tens to hundreds of millions of years. In this framework, the TBL thicknesses we estimate in Methods “*Xenolith geotherms*” (Extended Data Fig. 5) are time-averages (all c. 35 km) and the actual TBL thickness at any given time and location could vary from a few km to perhaps twice the time-averaged thickness. Geochemists and petrologists emphasise that the TBL beneath cratons is metasomatically enriched, may have a significantly different composition from the asthenosphere, and may remain isolated from asthenospheric convection for up to a billion years. Given these uncertainties in the composition, thickness and age of the TBL, we would be reluctant to emphasise the difference between convective removal of a 30 km or a 50 km thick TBL layer.

Several maps showing the depth to the base of cratons (i.e., lithospheric thickness) are available. All are based on seismic surface wave observations. Some show the depth to the base of the seismically fast layer; others apply a velocity-temperature conversion and show the base of the thermal lithosphere (modelled as the middle of the TBL described above). The maps based directly on seismic velocity show variability of order 10 km in thickness at a spatial scale of order 100 km (e.g., Extended Data Fig. 3a). We would be reluctant to interpret this small-scale structure in the context of the present paper, given the resolution of the underlying seismic models, the potential for strong compositional (and hence seismic velocity) variation within the TBL, and the expected spatio-temporal variations in the thickness of the TBL arising from normal plate cooling and mantle convection processes. The thermal lithospheric thickness maps (e.g., Priestley et al.) avoid these uncertainties by focussing on features more than many hundreds of km in scale, which are arguably more likely to represent real, long-term lithospheric thickness variations. The various lithospheric thickness maps all suggest that craton boundaries inferred from regions of thick lithosphere do not directly match the craton boundaries inferred from surface geological mapping. This observation is now quite well established but a more detailed understanding of why the two differ is not yet established, to our knowledge. We are, however, confident in the spatio-temporal co-relationships between kimberlite distributions and rifting presented in this paper. Furthermore, our proposed explanations are based on generally accepted models for the thermal structure of plates,

and the relationship between plates and mantle convection. Given these strong constraints, we believe our arguments are not fundamentally compromised by possible lateral variations in cratonic lithospheric thickness within kimberlite fields, and their relationships to surface geology.

The question of why convective removal of 30–50 km TBL might cause kimberlite formation, but other rejuvenation events not is now discussed in detail in Methods “*Relationship between kimberlite formation and craton rejuvenation*”. This section assesses all proposed mechanisms for cratonic thinning according to their potential to generate kimberlite melting. In the main paper, we first deal with the question of whether plumes are necessarily involved in cratonic rejuvenation related to kimberlite generation and find that they are not (lines 73-99). We then summarise the more comprehensive discussion (in the Methods, see previous point) in the main text (lines 100-153), as follows:

“We propose that rifting triggers the migrating pattern of kimberlite eruptions hundreds of kilometres inboard of the rift over tens of millions of years (Fig. 1c). We review the proposed mechanisms for tectonic and magmatic rejuvenation of cratons to assess which have the potential to explain both kimberlite melting and migration, and an association with rifting (Methods). Cratonic thinning by surface uplift and exhumation can be triggered by rifting²⁵ but occurs too slowly to generate melting. In cases where rifting occurs on the edge of the craton, the cratonic lithosphere appears not to stretch and thin mechanically. This interpretation is based on geologic observations of a lack of horizontal tectonic motions within cratons^{9,10,11,12,13}, and on dynamical models which show that the lithosphere >300 km inboard of rifted margins is not thinned by extension^{26,27}. The remaining potential cratonic rejuvenation mechanisms involve removal of the basal lithosphere²⁸, or keel, and the subset that can potentially explain both kimberlite melting and an association with rifting all involve mantle convective removal of keel (Methods). The open questions are: how does rifting trigger convective instability in adjacent cratonic lithospheric keel; and can removal of this keel cause melting of appropriate volume and composition?”

Diamond-bearing kimberlites are exclusively found in thick cratons (150–250 km)^{1,3,9,10,13,15} (Extended Data Figs. 3a–b). An inevitable consequence of fragmenting cratons is the generation of a physically steep-sided lithosphere–asthenosphere boundary (LAB)¹⁴. The steep edge of the LAB prompts edge-driven convection³⁴, in which convective downwelling of lithospheric keel (hereafter, keel) occurs on the side of the edge further from the rift, as demonstrated by numerical modelling^{25,26,27,35} and seismic tomography beneath modern passive margins³⁴. Though edge-driven convection is triggered by rifting and can remove keel near the rift, it is not clear whether it can explain convective removal of keel further (>300 km) from the rift. One possible mechanism is Rayleigh–Taylor instability – a well-described mantle convective process that can potentially occur beneath mature continental or oceanic lithosphere^{25,28,35,36,37,38}. This instability is driven by the density contrast between colder lithosphere and hotter asthenosphere, and metasomatism of the keel can augment this negative buoyancy driver in the case of cratons^{21,28}. Rayleigh–Taylor instability could therefore cause convective removal of keel, and subsequently a thinning of the lithosphere. Furthermore, convective downwelling is balanced by upwelling of asthenosphere that can feasibly cause melting, particularly if detached keel veined by hydrous/carbonate-rich metasomatic phases^{14,20,39} is entrained in the upwelling. Petrological observations^{14,21,40,41} and dynamic models³⁷ indicate that substantial removal of cratonic keel (tens of km thick) can occur abruptly (over

several Myr)^{11,14}, suggestive of a convective process. The question is then whether rift-related edge-driven convection can destabilize, and partially melt, adjacent keel hundreds to thousands of kilometres inboard of rifted margins via Rayleigh–Taylor instabilities (hereafter, instabilities).”

6) The activation energy is an important parameter in the development of instabilities- whether they form at all, and if they form, this parameter affects for example the size of the instabilities. See van Hunen, J., Zhong, S., Shapiro, N.M., and Ritzwoller, M.H., 2005, New evidence for dislocation creep from 3-D geodynamic modeling of the Pacific upper mantle structure: Earth and Planetary Science Letters, v. 238, p. 146–155, doi: 10.1016/j.epsl.2005.07.006. How do the models change when a different activation energy value is chosen for the mantle?

We agree that the activation energy is an important parameter in generating instabilities. The modelling described in our submitted manuscript did test the effects of different activation energy values. We found that modifying this parameter within experimental uncertainties does not change our conclusions (Fig. 4). We justify this statement for the analytical and numerical model in more detail below, and we have included the key points of this discussion in the revised manuscript and Methods section.

Our analytical, scaling law-based model of convective instability chains implicitly includes activation energy via the viscosity, which is proportional to the exponential of the activation energy (e.g., Turcotte & Schubert, 2001). The specific effect of variable viscosity/activation energy on our scaling law-based model is shown by comparison of Models 1 and 3 in Fig 3a. Model 1 represents asthenospheric mantle with the same viscosity/activation energy as the basal lithosphere, and Model 3 represents a limit where the asthenospheric mantle has very much lower viscosity and activation energy. For example, if the asthenosphere has a very low viscosity/activation energy, the instability wavelength is reduced from 67 to 54 km and the growth time reduced from 2.5 to 1.5 Myr, relative to the constant viscosity case (Extended Data Table 1a). The resulting change in lateral propagation rate is 7 km/Myr (compare Models 1 & 3, Fig. 3a). These predicted differences are probably upper bounds on the actual uncertainties because the viscosity/activation energy variation across the lithosphere/asthenosphere boundary is likely a continuous decrease, rather than a sharp and significant decrease. Conrad & Molnar (*GJI*, 1997, §6.1) discussed the case where the viscosity/activation energy decreases continuously and exponentially across the basal lithospheric layer. They showed that this case is the same as the case with no viscosity/activation energy variation (Model 1, Fig. 3a), in the limit where the length scale for the viscosity variation is similar to the lithospheric thickness.

Our numerical modelling of basal lithospheric instabilities using the ASPECT code explicitly includes the activation energies for diffusion and dislocation creep. The model case shown in Fig. 2 is based on the experimental results of Hirth & Kohlstedt (2004). In agreement with observed seismic anisotropy in the upper mantle, lithosphere and shallow asthenosphere viscosity in our models is dominated by dislocation creep, with activation energy values of 530 kJ/mol in the basal lithosphere and 480 kJ/mol in the asthenosphere (Extended Data Table 1b). These values for dislocation creep are within the acceptable activation energy range of 360–540 kJ/mol determined by van Hunen et al.

(2005), based on comparing seismic observations with geodynamic modelling results for a Pacific Ocean case study. Following the suggestion of the reviewer, we conducted additional experiments where we varied the asthenospheric activation energy values within experimental uncertainties of ± 40 kJ/mol while keeping all other parameters identical. When we lower the activation energy to 440 kJ/mol, the shallow asthenosphere and metasomatized layer exhibit a viscosity that is about 2 times smaller than for the reference case. This generates lateral propagation rates for the instabilities that are roughly 2 times larger than for the reference model, which is in excellent agreement with analytical Model 3 (Fig. 3a). When increasing the asthenospheric activation energy to 520 kJ/mol, the shallow asthenosphere becomes similarly strong as the overlying lithosphere and asthenospheric delamination is suppressed. This leads to unrealistic thermal thickening of the lithosphere, which is why we do not consider this scenario as realistic.

Summarizing these analytical and numerical results, we find that uncertainty in variation of activation energy across the lithosphere-asthenosphere boundary leads to uncertainties of <10 km in instability wavelength, and ~ 10 km/Myr in lateral propagation rate of the instability chain in our models. These uncertainties are rather small compared with our observations (Figs. 1, 3) and do not affect our conceptual model (Fig. 4).

We have added the following text to the Methods section (lines 826-845):

“Sublithospheric viscosity and in particular activation energy constitute important parameters for the development of instabilities (van Hunen et al., 2005). In agreement with observations of seismic anisotropy in the upper mantle, shallow asthenosphere viscosity in our models is dominated by dislocation creep, where we use experimentally derived wet olivine activation energy values of 480 ± 40 kJ/mol (Hirth & Kohlstedt, 2004) that are representative for the shallow asthenosphere. Notably, these values for dislocation creep lie within the independently derived activation energy range of 360–540 kJ/mol (van Hunen et al., 2005). We conducted alternative modelling runs where we varied the asthenospheric activation energy within experimental uncertainties while keeping all other parameters identical. When we lower the activation energy to 440 kJ/mol, the shallow asthenosphere and metasomatized layer exhibit a viscosity that is about 2 times smaller than for the reference case. This model generates lateral propagation rates for the instabilities that are roughly 2 times larger than for the reference model, which is in agreement with analytical Model 3 that constitutes the low-viscosity end-member of the analytical modelling suite (Fig. 3a).”

7) Where and when does ASPECT predict that melting occurs? This is important to know, as the instabilities permanently remove craton mantle, and melting may continue over large spatial and temporal spaces. Why would melting just be an isolated event as suggested by the conceptual figure, and not extend over longer timescales and spatial scales as suggested by the ASPECT modeling? In this respect, the conceptual model seems not supported by the geodynamic modeling results.

We agree that the question of how the locus of melting moves through space and time is important to our study. In the revised manuscript, we now explicitly address the question of how the size and location of the melting region relates to the propagating chain of convective instabilities in two new paragraphs of sub-section, “*Melting Calculations*” (Methods, lines 879-917):

“Neither the analytical modelling of Rayleigh-Taylor instability chains nor the numerical modelling carried out in ASPECT directly predicts where and when melting occurs. However, convectively driven upwelling is one of the important processes that can drive mantle melting, and the ASPECT modelling does indicate the relative locations and strengths of the convective upwellings through time (Extended Data Fig. 6). The strongest convective upwelling at any time is always associated with the leading convective instability in the propagating chain. The leading cell has two upwelling regions, one on either side of the central downwelling. The upwelling closer to the rift is the stronger of the two, though both have upwelling regions that are adiabatic and can potentially drive melting (subject to additional conditions). Earlier convection cells in the propagating chain remain visible to the rift-ward side of the leading cell. Circulation in the older cells is weaker than in the leading cell, though still adiabatic.

Another factor that could enable melting beneath thick cratonic lithosphere is that the upwelling mantle is enriched in volatiles. The ASPECT modelling shows that volatile-enriched, metasomatized keel detached by the downwelling limbs of the small-scale convection cells is partially recirculated in the upwelling limbs. The upwelling limb on the rift-ward side of the leading convection cell contains most re-circulated keel, together with a significant component of background asthenospheric mantle. Older convection cells may also contain volatile-enriched material but with a greater proportion of asthenospheric material. We note that the detailed spatiotemporal pattern of re-circulated keel material is difficult to interpret because the convection cells interact with the bottom of the computation box at 300 km. Nevertheless, the ASPECT modelling shows adiabatic upwelling of asthenosphere that is variably enriched with recently detached metasomatized keel material. Thus, it is feasible to infer that propagation of the instability chain will potentially be accompanied by kimberlite melting. The bulk of melting is predicted to occur across the footprint of the leading convection cell in the instability chain, and smaller fluxes of melting in the wake of the leading instability may be expected in some cases.”

Our observational database shows that kimberlites are not usually isolated events. Kimberlite fields typically have diameters of 30–50 km (labelled on Fig. 4c), and we already note (lines 196-200) that this spatial scale is comparable to the wavelength of convective instability predicted by both our scaling law and numerical models. Furthermore, kimberlites tend to occur over a wide area several tens of Myr post-breakup (Fig. 1c; Extended Data Fig. 3c). Thus, our observations show that melting does indeed continue over spatial and temporal scales comparable to, or larger than, a single convective instability, as suggested by the reviewer.

Our modelling results are in line with these observations. The numerical modelling carried out in ASPECT does not directly predict where and when melting occurs; as already stated in the Methods (“*Thermo-mechanical models*”), the ASPECT modelling focusses on first-order thermo-mechanical processes and does not explicitly account for chemical alterations, melt generation and magma ascent. However, convectively driven

upwelling is one of the important processes that drives melting, and the ASPECT modelling does show the locations and relative strengths of the convective upwellings through time. The strongest convective upwelling at any time is always associated with the leading convective instability in the propagating chain. The leading cell has two upwelling regions, one on either side of the central downwelling. The upwelling closer to the rift is the stronger of the two, though both have upwelling regions that are adiabatic and can potentially drive melting (subject to additional conditions). Earlier convection cells in the propagating chain remain visible to the rift-ward side of the leading cell. Circulation in the older cells is considerably weaker than in the leading cell.

The melting calculations (Methods: “*Melting calculations*”, lines 878-952) show that melting can only occur where the upwelling limbs of the convective instabilities contain significant volatiles. This is a key point: it means that although Extended Data Fig. 5 shows a chain of convective upwellings in the wake of the leading instability in the chain, this does not imply that melting will occur equally within all the upwellings across the entire region. In our conceptual model, we propose that the volatile-rich material originates as metasomatized lithospheric keel. The ASPECT modelling shows that the central downwelling limb of the leading convective instability in the propagating chain detaches keel material, and some of this keel material is re-circulated within the adjacent upwelling limbs (Fig. 2; Extended Data Fig. 5). Thus, we predict that melting should occur within both upwelling limbs of the leading convective instability. The surface expression will be a field of active kimberlite volcanoes with diameter similar to the convective instability (c. 50 km), in agreement with observations. The older convective cells in the chain are expected to contain little freshly detached keel material because most of the keel was removed efficiently when the cell first formed. Any keel material re-circulating within these older cells is likely to have been significantly depleted of volatiles by melting associated with the initial cycle of upwelling. Thus, we predict that considerably less melting should occur in the upwelling limbs of the older cells in comparison with the leading cell, in agreement with observations. We also predict that minor melting within the older cells will not be sourced from freshly detached lithospheric keel. This prediction is in line with the observed transition from predominantly Group 2 kimberlites (high proportion of metasomatized lithosphere relative to asthenosphere in the source) to Group 1 kimberlites (low proportion of metasomatized lithosphere relative to asthenosphere in the source) through time across southern Africa (Extended Data Fig. 9).

8) Related to the former point: The formation of convective instabilities in the asthenosphere is a three-dimensional process. This is nicely shown in for example work by Ballmer and van Hunen, with applications in oceanic and continental domains. Their work suggests that small-scale convective instabilities are not necessarily associated with time-progressive volcanism-age-distance patterns are complex (duration of melting may be prolonged below oceanic lithosphere). How does this apply to continental domains (and in three dimensions)?

The reviewer makes a good point that the formation of convective instabilities is an intrinsically three-dimensional process. Because our analytical models and simulations are 1D and 2D, respectively, this is not a feature we can address categorically within this paper. However, we do not think this is a critical shortcoming of our work, because (1) it

is commonly observed (e.g., Jelsma et al., 2009) that kimberlite occurrences are confined to linear chains (which we argue below primarily reflects lithospheric thickness *and* composition); and (2) the age-distance relationship we show (Fig. 1c)—coupled with the fundamental dynamic constraints on kimberlite melting and ascent (i.e., melting occurring during/shortly after convective removal followed by rapid ascent)—indicate that prolonged, complex melting patterns (or lack thereof) inferred for thin oceanic lithosphere are not a feature associated with kimberlite petrogenesis.

Notwithstanding this, we consider the possible implications of this three-dimensionality in continental domains. Ballmer et al. (2010) describe 3D thermomechanical small-scale sub-lithospheric convection in oceanic crust, which self-organise as ‘Richter rolls’ that are broadly aligned with the plate motion. Because our convective instabilities form at the physically steep LAB beneath continental rifts, it seems inescapable that similar ‘rolls’ propagate across an extensive area of the cratonic keel. However, as we discussed in response to reviewer 2 (below), we do not expect kimberlites to form everywhere in cratonic interiors. Rather, where these instabilities trigger the delamination required to drive kimberlite melting is sensitive to: (1) the local thickness of the lithosphere (specifically, whether it is thick enough to sustain carbonates, a precursor to kimberlite melts, e.g., Gudfinnsson & Presnall, 2005); and (2) the availability of metasomatized material (sensitive to metasomatic/removal history). Given what the reviewer stated earlier about local variations in cratonic thickness, including channels (e.g., Celli et al., 2020), it is feasible that the ‘rolls’ either separate or do not come into uniform contact with the lithospheric root. This local thickening and channelling would explain the long-recognised tendency for kimberlites to be oriented in linear chains where lithosphere is thickest, broadly perpendicular to plate boundaries. Again, in contrast to the oceanic lithospheric domains described by Ballmer et al. (2010), we find evidence for a reasonably well-defined age-distance pattern to kimberlite volcanism (Fig. 1c; Extended Data Fig. 3c).

We propose the main difference between our model and that of Ballmer et al. (2010) is that the melting occurs during convective removal of metasomatized keel, which drives carbonate-rich melting, as opposed to upwelling of asthenospheric mantle and prolonged interaction with, and melting of, thinned oceanic lithosphere.

Again, kimberlite generation is a characteristically rapid process, fuelled by high volatile contents (Sparks et al., 2006), meaning the volcanism is unlikely to linger in one place for a long period (because the volatile components needed for kimberlite generation are quickly exhausted during melting). On the contrary, melting below oceanic lithosphere may be more prolonged and complex, owing to slow convective upwelling of the asthenosphere, and to heterogeneous geochemical assemblages (veined ‘marble cake’) alluded to by Ballmer et al. (2010).

Therefore, although the instability formation processes we refer to are broadly similar, the resulting patterns of volcanism are rather different due to the occurrence of delamination and major contrasts in petrology.

We have added a short paragraph to discuss this issue in the paper (lines 266-296):

“Sublithospheric convective instabilities are also thought to drive melting in the oceanic realm, where complex (or absent) age-distance patterns of volcanism have

been attributed to prolonged interaction of asthenospheric mantle with the thinned lithosphere³⁸. In contrast, a simple, systematic age-distance pattern emerges from our statistical analysis of kimberlites (Fig. 1c), although there is clearly variability about this general trend (Extended Data Fig. 3c). Our data and observations indicate that kimberlite melting beneath cratons occurs via convective removal of keel (Fig. 4), which drives characteristically transient explosive kimberlite eruptions^{2,3} that occur directly above the migrating locus of melting (Fig. 4). Indeed, kimberlite melting likely occurs in several phases reflecting the dynamic evolution of rifts. It is tempting to interpret the bimodal peak in kimberlite lag times—evident both in Africa and South America (Fig. 1d)—to relate to two main phases of instability growth and propagation: the first linked to rift onset with kimberlite magmatism peaking 20–40 Myr later (coinciding with continental breakup); and the second linked to rift necking with kimberlite magmatism peaking 25–45 Myr after breakup (Fig. 1d). Because convective instabilities can endure beneath cratons for tens of Myr (Fig. 2; Supplementary Animation 1), the longevity of kimberlite magmatism will be limited by the availability of metasomatized cratonic keel, which may be prolonged or ephemeral, depending on its rate of entrainment in convective upwellings and subsequent exhaustion during melting (Fig. 4). Under both scenarios, kimberlite compositions should evolve from exhibiting dominantly lithospheric signatures (reflecting a relatively high proportion of entrained keel) towards asthenospheric signatures (reflecting a lower proportion of entrained keel).”

Figure 1 caption: please replace rift separation with continent separation

We confirm that we have made this change to the caption of Figure 1 (in addition to several other places in the manuscript).

Referee #2:

That is an interesting and truly interdisciplinary paper suggesting indeed new idea about the origin of kimberlites. I appreciate that authors use global database of kimberlite locations and ages, although they focus more on African kimberlites. This work has several strengths, but also some weaknesses. There are however a number of important questions along with few smaller issues that authors do not answer. I suggest that authors should answer the questions in the revised manuscript.

We are very pleased that the reviewer appreciates the interdisciplinary nature of our paper and the use of global databases to quantify the links between global tectonics and kimberlites. We have answered all the major and minor questions below and have updated the manuscript accordingly to clarify these points.

1. Is accuracy of reconstructions of timing of continental break-up for the age more than 200 Ma enough high to resolve lag between kimberlite eruptions and break-up of only about 20 - 30 My?

The reviewer makes a valid point that continental reconstructions—used to calculate the fragmentation index utilized here to characterize global continental breakup—become increasingly uncertain further back in time. This effect is especially relevant prior to 200

Myr ago, because little oceanic lithosphere older than 200 Ma is preserved today, and seafloor isochrons at least partly underpin paleogeographic reconstructions (Müller et al., 2008; Gernon et al., 2021). We investigated this issue in some detail, both here and in an earlier paper (Gernon et al., 2021), and are confident that it should not affect our overall conclusions for several reasons.

- 1) Clearly there is some degree of uncertainty associated with plate tectonic models used to generate the ‘fragmentation index’ (Fig. R4). However, it is difficult to simulate uncertainty in the rate of change of fragmentation, ΔF , because it is not something that can be explicitly characterized; introducing random variation without a firm physical basis would introduce synthetic features in the data that may yield misleading results, defeating our overall objective. We therefore tackled this issue in two alternative ways. First, when calculating ΔF , we used a 9 Myr moving window (note that the total width is odd as we require a symmetric window, and the input data have a 1 Myr time step), which averages the model output over a relatively wide time interval. Second, uncertainty in breakup age may be partly accounted for by comparing alternative plate kinematic models; we used MER19 and MER21, which were developed using independent, internally consistent constraints (synthesis of palaeomagnetic and geological data and kinematic constraints). Below (point 5a), we demonstrate that using different indices for plate fragmentation does not significantly affect our overall result: it only shifts the main lag from -26 to -21 Myr (with an uncertainty of ± 4 Myr) after breakup.
- 2) Pre-200-Ma plate models are constructed through a synthesis of palaeomagnetic data and diverse strands of geologic data. Mapping of arcs, rift basins, passive margins and metamorphic facies, among other indicators, are used to identify the location and nature of plate boundaries. Kinematic constraints are used to assess and evaluate the balance and interpretation of palaeomagnetic and geologic data to produce a continuous, self-consistent model (for example, if an interpretation of an event at 800 Ma requires a large continent to then move at a velocity of 30 cm yr⁻¹ for the next 100 Myr to fit the data at 700 Ma, it is deemed that one of the two interpretations is unlikely and closer scrutiny of the data, or re-evaluation of the model is warranted (e.g., see discussion in Merdith et al., 2017b, *Precambrian Research* 299, 132)). A key aspect of these styles of models are that tectonic coherency must be maintained, that is, interpretations made in one part of the world do not nullify the data in other parts of the world (or at younger or older times). While this is not necessarily proof that the model is therefore the ‘true’ model, this coherency (in association with fitting what data are available) ensures that the model provides the best available solution to fit all the available constraints, globally.
- 3) The observed exponential distribution of kimberlites—which peaks in the Cretaceous (i.e., post-200 Ma; Fig. 1a)—means that most kimberlites included in our compilation (those spanning 500-0 Ma, used to produce Fig. 1b) are in fact younger than 200 Ma in age (comprising ~80% of the total).
- 4) Motivated by the reviewer’s point, we tested how uncertainty prior to 200 Ma might affect our results by isolating those occurrences younger than this age and running a separate Bayesian network to determine if the overall results (i.e., the absolute value of peak correlation and its lag) changed in any way (Fig. R4; now

shown in Extended Data Fig. 2d and referred to in the main paper). We found that eliminating kimberlites older than 200 Ma led to an increase in the strength of correlation between kimberlite frequency and ΔF (i.e., rate of change of fragmentation) from 0.41 to 0.52, and a very slight change in the time lag of this peak from 26 to 28 Myr after breakup. The main lag moves closer to that calculated in our regional analysis (25-35 Ma). This result indicates that inclusion of older kimberlites and breakup metrics may lead to a reduction in correlation and small reduction in the time lag of 2 Myr (although this is within the uncertainty of the estimate).

Overall, these additional statistical tests give us confidence that increased uncertainty prior to 200 Ma does not significantly impact our overall conclusion that kimberlites most typically lag continental breakup by time intervals of ~30 Myr. We now highlight this observation in the paper, referring to new Extended Data Figure 2d (lines 44-47).

Figure R3: Relationship between dynamic continental fragmentation and global kimberlites | **Left:** Cross-correlations between kimberlites⁶ and the rate of change of tectonic fragmentation (ΔF ; 9 Myr window) spanning 500 to 0 Ma, showing dominant time lags at -26 ± 4 Myr (i.e., fragmentation preceding kimberlites). **Right:** Cross-correlations between kimberlites⁶ and ΔF (9 Myr window) from 200–0 Ma (Methods) showing the strongest correlation ($\rho=0.52$) at a time lag of -28 Myr; dashed blue lines show the 95% confidence intervals.

2. If kimberlite eruptions are related to continental break-up, why are they located only in few places along the continental margin and not everywhere where enough thick lithosphere is situated not far away from the margin?

This question arises because we did not provide a map showing the relationship between our kimberlite dataset and the associated continental margins. We have included such a map in Extended Data Fig. 3a of the revised draft (Fig. R2 above). This map shows that kimberlites occur along almost all segments of existing or ancient continental margins where thick lithosphere exists inboard of them. The exceptions are areas where there are difficulties in mapping kimberlites (e.g., due to permanent ice cover), such as Greenland and Antarctica. However, the density of kimberlites does vary between margin segments. We also must stress that kimberlites won't necessarily form everywhere the lithosphere is thick enough. Some cratonic areas may not have experienced a prolonged history (spanning several billion years; see Pearson et al., *Nature*, 2021) of metasomatism thought necessary to explain the geochemical characteristics of most kimberlites. In addition, the formation of kimberlites will be sensitive to the previous rifting history (convective removal of an ancient keel may have occurred in an earlier supercontinent breakup, e.g., Rodinia).

3. While their model looks convincing for South African kimberlites, what about kimberlites of the Siberian craton? In particular, was there continental break-

up in eastern Siberia prior to Triassic and Jurassic kimberlites? Some recent publications related to the Siberian craton are overlooked, for example: Ernst R.E. et al. (2016) Russ. Geol. Geophys.; Polyansky O.P., et al. (2018) Russ. Geol. Geophys.; Sun J. et al. (2018) Chem. Geol.

Once again, this question was difficult for the reviewer to assess because we did not provide a map of our global kimberlite distributions. The map we now provide (Extended Data Fig. 3a) shows that our database does contain a significant number of kimberlite occurrences from Russia (N = 94; 11%). That map also shows the palaeo-continental margins to the north of Siberia, which are significantly younger. The nearby Eurasian Basin and Eurasian continent-ocean boundary, or COB, were active from approximately 50-60 Ma, whereas the more distal Eurasian COB to the east was probably fully developed from 145 Ma [Mandea and Gaina, 2013]. These passive margins therefore post-date the Late Triassic kimberlites (226-18 Ma; Sun et al., 2018) but overlap in time with mid-late Jurassic (161-144 Ma) kimberlites. Given the protracted nature of kimberlite volcanism in the Siberian Craton (spanning the Silurian to the Jurassic), these kimberlite fields are thought to be linked to major, long-lived extensional rift systems across the whole area related to the breakup of Pangaea, with major basins developing subparallel to the major terrane boundaries (see Griffin et al., *Tectonophysics* 310 (1999) 1–35). Specifically, these are the Urinsky, Ygiattinsk and Kempendyay rift systems, whose attributes are summarized in the compilation of Sengor and Natalin (2001, *Special Paper of the Geological Society of America* 352, 389), and which run subparallel (NE-SW) to the major kimberlite clusters. Indeed, Griffin et al. (1999) found that lithosphere thickness at the time of kimberlite eruption varied from approximately 190-240 km (consistent with our analysis above, Fig. R2b), and that thinning of the lithosphere by up to 50-60 km occurred locally along parts of the rift (as we document in South Africa). The locations of the Siberian rifts (on thick cratonic crust; Extended Data Fig. 3a) and orientations (parallel to mapped kimberlite clusters) suggest that they represent “failed” rift systems, most likely linked both spatially and temporally to the eruption of the Permo-Triassic Siberian Traps (Sobolev et al., *Nature* 477 (2011), 312). If this were the case, it would indeed corroborate our finding (inspired by R#1’s point) that kimberlites can erupt during the time between rift initiation and continental breakup.

Overall, this review suggests that the Siberian Craton kimberlite fields are consistent with what we observe in Africa/South America. However, because the precise age of rift activity in the Siberian region is largely inferential or unknown (see Sengor and Natalin, 2001), understandably we opted not to analyse the links between Siberian rift systems and kimberlites in the paper, instead focussing our regional analysis on two adjoining continents with well-dated kimberlites and rift kinematic constraints.

We hope that inclusion of the global map of kimberlite occurrences addresses the reviewer’s main question. We apologise for not citing all the Siberian papers referred to by the reviewer, largely due to space constraints. However, we now refer to this example in the main paper, with reference to Sun et al. (2018).

4. Authors convincingly show that mantle convection associated with the continental break-up can destabilize the lowermost layer of lithosphere, but why mantle plume can not do the same? Maybe in fact all processes that can destabilize lower metasomatized layer of continental roots can generate kimberlites including plumes and continental fragmentation?

We agree with the reviewer here, in that isolated occurrences of kimberlites could potentially owe their origins to mantle plumes and their interaction with lithospheric keels. However, for the reasons outlined above, the global evidence is inconsistent with a controlling role for mantle plumes in supplying melts to the kimberlite source, specifically because the parent melts are considered likely to represent convective mantle (asthenosphere) rather than hot, upwelling plumes (Giuliani et al., 2020). Indeed, we show how the major peak in kimberlite magmatism post-dates plume magmatism by 35-40 Myr. Whilst this doesn't preclude some of the pre-breakup kimberlites being linked to plumes, there are several arguments against: (1) the earlier kimberlites (at least in Southern Africa) exhibit strong lithospheric signatures for several tens of Myr, and a highly productive plume capable of producing LIPs at the surface would be expected to dominate the geochemical signal after 1-3 Myr; (2) the early kimberlites form a clear temporal progression (Fig. R1c, R2c) toward the cratonic interior that cannot easily be explained by plumes; (3) recent geochemical work on kimberlites spatially associated with the Paraná-Etendeka flood basalts demonstrates that "*the plume has no chemical contribution to the generation of kimberlite*", except perhaps a thermal influence (Conceição et al., *Lithos* 328–329 (2019) 130–145). We feel that our rift-related mechanism can reconcile all these inconsistencies and provide a new conceptual model to explain most kimberlite magmatism.

Here are few smaller issues.

1. Authors use their fragmentation function (ratio between continental perimeter to continental area versus time) citing paper by Merdith et al., 2019. However, this function published in Merdith's paper (Fig. 3a) is significantly different from that used in the manuscript (Extended Data Fig.1d). Please explain.

We thank the reviewer for highlighting this issue. We should have cited the model of Merdith et al. (2021) or MER21—which was under review at the time of submission—as opposed to Merdith et al. (2019) or MER19, and we have now rectified this issue (updated reference 16). We note that the fragmentation index of MER19 used an assembly of global plate kinematic models, including Merdith et al. (2017a, *Gondwana Research* 50, 84), Matthews et al. (2016, *Global and Planetary Change* 146, 226) and Domeier et al. (2016, *Gondwana Research* 36, 275) and Domeier et al. (2018, *Geoscience Frontiers* 9, 789). In contrast, the model we use in the current paper is from MER21, which uses a completely internally consistent set of polygons and rotations. The main benefit of using this new model is that the polygons and rotations, characterizing continental motions over time, are self-consistent from 1 Ga to present. In effect, this meant that we could produce a more consistent and reliable analysis to test the effects of continental fragmentation further back in time (i.e., prior to 410 Ma). Thus, our main analysis (Fig. 1b) is fully consistent, and comparable, with the 1-0 Ga test (Extended Data Fig. 2b).

In addition, we now compare the main attributes of the MER19 and MER21 models. Although there are some clear differences in the absolute value of fragmentation (i.e., continental area/perimeter; see Fig. R4 below), the overall trends of both curves are similar, showing those expected 'lows' during Rodinia and Pangea supercontinent stability and 'highs' reflecting Rodinia and Pangea breakup, consistent with the original hypothesis.

The main difference that the reviewer pointed out arises because MER19 was a non-complete model that used four different models and sets of polygons, including synthetic

COBs, and continental polygon outlines (leading to ‘artificial steps’ where they join; see Fig. R4), whilst our analysis using the MER21 model used a unified set of cratonic polygons for the Neoproterozoic and Phanerozoic. Times where there is some deviation coincide with the shifts in model for MER2019 (i.e., 500 and 250 Ma), and at 600 Ma where the MER21 model updated large parts of the Meredith et al. (2017a) model.

Figure R4: Two alternate continental fragmentation indexes | Continental fragmentation (continental perimeter/area) derived from paleogeographic reconstruction for two alternate models, MER19 and MER21 (labelled, with the latter used in our analysis). Bottom panels show the rate of change of fragmentation (ΔF), corresponding to the two upper panels, calculated using a 9 Myr window.

To test how using these different models affects our results, we repeated our analysis using MER19, the results of which are shown in Fig. R5. Although there is a broader distribution of data between lags of 40 and -50 Myr, the analysis shows the strongest peak (empirical correlation coefficient = 0.3) appears at lags of -21 Myr (i.e., 21 Myr after continental breakup), only slightly earlier than in MER21 (-26 Myr), but within expected bounds of uncertainty. This allows us to conclude that the choice of fragmentation index does not significantly affect the overall conclusions of our paper. Notwithstanding this, we strongly prefer to use the updated, more consistent model (MER21), in our main analysis (Fig. 1b).

Figure 5: Relationship between dynamic continental fragmentation and global kimberlites using MER19 | Cross-correlations between kimberlites⁶ and the rate of change of tectonic fragmentation (ΔF ; 9 Myr window) spanning 500 Myr, here using the MER19 fragmentation model, showing dominant time lags at -21 ± 4 Myr.

2. Are correlation numbers on Extended Data Fig.2a, r or r -squared?

We apologize for not making this statistical definition clearer in Extended Data Fig. 2a. This plot shows the conditional correlations for ΔF (slope over 9 Myr moving window) and kimberlites (count), calculated using a Bayesian network. In short, when using this method, the nature of the correlation values depends on the time window being analysed: the output we use from UNINET (see Methods) is the rank or Spearman correlation for the first pair of nodes (i.e., time lag = 0), and conditional rank correlations for subsequent nodes (i.e., accounting for the conditional dependence on nodes higher up in the network hierarchy, as described in Gernon et al., *Nature Geoscience*, 2021). This statistical concept is defined in Kurowicka and Cooke (2006), page 33: “The conditional correlation of Y and Z given X is the product moment correlation computed with the conditional distribution of Y and Z given X ”, summarized as follows:

$$\rho_{YZ|X} = \rho(Y|X, Z|X) = \frac{E(YZ|X) - E(Y|X)E(Z|X)}{\rho(Y|X)\rho(Z|X)}$$

This statistical definition is now provided/explained in the relevant Methods section (lines 481-489) the above equation (3) is provided for completeness, with a reference added to Kurowicka and Cooke (2006).

In line with the above changes, we have needed to make a series of comprehensive updates to the Methods section. These are shown in blue in the revised document.

Finally, we thank the reviewers again for their very constructive suggestions, and especially for helping us to clarify some of our arguments and strengthen elements of our analytical approach. We strongly believe that the resulting paper is much more robust and compelling.

Reviewer Reports on the First Revision:

Referee #1:

This is my second review of this manuscript. I am impressed with the additional analyses on the data, and agree with the authors that the geodynamic models and data support the presented conceptual model for kimberlites. This is a wonderful contribution!

Referee #3:

Review of “Diamond ascent by rift-driven disruption of cratonic mantle keels” by Gernon et al., submitted to Nature

This is a very interesting and thought-provoking study that sets out to related “global tectonics” to kimberlite emplacement mechanisms. The study is wide-ranging, integrating geochronological data with dynamic modeling, statistical evaluation, magma geochemistry and even mantle xenoliths, which is commendable.

A key question is how generally applicable is the resulting model in explaining “90% of all kimberlites post 500 Ma)” and thus what will its broader impact be, in terms of having the sort of reach required by Nature papers. While the authors have done a diligent and thorough job of replying to the reviews, the reception that this paper gets will boil down to whether the model is broadly applicable. My feeling, for the reasons outlined below, is that while it works quite well for southern African and S. American kimberlites, special cases and modifications have to be made for many other places, and there appears to be little applicability to any of the great many kimberlites erupted across North America in the last 500 Ma. As such, I just don’t find adequate justification for the claim made that this model explains 90% of kimberlite occurrences.

In many places the text still uses phraseology that gives the impression that the model is global, even though the authors recognise openly that it is not, especially in their replies to the reviews. Beyond this, as expanded on below, the extended data and text need to document how the geochronology database was used. Detailed scrutiny of the database that the paper appears to be based on is required and a full explanation of how the ages are judged to be reliable is needed. The extended data/methods is really a collection of figures with long captions. Far more detail is needed so that other can replicate the results here or even fully understand which ages, and their accompanying uncertainties, are used in the statistical models.

Overall, there seem to be many processes that can influence how kimberlites are triggered. If one focusses solely on southern Africa/ S. America a reasonable case can be made for a relationship to rifting as a trigger / driving force. But this is very far from a model that is generally applicable in my opinion. Perhaps a more convincing case can be made in a more extended format, disappointing as an outcome that would be for the authors. But it may lead to a greater acceptance of their model if the “detail” can be properly explored in such a format. If this approach is taken then far more detail is required to fully justify the end conclusions and even the results plotted in the figures themselves.

Comments on the title and abstract: As the focus of this paper is on kimberlites and not diamonds, I feel that the use of the word diamond in the title is not appropriate or even descriptive of the main focus of the paper. It's about kimberlite genesis and ascent. Diamonds are a passenger in this story, very much in the periphery, on the back seats! Given this, then why is diamond the subject of the first sentence of the abstract? Surely the enigma is the formation of kimberlite itself? Also, in the last sentence of the abstract – the only supercontinent that can be invoked to have possibly played a role in the main story and data presented by the authors is Pangea, so I think making “supercontinents” a plural is a little bit of a stretch given the data presented and even the comments made in the paper itself.

Beyond these issues, there are some important questions regarding the phase equilibria modelled and used in this study, that are the critical input to Fig. 3 of the main MS and Extended Data Figure 8 of the Extended Data. I outline the lack of clarity and inadequate documentation of this strand of the modelling below. These issues alone prevent, in my view, the manuscript being published in its present form.

Specific comments

The authors were up-front with a statement about the study initially targeting southern African kimberlites. They then extend the study to S. America. I did not find (but may have missed) the details of the kimberlite age database used, in terms of screening for accurate ages, age uncertainties and ensuring that there is one age per kimberlite body. Critical to this is whether the data published by Tappe, and stated as used by the authors, has been used without further filtering. That database contains multiple ages for individual kimberlites, with an over-abundance of ages for kimberlites that have economic importance. There is little screening for the quality of the age and so, without the authors detailing how they selected the most appropriate age for a given kimberlite, and screened out duplicate ages, I am left wondering how accurate the statement in the figure caption (“well dated kimberlites”) is.

How was this addressed in the current study? Many of the ages in that database have no attributed published data, and no laboratory source where published methods are available. Moreover, the ages have no uncertainties so how is reliability assessed? The geochronology database critically underpins this whole study and demands much more detail than I can find in outlining how it was scrutinised. This issue is critically important in the age of “big data” that we appear to live in. If there are issues with a published database it is best to highlight how those issues have been addressed rather than create the impression, through brevity of text, that the database was simply used “as is” and taken at face value. If the Tappe et al data base has been used “as is” then I would say that this is a critical flaw in the paper.

Moreover, the South African and S. American kimberlites studied appear not to represent such a large fraction of kimberlites that the authors imply.

Specifically, In examining the responses to reviewers, it is not clear to me that the authors validate the statement that their model can explain the “vast majority (~90%) of kimberlites over the last 500 Myrs”. I find no mention of the NWT kimberlites, a field of ~ 400 kimberlites. These kimberlites barely feature on Extended data Fig 1, which may be a function of the geochronological database used.

Within a single field (Lac de Gras) there is an age range from 325 to 45 Ma, with 75 Ma kimberlites spatially overlapping those erupted at 50 Ma or so. The pulse at ~ 50 Ma in that field is significantly outside the post rifting "lag time" of 22 +/- 4 Ma quoted in the text by authors. Much further north are the Somerset Island kimberlites, which plausibly could be related to rifting during formation of the High Arctic Igneous province, but there are few of these kimberlites, which all cluster in a small area at 90 Ma. Further south are Jurassic kimberlites – how does that age "progression" fit with the model (ignoring the 500 to 600 Ma kimberlites all along the northern coast). Furthest from any rifting associated with formation of the Sverdrup Basin are the kimberlites of Lac de Gras, most of which erupted between 75 and 45 Ma, all clustered in a very tight space.

Moreover, there is no manifestation of early magmas dominated by lithospheric input, anywhere in this region, over this whole time period. So I can't see how the current version of the model that the authors present here is easily applicable to this major kimberlite field. Furthermore, it seems that it is difficult to apply this model to any North American kimberlite fields. For instance the large ~ 90 Ma Forte a la Corne field in Saskatchewan and ~ 100 kimberlites of that age and younger in Alberta. I don't see any spatial pattern related to rifting at all for these fields either.

Beyond North America, I cannot envisage how the model can account for the lack of Tertiary and more recent kimberlites in West Greenland. Kimberlites are present in W Greenland that are of 500 to 600 Ma age and also of Jurassic age, but there are none of the young ages that one would expect to result from the young rifting associated with the opening of the North Atlantic. So this seems a case of global statistics perhaps creating an impression of general relationships but closer examination of individual locales failing to live up to the story in many cases. So the model, as presented, is only a local one at best. Is this a Nature-worthy result? I don't think so as other papers have explored, to varying extents, the relationship between kimberlite occurrences in southern Africa and the opening of the Atlantic.

Phase Equilibria aspects: In general I am puzzled as to why the authors have only attempted to use phase equilibria constraints from model systems with limited components rather than trying to use some of the relatively recently available data for systems that more closely resemble kimberlites. The usual retort to such a point would be "how do we know what a real kimberlite melt is?" Yet using highly simplified model systems or thermodynamic formulations that are at present rather inadequate does not "improve" the situation. The input from these models forms one of the key figures in the paper – Figure 3 and so a much fuller discussion of why equilibria has been used that has been little used by people that usually study kimberlite genesis would be a must in this case.

In figure 3b, the authors state that "Carbonated melts generated at point M, which detach and transport a xenolith cargo, likely follow pathway 1-2-3 (red arrow)..." Point M is supersolidus, but points 1-2-3 as shown are all definitely subsolidus. In other words, they froze their melt immediately upon formation. How can a subsolidus assemblage be a melt? Moreover, how can such a sub-solidus assemblage transport xenoliths? Perhaps I have missed something but I doubt I am alone in this regard.

Figure 3b phase relationships are for CMAS-CO₂ – there is always a choice but some justification of why the authors chose not to use the phase relationships in the more complex natural system

determined by, for example, Gerhard Brey and coworkers? How does the neglect of Fe, the alkalis, and H₂O in this system affect their conclusions? Probably dramatically is my guess.

What evidence do the authors have that their thermal boundary layer peridotite is sufficiently oxidized to contain carbonate (as well as diamond)? This oxidation state is not consistent with the xenolith record we have.

P 4, last paragraph (lines 211-212) refers to "...the characteristically small-volume CO₂ and H₂O-rich melts parental to kimberlites..." but the phase diagram presented does not contain H₂O. If it does, then this needs to be made more obvious.

P 5 lines 243-243: As noted above, the ascent paths shown in Figure 3b are subsolidus.

P 5 lines 244-246 It is not clear where on Fig. 3b decarbonation occurs. What are the black lines and arrows in the subsolidus region supposed to signify?

Methods, melting calculations section: It is far from clear what the authors used the Katz et al. (2003) parameterization for (lines 928-952), and how that ties in with their phase diagram shown in Fig. 3b. It is puzzling that they jump from this calculation immediately to Perple_X modelling in an anhydrous model system (CMAS-CO₂).

The commentary in the "Phase diagrams and phase equilibria" section is contradictory. For example, the authors state (lines 965-970) that "The shaded area (Fig. 3b) represents the experimentally determined region of carbonate-silicate melts (Gudfinnsson and Presnall 2005), but phase relations in this region cannot be estimated using Perple_X because no adequate model for carbonate-silicate melts has been developed." This is correct. But then they claim they can calculate the onset of carbonate melting using Perple_X (lines 970-973). This is not possible without a thermodynamic model for the carbonate-silicate melt – something they just stated is lacking in Perple_X. It is very far from clear what the authors did but it is possible that they used a CaCO₃-MgCO₃ melt model of some flavour. They refer to the text document "solution_model.dat" and carbonate melt model "LIQ(EF)" on the Perple_X website. However, the reference in that section of the "solution_model.dat" document is to Franzolin, Schmidt, and Poli (2011, CMP 161:213-227) – which discusses solution models for solid carbonates, not liquid ones. It is therefore unclear where the carbonate melt model comes from as its source cannot be traced from the data provided here. This is critical for the model and must be clarified in any published version of his paper that contains the phase diagrams.

In any event, it is hard to be sure that the authors "new" modelling did any better, or is any more applicable than the phase relationships shown in figure 1 of Gudfinnsson and Presnall (2005) (below). As such, it is unclear what exactly the purpose of the Perple_X modelling was, given the absence of (a) a solution model for the melt and (b) the absence of H₂O (and other constituents) in the modelled system."

Figure 1 of Gudfinnsson and Presnall (2005)

Finally, under “Code availability” (page 14; extended data), the authors state “More details on the computational methods and tools used for this study are available from the corresponding author...upon request.” In my view, this is unacceptable. What happens in ten years? Or if the corresponding author doesn’t bother to answer an inquiry? This data has to be provided, in full, as supplemental data in a Nature-controlled archive.

Other comments

Lines 225 – 228: While it is impressive how much data the authors have tried to co-opt into supporting their model, drawing in the high-T sheared peridotites into the argument here is a little opaque. Yes, the chemical zoning in these peridotites indicates a recent process, close to kimberlite sampling. But the sheared texture requires high strain rates, far higher than that available via convective mantle flow, and so while the statements used are not explicit it is not clear to me what is to be gained by referring to these samples.

Extended data Figure 3. I find these correlations un-compelling. Perhaps they are statistically significant, because of the reasonably large number of data points, but the correlations seem to be dominated by the fewer data points to the left of the figures. Why are there no uncertainties apparent on the X-axis (the time-lag)? This parameter uses a kimberlite age, for which there will be (or should be!) an uncertainty) and a breakup age, which should also have an uncertainty. Maybe the uncertainties on the distance are small, but if the ages relating to the ages are included I would imagine that the error envelopes plotted will not look nearly as tight.

If the lag period between say -10 and + 20 is regressed, one suspects that the correlation is very weak indeed, raising the question of whether all this data should be regressed as one population. Similarly, there is very significant scatter within kimberlites of certain ages. This gets back to my underlying sense that there are several processes that trigger kimberlite generation. This does not negate the process invoked here but it does weaken the case that this model is widely applicable.

Extended Data Figure 8. After looking at this figure several times I am still unclear what the authors are trying to say/do with the Katz et al. model and why they have chosen it over other models – why is it particularly germane to the issue of kimberlite genesis rather than experiments?

Further examination of Figure 8d confirms that the modeled composition JADSCM-7 is completely solid (no melt) along path 1 – 2 – 3 in figure 3b. Yet the text refers to this as the path of the kimberlite magma. Please clarify.

Finally, the phase proportions in Figure 8d are puzzling, in particular the presence of coesite. That composition, according to Gudfinnsson and Presnall, is 43.15% SiO₂, 4.02% Al₂O₃, 33.38% MgO, 10.95% CaO and 8.50 CO₂ (all wt.%), and all experiments GP05 ran had the assemblage ol + opx + cpx + gt + liq (Table 2 of GP05). But the Perple_X calculations have 5% coesite and no olivine... this could only make sense to me if the calculated stability of magnesite frees up enough SiO₂ to destabilize olivine in favour of orthopyroxene. However, if you recalculate JADSCM-7 as magnesite + diopside + enstatite + pyrope + forsterite, you get (mole fractions) 0.328 mag, 0.332 di, 0.067 py, 0.191 fo, 0.083 en. So the presence of coesite in the phase diagram remains a puzzle and the documentation provided is a long way off being sufficient to provide an explanation. Reviewers

should not have to work this hard and neither should readers. Anyone who knows the slightest thing about kimberlites will have their attention drawn to the presence of a free silica phase here... Please explain more fully.

To summarise my issues with the phase equilibria modelling

- 1) The authors provide no real justification for the approach they have taken, using thermodynamic models that are greatly simplified compared to the natural systems represented by kimberlites. Just because certain models allow greater quantification of the parameters sought for further modelling, does not make them the best ones to use from a geological relevance point of view. This needs to be justified in detail.
- 2) The documentation of exactly how they have arrived at the plotted phase equilibria is inadequate for anyone to replicate their approach and requires a great deal of additional information
- 3) Why are the chosen trajectories that are plotted in Fig 3 of the main MS sub-solidus?

Author Rebuttals to First Revision:

Referee #1:

This is my second review of this manuscript. I am impressed with the additional analyses on the data, and agree with the authors that the geodynamic models and data support the presented conceptual model for kimberlites. This is a wonderful contribution!

We thank the reviewer once again for their detailed, constructive, and insightful comments on our paper. We are pleased to read that they are satisfied with the additional data analysis we performed in response to their comments, and that they now fully support our new conceptual model for kimberlite formation.

Referee #2:

Although Reviewer #2 was not able to comment on our revisions, they shared a similar view to Reviewer #1 that our paper is “*an interesting and truly interdisciplinary paper suggesting indeed [a] new idea about the origin of kimberlites.*” Here, we refer to our detailed rebuttal to their original review, which we believe have comprehensively and robustly addressed the helpful points that they raised.

Referee #3:

Review of “Diamond ascent by rift-driven disruption of cratonic mantle keels” by Gernon et al., submitted to Nature

This is a very interesting and thought-provoking study that sets out to related “global tectonics” to kimberlite emplacement mechanisms. The study is wide-ranging, integrating geochronological data with dynamic modeling, statistical evaluation, magma geochemistry and even mantle xenoliths, which is commendable.

We thank Reviewer #3 for providing a detailed and critical review and recognising the interdisciplinarity of our study. We are encouraged to read that they found it “very interesting and thought provoking”. We welcome a healthy degree of scepticism in assessing our model, which is an essential part of moving the science forward.

Reviewer #3 is critical of several aspects of our work, notably two major issues concerning: (1) the general applicability of our model to kimberlites in North America (and indeed globally); and (2) technical concerns regarding the phase equilibria model used. We think these are completely valid concerns. Their insightful comments have prompted us to make some key changes to the manuscript, detailed below, which we hope the reviewers will agree have strengthened our paper. We will of course make all data and analytical code available on publication so that the scientific community can evaluate and independently verify our results.

Geotectonic factors

A key question is how generally applicable is the resulting model in explaining “90% of all kimberlites post 500 Ma)” and thus what will its broader impact be, in terms of having the sort of reach required by Nature papers. While the authors have done a diligent and thorough job of replying to the reviews, the reception that this paper gets will boil down to whether the model is broadly applicable.

My feeling, for the reasons outlined below, is that while it works quite well for southern African and S. American kimberlites, special cases and modifications have to be made for many other places, and there appears to be little applicability to any of the great many kimberlites erupted across North America in the last 500 Ma. As such, I just don't find adequate justification for the claim made that this model explains 90% of kimberlite occurrences.

The reviewer is right in saying that, alongside Africa and South America, most of the additional kimberlites are found in North America. This point has encouraged us to analyse the relationships between these kimberlites and breakup along the associated rifted margins. To be clear, as we stress in the paper, we are not describing a mechanism that is spatially restricted to those regions proximal to rift zones (<300 km or so). Rather, the chains of convective instabilities we describe, which instigate during rifting and breakup, can migrate inboard of rifts over distances of many hundreds to thousands of kilometres: a process limited only by the scale of the continent.

To tackle this issue, we have also explicitly modelled the age uncertainties. The results of this analysis show that the major spatio-temporal trends (and apparent migration rates) of kimberlites that erupted across North America over the last 500 Ma hold key similarities to those across Africa and South America. In North America, kimberlite volcanism intensified in the Lower Jurassic immediately after initial rifting and breakup of the Pangaea supercontinent. We summarise our new results against the relevant points below. Based on this additional data analysis and other arguments outlined below (and in the revised manuscript), we respectfully disagree with the reviewer's point that “special cases and modifications have to be made”, and that “there appears to be little applicability to any of the great many kimberlites erupted across North America in the last 500 Ma”.

In many places the text still uses phraseology that gives the impression that the model is global, even though the authors recognise openly that it is not, especially in their replies to the reviews.

To be clear, we do not openly recognise in our paper or rebuttal that the model does not apply globally. We did accept that “the mechanism described in our paper cannot explain all kimberlite occurrences” (see rebuttal for revision 1) and that some kimberlite occurrences may be “due to localised thermo-mechanical thinning of lithospheric root”—a similar, but more localised, mechanism. On the contrary, a major stage in our analysis was demonstrating that a consistent trend does emerge in our global cross-correlation analysis (Fig. 1; Extended Data Fig. 2), even when scrutinised using different time intervals; and this trend then matches that found in our regional studies (now extended to include North America). Further, when we tested this relationship using robust statistical frameworks (i.e., Bayesian networks), we found that the same dominant time lag prevails—strongly suggesting that this is a real feature of the global dataset. We are not denying that there are local departures from the global

model, but the existence of local scatter does not disprove the global model. The local scatter is shown in plots (see Fig. 1c) that also clearly show the global trend; the latter remains even when we undertake robust uncertainty analysis (described more fully below). A significant degree of both regional and global variability is to be expected, due to the complex temporal and spatial distribution of rifts and their relation to kimberlites. As rifts form and propagate, it is not always straightforward to associate an individual kimberlite or cluster with single initiating rifting event in one geographic location and point in time. Further, as we demonstrated in response to reviewer #1's earlier comments, there are two major phases of migrating convective instabilities, starting with rift onset. The second rift necking/breakup phase (typically ~40 Myr after rift initiation) is expected to drive renewed magmatism near the rift, resulting in variability. Our latest statistical approach (described below) enables us to evaluate the model uncertainty by considering the end member cases for initiating kimberlite eruption.

Beyond this, as expanded on below, the extended data and text need to document how the geochronology database was used. Detailed scrutiny of the database that the paper appears to be based on is required and a full explanation of how the ages are judged to be reliable is needed. The extended data/methods is really a collection of figures with long captions.

We beg to disagree – our Methods section provides a comprehensive description (spanning 9 pages) of all approaches used in this study, which we would argue is complete and allows for replication of all aspects of our work. The Extended Data Figures and Tables conform to *Nature* formatting guidelines. The captions are sufficiently long to enable the reader to follow them and are comparable in length to those of other recent *Nature* papers. We note that some of the longer captions resulted from changes in addressing comments from Reviewers #1 and #2, so we are reluctant to truncate these descriptive materials. However, where possible, we have edited the captions down for clarity.

Far more detail is needed so that other can replicate the results here or even fully understand which ages, and their accompanying uncertainties, are used in the statistical models.

We strongly object to this comment as we have been completely transparent about how all aspects of all our statistical analyses were performed. For instance, in the Methods we provided over 100 lines of detail on how this is undertaken with care to avoid over-representation of well-sampled or well-studied regions. We feel this is an over-cautious approach given that the Tappe et al. compilation that we used already “contains entries for each major kimberlite cluster from every continent without over-representing economic clusters that ... are more intensively studied”. Even still, our latest analysis including Africa, South America, and North America, shows that whether we use the clustered or raw data, we still obtain median migration rates of 20.6 km Myr⁻¹ (clustered; Fig. 1e) and 25 km Myr⁻¹ (individual kimberlites), which is broadly in line with convective migration rates derived from analytical and geodynamic models (Fig. 3a).

In the interests of transparency, we will adhere to the FAIR Guiding Principles for scientific data management and stewardship and make all analytical codes and databases available in the published Article, so that the study can be fully replicated by other groups and that the databases can be updated as more information becomes available. We respond more fully to this point under “specific comments”, below.

Overall, there seem to be many processes that can influence how kimberlites are triggered. If

one focusses solely on southern Africa/ S. America a reasonable case can be made for a relationship to rifting as a trigger / driving force. But this is very far from a model that is generally applicable in my opinion. Perhaps a more convincing case can be made in a more extended format, disappointing as an outcome that would be for the authors. But it may lead to a greater acceptance of their model if the “detail” can be properly explored in such a format. If this approach is taken then far more detail is required to fully justify the end conclusions and even the results plotted in the figures themselves.

We thank the reviewer for making this important point and accepting that a reasonable case can be made for rifting as a driver of the South American and Southern African kimberlite fields. This in our view is an important finding, not least because it is clearly distinct from the models proposed in other high-profile papers on this area (e.g., Torsvik et al., *Nature*, 2010). That said, we’ve incorporated additional data analysis that indicates the wider applicability of our model. This is perhaps not surprising given that our initial global analysis (Fig. 1b) strongly suggests that the underlying process is globally applicable.

We believe that this advance is sufficiently significant to merit publication in *Nature* under the standard six- to eight-page *Article* format. *Nature* Articles allow for a comprehensive self-contained Methods section as well as Extended Data Figures & Tables, and Supplementary Materials. These are appended to the PDF when downloaded, meaning the ancillary details can be properly explored in this format. Naturally, it is up to the reader whether they want to read it, but this is a case with any paper. Again, we will include all data arising from this study as Supplementary Data Files so that our model can be rigorously scrutinised in future studies. The detail we provide in the paper and standard accompanying materials (produced in accordance with *Nature* policies), fully justify the end conclusions we have reached.

Comments on the title and abstract: As the focus of this paper is on kimberlites and not diamonds, I feel that the use of the word diamond in the title is not appropriate or even descriptive of the main focus of the paper. It’s about kimberlite genesis and ascent. Diamonds are a passenger in this story, very much in the periphery, on the back seats! Given this, then why is diamond the subject of the first sentence of the abstract? Surely the enigma is the formation of kimberlite itself? Also, in the last sentence of the abstract – the only supercontinent that can be invoked to have possibility played a role in the main story and data presented by the authors is Pangea, so I think making “supercontinents” a plural is a little bit of a stretch given the data presented and even the comments made in the paper itself.

In line with the reviewer’s comment, we have changed the title to “Kimberlite ascent by rift-driven disruption of cratonic mantle keels”. Accordingly, we have changed the first line of the abstract accordingly to read:

“Kimberlites are volatile-rich, occasionally diamond-bearing magmas that have erupted explosively at Earth’s surface in the geologic past^{1,2,3}.”

We do, however, feel that we are justified in using the word “supercontinents” in the abstract. Our long-term cross-correlation analysis, targeting at least two supercontinent cycles, provides compelling support that this is a recurring mechanism over the past billion years at least (Extended Data Fig. 2). This helps explain the peaks in kimberlite magmatism that follow the breakup of the Columbia and Rodinia supercontinents. Even if one is sceptical about “big data” approaches, the important time lags we reveal are robust and indisputable, and our regional analysis (now spanning three different continents) corroborate these global

trends. Given the reviewer fully accepts that our model can feasibly provide a causal linkage between rifting and kimberlite magmatism in southern Gondwana—and that both our analytical and geodynamic models provide a fundamental physical mechanism that accompanies rifting and breakup of cratonic lithosphere—we don't think this is too much of a stretch. We hope that our latest analytical results will help alleviate the reviewer's concerns regarding the global applicability of our model. We urge them to consider the merits of our model with respect to existing models for kimberlite generation and triggering.

Beyond these issues, there are some important questions regarding the phase equilibria modelled and used in this study, that are the critical input to Fig. 3 of the main MS and Extended Data Figure 8 of the Extended Data. I outline the lack of clarity and inadequate documentation of this strand of the modelling below. These issues alone prevent, in my view, the manuscript being published in its present form.

As outlined in response to the reviewer's specific comments below, we've tackled these issues with the phase equilibria and amended the relevant figures and text accordingly. To reflect these key changes, we have also re-written the phase equilibria modelling part of the Methods section. We thank the reviewer for their helpful suggestions on these aspects.

Specific comments

The authors were up-front with a statement about the study initially targeting southern African kimberlites. They then extend the study to S. America. I did not find (but may have missed) the details of the kimberlite age database used, in terms of screening for accurate ages, age uncertainties and ensuring that there is one age per kimberlite body. Critical to this is whether the data published by Tappe, and stated as used by the authors, has been used without further filtering. That database contains multiple ages for individual kimberlites, with an over-abundance of ages for kimberlites that have economic importance. There is little screening for the quality of the age and so, without the authors detailing how they selected the most appropriate age for a given kimberlite, and screened out duplicate ages, I am left wondering how accurate the statement in the figure caption ('well dated kimberlites') is. How was this addressed in the current study? Many of the ages in that database have no attributed published data, and no laboratory source where published methods are available. Moreover, the ages have no uncertainties so how is reliability assessed? The geochronology database critically underpins this whole study and demands much more detail than I can find in outlining how it was scrutinised. This issue is critically important in the age of "big data" that we appear to live in. If there are issues with a published database it is best to highlight how those issues have been addressed rather than create the impression, though brevity of text, that the database was simply used "as is" and taken at face value. If the Tappe et al data base has been used "as is" then I would say that this is a critical flaw in the paper.

Categorically, we can say that there is no such critical flaw in our paper, as we have not used the Tappe et al. database "as is". Again, we must also clarify that Tappe database "contains entries for each major kimberlite cluster from every continent without over-representing economic clusters that ... are more intensively studied" (Tappe et al., 2018), so it is pre-filtered, and this should hopefully quell the reviewer's doubt. As the reviewer correctly states, there may well be multiple ages for some individual kimberlites. This is something we addressed during the previous round of revisions, where we applied a spatio-temporal

clustering approach that further mitigates over-representation of similar-aged kimberlites in clusters. As we stated, the rationale for doing this is as follows:

We define kimberlite clusters as points that are close enough in space and time that they could plausibly be attributed to the same initiating event. By using clustered data, we attempt to eliminate the over-representation of similar ages within relatively small spatial areas. This eliminates some of the reporting bias in the catalogue, as eruptive products are more likely to be identified and radiometrically dated in proximity to previously discovered kimberlites. This approach also (to a limited extent) accounts for some of the age uncertainty in the records, i.e., where there could be two or more different dates (within the bounds of uncertainty) for a single kimberlite event.

We have added details on the spatio-temporal clustering approach in the main text, referring to the Methods, where the details on how the kimberlite database was used are detailed. With respect to the reviewer's criticisms of the Tappe et al. database, we feel that a re-review of that EPSL paper is neither warranted nor appropriate, as we are assured the database was assembled with care and subject to rigorous peer review at a high-quality journal. The attendant mean, median and modal age uncertainties are 6, 4, and 2 Myr, respectively. To address this point, we've added a section to the Methods [lines 484-509] that describes the database and justifies its use, whilst highlighting the more worrying age uncertainties associated with alternative databases, such as that used by other high-profile studies (e.g., Stern et al., 2016).

With the helpful input of Sebastian Tappe, we now incorporate radiometric age uncertainties in the relevant scatter plots, as advised under the "other comments" section below. In summary, we performed 5,000 Monte Carlo simulations using the uncertainty distributions to explore how slopes of the linear regressions (equating to migration rates) could be affected by age uncertainties. As discussed below, we conclude from this analysis that the age uncertainties do not significantly affect our overall results.

We have added a substantial new section to the Methods [lines 721-848] that sets out in detail how we produced an ensemble of 5,000 Monte Carlo simulations, generating a set of distance, lag, and migration rates for each kimberlite. We now explicitly factor all these uncertainties into our analysis, which serves to strengthen our conclusions.

Moreover, the South African and S. American kimberlites studied appear not to represent such a large fraction of kimberlites that the authors imply. Specifically, In examining the responses to reviewers, it is not clear to me that the authors validate the statement that their model can explain the "vast majority (~90%) of kimberlites over the last 500 Myrs".

Though it may seem subtle, we would like to give the complete and precise quotation from our rebuttal ("our updated analysis (described below) can ***feasibly explain*** the vast majority (approximately 90%) of kimberlite occurrences, at least over the past 500 million years"). This statement is not factually incorrect: our cross-correlation analysis between continental fragmentation and the kimberlite distribution (n = 860; Fig. 1b) includes lags of ± 150 Myr and shows the associated confidence intervals. This shows that most of the studied kimberlites in this time frame (500-0 Ma) form in the period between 50 Myr before breakup (the rifting interval) and 75 Myr after breakup. We originally interpreted these data to indicate an underlying systematic mechanism. The key phrase above is "can feasibly explain": these kimberlites erupt between rifting onset and 75 Myr after breakup, in line with expectations from our thermo-mechanical models.

The above arguments relate to our global analysis (Fig. 1b) but do they hold true for the regional spatio-temporal analysis? For the three case studies of interest (South America, North America, and Africa) we are of course analysing kimberlites tied to rifting that occurred since the earliest stages of Pangaea fragmentation (240 Ma according to Müller et al., 2019). The uncertainty analysis includes a total of 623 kimberlites assigned to 253 clusters. Given that there are 712 dated kimberlites of this age in the catalogue, our high-resolution geospatial analysis explicitly considers 87.5% of these, which is broadly consistent with our original estimate (~90%). We hope that the reviewer can appreciate the significance of the new data that we've incorporated, which supports a systematic, highly organised mechanism to drive a physical migration of kimberlite volcanism after rift initiation.

Finally, we are not denying that there are local departures from the global model; importantly though, the existence of local scatter—including examples described by the reviewer—does not disprove the global model. As we stated in the paper, “the longevity of kimberlite magmatism will be limited by the availability of metasomatized keel, which may be prolonged or ephemeral, depending on its rate of entrainment in convective upwellings and exhaustion during melting (Fig. 4)”. Our observations and models suggest, therefore, that kimberlites could erupt sometime after the passage of an initial convective instability, which can feasibly explain some of the scatter. Furthermore, as we concluded in response to earlier reviews (a conclusion that is bolstered by our latest analysis, described below), there strongly appear to be two major phases of instability migration: one that lags rift onset (resulting in a subordinate peak in kimberlite volcanism), and another that lags continental breakup (resulting in the dominant peak in kimberlites).

I find no mention of the NWT kimberlites, a field of ~ 400 kimberlites. These kimberlites barely feature on Extended data Fig 1, which may be a function of the geochronological database used.

We note that many NWT kimberlites are included and plotted on the map. Please also note however, that some of kimberlites in this area such as Snap Lake (Gernon et al., *J. Geol. Soc. London*, 2012) are somewhat older (~541 Ma) than those kimberlites shown, which are <500 Ma). Still, there are other younger kimberlites here such as Diavik and Ekati. As Tappe et al. (2018) state: “Although quality age information is not available for every occurrence, our database contains entries for each major kimberlite cluster from every continent without over-representing economic clusters that host diamond mines and are therefore more intensively studied”. To illustrate this, we can plot the Tappe et al. kimberlites (500 to 0 Ma, as shown on Extended Data Fig. 3a) alongside a global distribution of kimberlites (n = 4,287, from Faure et al. (2010), as used in Stern et al. (2016)), plotted as a heat map (below) with darker areas denoting more densely clustered kimberlites and vice versa – please note that the points used to generate the heat map have been added to Extended Data Figure 3 (as black diamonds) to show the “global distribution”.

Based on these maps, we are satisfied that our analysis captures all the main kimberlite clusters where decent age controls exist (noting that this is consistent with the filtering method used by Tappe et al. (2018)). Still, to err on the side of caution, we tackled spatial bias further by applying a spatio-temporal clustering approach. As the reviewer advised (not simply using the database “as is”), we do not count multiple kimberlites of the same age in the same location, so a field of ~400 kimberlites would only be counted once if they are within a specific time/distance of one another, as specified in the Methods where our spatio-temporal clustering approach is described.

Within a single field (Lac de Gras) there is an age range from 325 to 45 Ma, with 75 Ma kimberlites spatially overlapping those erupted at 50 Ma or so. The pulse at ~ 50 Ma in that field is significantly outside the post rifting “lag time” of 22 ± 4 Ma quoted in the text by authors. Much further north are the Somerset Island kimberlites, which plausibly could be related to rifting during formation of the High Arctic Igneous province, but there are few of these kimberlites, which all cluster in a small area at 90 Ma. Further south are Jurassic kimberlites – how does that age “progression” fit with the model (ignoring the 500 to 600 Ma kimberlites all along the northern coast). Furthest from any rifting associated with formation of the Sverdrup Basin are the kimberlites of Lac de Gras, most of which erupted between 75 and 45 Ma, all clustered in a very tight space.

The reviewer is correct in stating that many of the North American kimberlites erupted quite some time, and quite some distance away (in some cases, thousands of kilometres) from rift systems. We need to stress here that the post-rifting lag time of 26 ± 4 Myr is a global peak value when we analyse the inter-relationship between the kimberlite record and the global tectonic fragmentation index (as outlined in the second section of the Methods). However, the range of lags (h) where the CCF exceeds the 95% confidence interval spans from $h = -75$ to 40 Myr. Correlation begins to increase from ~40 Myr before breakup ($h \approx 40$), peaking at around 26 Myr after breakup ($h = -26$), dropping sharply around $h \approx -50$, and falling below the 95% CI at $h \approx -75$ (75 Myr post-breakup).

A key consideration here is that the disruption of cratonic lithospheric keel appears to commence sometime after rifting onset, as discuss in the manuscript in response to suggestions from Reviewer #1 [lines 70-72, 240-244, 332-343]. Based on the two major

peaks in kimberlite magmatism observed in both Africa and South America (Fig. 1d), we previously suggested that this occurs in two broad phases, one lagging rift onset, and the other lagging rift necking or breakup. This of course allows for kimberlites forming at different times with respect to breakup, as we discussed in response to Reviewer #1.

Importantly, like the two broad phases observed for Africa and South America (Fig. 1d), we also identify two peaks for the North American kimberlite fields (see figure below, shown in Extended Data Fig. 5), with the first, smaller, peak occurring between (protracted) rift onset and breakup, and the second, major, peak occurring 30–60 Myr after breakup. For North America, the total number of individual kimberlites included in the analysis is 207, which are assigned to 65 clusters using the Methods described.

The novel process we propose (that is, migration of convective instabilities along the keel of the craton) occurs over lateral distances exceeding 10^3 kilometres (shown in Fig. 2 & Supplementary Animation 1), limited only by the continental length-scale. Because of the large difference in continental length-scales between Africa/South America and North America, for ease of comparison we visualise them using a density plot of apparent migration rates. In the case of North America, we find that the median rate of migration of kimberlites with respect to rift onset is 26 km Myr^{-1} (Fig. 1e), which is broadly similar to the expected range based on the other kimberlite regions, the analytical models, and the thermo-mechanical simulations (Fig. 3a).

Fig. 1e: Probability density plots showing migration rates of kimberlites ($n=623$; clusters $n=253$), here relative to rift initiation, for continental margins of Africa, South America, and North America; lobar median migration rates are 20.6 km Myr^{-1} for clustered data (black dashed line), accounting for age, distance, and model uncertainties (Methods).

We have added the following paragraph to the paper to describe the results of our latest North American kimberlite analysis [lines 82-111]:

This finding prompts us to ask whether similar migration patterns can be identified in the many kimberlite fields that erupted across North America during the Phanerozoic, with this activity escalating in the Jurassic (Fig. 1a; Extended Data Fig. 3a). The Cretaceous kimberlite ‘corridor’ (encompassing Somerset Island, Kansas, and Saskatchewan) tracks the edge of the North American craton, far from any known mantle plume influence¹⁶. We perform a similar distance-lag analysis for these kimberlites relative to the major Pangaea rift system that initiated along the Atlantic continental margin of the United States in the Middle Triassic, 240 Myr ago²⁶. Like Africa and South America (Fig. 1d), we can identify two peaks in volcanism—one occurring between rifting and breakup, and the other lagging breakup by 30–60 Myr (Extended Data Fig. 5). Collectively, these data reveal median migration rates of 26.3 km Myr^{-1} relative to rifting onset (Fig. 1e). The variability in migration rate estimates (Fig. 1e) can be explained by the complex spatial and temporal evolution of rifting (and subsequent breakup) in relation to kimberlite clusters. To address this, we quantify the uncertainty associated with attributing a kimberlite eruption to a specific initiating rifting event (Fig. 1e; Methods). Median migration rate estimates for the end-member cases are 11 and 30 km Myr^{-1} for kimberlite clusters globally (Extended Data Fig. 4), with an overall median of 20.6 km Myr^{-1} (Fig. 1e; including $n=623$ kimberlites, or 87.5% of the catalogue from 240 Ma). Notably, these rates are consistent across all three continents, irrespective of continental length scale (Fig. 1e), suggesting that some fundamental mechanism gives rise to migration of kimberlite volcanism far inboard of rift zones in the tens of millions of years after rifting commences.

Moreover, there is no manifestation of early magmas dominated by lithospheric input, anywhere in this region, over this whole time period. So I can’t see how the current version of the model that the authors present here is easily applicable to this major kimberlite field. Furthermore, it seems that it is difficult to apply this model to any North American kimberlite fields. For instance the large $\sim 90 \text{ Ma}$ Forté a la Corne field in Saskatchewan and ~ 100 kimberlites of that age and younger in Alberta. I don’t see any spatial pattern related to rifting at all for these fields either.

We thank the reviewer for airing their scepticism with regards to the composition of early magmatism in the North American kimberlite fields. This point prompted us to undertake an analysis like that we performed for the southern African kimberlites (Fig. 4; Extended Data Figs. 8-9). We identify a remarkably similar step change in chemistry of North American kimberlites that mirrors that seen in Cretaceous southern Africa. This shift is manifested as an abrupt change from lithospheric (ϵNd_i -10 to -20) to mantle (ϵNd_i 0 to +5) kimberlite ϵNd_i and ϵSr_i isotope compositions shortly after 105 Ma (Fig. 4b, shown below) [lines 367-373]. The nature and magnitude of this shift—which we attribute to an early peak in lithospheric input followed by upwelling of asthenosphere—is very similar to that seen in the southern African kimberlites. We hope that this information will allay the reviewer’s concerns here.

Fig. 4: Geochemical evidence for removal of cratonic keel | Major kimberlite fields in **a**, South Africa, and **b**, North America, show abrupt shifts, identified using conjugate partitioned recursion (red lines; see Methods), from lithospheric (ϵNd_i -6 to -20) to mantle (ϵNd_i 0 to +5) Nd isotope compositions. Data sources provided in Extended Data Figs. 8–9; shaded band in **(a)** denotes independent evidence for removal of ~40-km-thick keel⁴².

Beyond North America, I cannot envisage how the model can account for the lack of Tertiary and more recent kimberlites in West Greenland. Kimberlites are present in W Greenland that are of 500 to 600 ma age and also of Jurassic age, but there are none of the young ages that one would expect to result from the young rifting associated with the opening of the North Atlantic. So this seems a case of global statistics perhaps creating a impression of general relationships but closer examination of individual locales failing to live up to the story in many cases.

The reviewer is justified in being critical of our model, and we expect many geologists will point out cases where cratonic or continental breakup did not apparently result in kimberlite magmatism. However, we believe it is not fair or warranted to test our model in Greenland, given that 80% of the continental surface is ice-covered, and much is under-explored. Absence of evidence should not be taken as evidence of absence.

However, let us assume for a moment that Greenland is not largely covered in a continental ice sheet. We'll first consider whether the Jurassic kimberlites exposed in the coastal strip of western Greenland (i.e., at that time part of the North American Craton) *could* be consistent with our model, by evaluating whether the migration rate, and composition, of erupted kimberlites would allow for this. The Jurassic kimberlites of Greenland—exposed predominantly in the region around Tikiusaaq, Nuuk—are thought to range in age (using U-Pb dating of groundmass perovskites) from 165.9 to 157.4 Myr (Tappe et al., *Chem Geol.* 320, p. 113, 2012). This region is located approximately 1,590 km from the nearest contemporary major rifted boundary—the northernmost extent of the Central Atlantic rift system around present-day Nova Scotia and Newfoundland. The Greenland kimberlite volcanism initiated 77 Myr after the onset of this major rift zone at 240 Ma (Müller et al., 2019). Assuming for a moment that this volcanism *was* related to migration of convective instability migration, it would indicate an average migration rate of 20.4 km/Myr. This is squarely in line with the median migration rate derived for Africa, South America, and North America (combined), which is 20.6 km/Myr (see Fig. 1e). This rate is in line with expectations from our analytical models and geodynamic simulations.

The geochemistry of Greenland kimberlites of this age would also seem to lend support to our model: Jurassic carbonatite-aillikite magmatism is thought to contain signatures derived from metasomatized rifted cratonic mantle and a carbonate-rich upper mantle (Tappe et al., *Lithos* 112, p. 385, 2009). We interpret this as due to the initial breakup of Pangaea, which disrupted cratonic keels, exhausting some or all the metasomatized portion of keel and thus making it less likely, though not impossible, to form kimberlites in relatively close succession (i.e., during the Cretaceous). As previous studies have shown (Pearson et al., 2021; Capitanio et al., 2020), the metasomatized keels of cratons build up over hundreds of millions, perhaps even billions, of years, via metasomatic fluids driven off subducting slabs. It is replenished very slowly over geologic timescales (i.e., requiring a rich history of subduction near the closest continental margins). As we state in the paper (lines 327-333): "... the longevity of kimberlite magmatism will be limited by the availability of metasomatized cratonic keel, which may be prolonged or ephemeral, depending on its rate of entrainment in convective upwellings and subsequent exhaustion during melting (Fig. 4)."

We can provide another perfectly reasonable explanation as to why Cenozoic kimberlites are not found in this part of Greenland. The tectonic situation here was fundamentally different in the Cretaceous (when it was rift-proximal) relative to the Jurassic (when it experienced the far-field effects of rifting). The western marginal extent of the Greenland ice sheet today is only 150–200 km away from the continental-ocean boundary that formed in the late Cretaceous/Cenozoic. The exposed region is within the zone of extension of the Greenland-North America rift, which was active since c. 120 Ma (Müller et al., *Tectonics* 38, p. 1884, 2019). As noted in the paper, globally we find that kimberlites tend to erupt >200 km inboard of rift margins (Fig. 1c); we do not find kimberlites within the main zone of extension where more crustal thinning has occurred. This is probably because kimberlites can only form where the craton is sufficiently thick (generally ≥ 150 km; Extended Data Fig. 3a-b). Therefore, magmatism and its subsequent migration appears to initiate some distance (i.e., several

hundred kilometres) inboard of contemporaneous rifted margins. As such, we would not expect to find Cenozoic kimberlites in the narrow exposed coastal strip to the west of the ice sheet.

So the model, as presented, is only a local one at best. Is this a Nature-worthy result? I don't think so as other papers have explored, to varying extents, the relationship between kimberlite occurrences in southern Africa and the opening of the Atlantic.

We respectfully disagree with the reviewer's assertion that our finding is "local at best" and that the process we describe doesn't advance on earlier work. Without undermining the importance of earlier studies that explore links with the opening of the Atlantic, these are largely qualitative (relying on coincidences), generally invoke plumes, and/or require mechanisms that are not considered geologically plausible. Our major findings are clearly distinct from these prior studies for several key reasons. First, those studies do not offer a mechanistic link between rifting and kimberlites. They often invoke tectonic structures that traverse the crust and lithosphere. However, as we outline in the introduction, such mechanisms pose a paradox because cratons are characterized by their mechanical strength and long-term stability. Other models invoke plumes or LLSVPs, which are inconsistent both with geochemistry (refs.¹⁴⁻¹⁶) and our global statistical analysis [lines 122-139] (Extended Data Fig. 6).

Our preferred mechanism of lateral propagation of chains of Rayleigh-Taylor instabilities is completely novel with regards to kimberlite petrogenesis. Even in geodynamic terms, it has not previously been demonstrated that Rayleigh-Taylor instabilities migrate inboard of rifts over time. This finding is important and entirely new. As the reviewer accepts, we then present "reasonable" evidence that the kimberlites in South Africa and South America show migration that falls broadly in line with the expected migration rates of such instabilities, which is another novel finding that holds several important implications (e.g., including for mineral exploration). Based on the reviewer's points, we now find that another important regional subset (i.e., North America) shows broadly the same spatio-temporal migration trends as Africa and South America, corroborating our initial global cross-correlation analysis. Furthermore, we quantify the melting associated with Rayleigh-Taylor instabilities and find that this can match the expected characteristics of small-volume low-degree melts in the mantle, thought to characterize the kimberlite mantle source.

Taken together, integrating all these complimentary approaches, data, and observations, which are all in close agreement, has in our view yielded an important and novel shift in our understanding of kimberlites (a view expressed by reviewers #1 and #2).

Phase Equilibria

Phase Equilibria aspects: In general I am puzzled as to why the authors have only attempted to use phase equilibria constraints from model systems with limited components rather than trying to use some of the relatively recently available data for systems that more closely resemble kimberlites. The usual retort to such a point would be "how do we know what a real kimberlite melt is?" Yet using highly simplified model systems or thermodynamic formulations that are at present rather inadequate does not "improve" the situation. The input from these models forms one of the key figures in the paper – Figure 3 and so a much fuller

discussion of why equilibria has been used that has been little used by people that usually study kimberlite genesis would be a must in this case.

We reflected on the reviewer's constructive points with regards to the phase equilibria aspects of the paper and realised that we had perhaps considered this problem from the wrong angle. We need to stress that the phase equilibria are not a fundamental, or novel, part of our model – they are shown largely for context/completeness. We are primarily concerned with evaluating whether the P - T field of the thermal boundary layer (TBL) lies within the fields of carbonate stability and melt generation – this would then demonstrate the viability of convectively removing TBL as a means to drive kimberlite generation in the uppermost asthenosphere.

Whilst many workers accept the CMAS-CO₂ model system is appropriate for studying the carbonated melts parental to kimberlites, we accept that its use may provide an over-simplistic view, given that we later invoke hydrous melting in our calculations. It is perhaps better then to frame the question: “if a convective instability migrates along the cratonic root, what compositions/conditions will it likely encounter” (our new approach), rather than “what compositions could it potentially generate” (our previous approach). This shift in approach prompted us to make some key changes to address the reviewer's points. We think that these changes greatly strengthen the justification for our melting models (Fig. 3c) that then invoke the presence hydrous/carbonate material (again, necessitated by observations from multiple studies cited in our paper). In this respect, the phase equilibria and melting models are now self-consistent. We thank the reviewer for helping us to refine this part of our argument (which is expanded upon below).

In figure 3b, the authors state that "Carbonated melts generated at point M, which detach and transport a xenolith cargo, likely follow pathway 1-2-3 (red arrow)..." Point M is supersolidus, but points 1-2-3 as shown are all definitely subsolidus. In other words, they froze their melt immediately upon formation. How can a subsolidus assemblage be a melt? Moreover, how can such a sub-solidus assemblage transport xenoliths? Perhaps I have missed something but I doubt I am alone in this regard.

In line with our response to the previous point, we have changed the plot and, in the process, removed these details for clarity. Again, we thank the reviewer for helping us to simplify and streamline this part of the paper.

Figure 3b phase relationships are for CMAS-CO₂ – there is always a choice but some justification of why the authors chose not to use the phase relationships in the more complex natural system determined by, for example, Gerhard Brey and coworkers? How does the neglect of Fe, the alkalis, and H₂O in this system affect their conclusions? Probably dramatically is my guess.

The modifications we have made are summarised in response to the reviewers' point above. Again, we have changed our phase equilibria to show peridotite in the presence of CO₂ + H₂O, bringing this part of our framework into closer alignment with our melting calculations. We have updated the main text, Fig. 3b (below), and the Methods to reflect these changes. The updated paragraph in the main paper [lines 258-282] is as follows:

Our models suggest that entrained, variably metasomatized keel (Fig. 2) could be an important contributor of carbonate and hydrous phases to the kimberlite mantle source^{2,3,41,49}. The question is

whether these phases coexist in the lithosphere's thermal boundary layer, which we propose is detached via convective erosion before rapidly recirculating upward and melting (Fig. 2). To investigate this, we can examine phase equilibria models of peridotites³⁰, strongly depleted rocks that dominate the roots of ancient cratons^{8,11,12}. In our model, peridotites will have been encountered by any migrating instability, whereas kimberlite melts are the net product of this instability. Considering peridotite melting in the presence of CO₂ + H₂O, the thermal boundary layer (as defined by our geotherm analysis; Extended Data Fig. 6) is expected to have an important reduced solidus that juxtaposes carbonate-rich and hydrous incipient silicate melts (Fig. 3b). The shallowest keel is metasomatized, containing diamonds that probably formed via redox freezing³⁰. Melting experiments show that hybridization of these reduced, depleted peridotites with oxidized, hydrous CO₂-rich melts should drive melting reactions³⁰. Indeed, it has been proposed that these reactions can account for the formation of key kimberlitic minerals including phlogopite, alkali-amphibole, garnet, clinopyroxene, and perhaps even diamonds³⁰. However, until now, a mechanism to abruptly decouple and hybridize these oxidized and reduced domains was lacking.

Revised Figure 3: Rates of instability migration and conditions of melt generation | **a**, Lateral propagation rates for Rayleigh-Taylor instabilities (equation 1) for six analytical fluid dynamical models (Extended Data Table 1). Models 1, 4 and 5 (black lines) are most applicable, and describe a dense, viscous layer representing the lithospheric keel that overlies a less dense, viscous half-space representing asthenosphere. A kinematic viscosity of $7 \times 10^{15} \text{ m}^2 \text{ s}^{-1}$ gives good agreement with constraints from kimberlite migration (Fig. 1e), numerical simulations of basal lithospheric instabilities (Fig. 2), and lithospheric root thicknesses inferred from modelling xenolith P-T data (Methods). **b**, Phase diagram for peridotite showing the oxidized solidus (CO₂+H₂O) and reduced solidus (CH₄+H₂O) from ref.³¹ (see Methods for further details) and graphite-diamond phase boundary from ref.⁴⁸; red box shows P-T conditions in the thermal boundary layer (Extended Data Fig. 6). **c**, Degree of melting as a function of depth; also shown is the associated melt productivity per unit area of convective upwelling, assuming a mean upwelling rate of 30 km Myr⁻¹ determined from numerical models (Fig. 2); and the wt% water in the resulting melts for bulk water contents of 0.1, 0.15 and 0.2 wt%; these calculations use the hydrous decompressional melting parameterization of ref.⁴⁹ (Methods).

P 4, last paragraph (lines 211-212) refers to "...the characteristically small-volume CO₂ and H₂O-rich melts parental to kimberlites..." but the phase diagram presented does not contain H₂O. If it does, then this needs to be made more obvious.

Thank you – we now label the hydrous melting field more clearly on the plot (see Fig. 3b, shown above).

P 5 lines 243-243: As noted above, the ascent paths shown in Figure 3b are subsolidus.

We thank the reviewer for making this valid point. We have rectified the matter by using an alternative model system as recommended by the reviewer.

P 5 lines 244-246 It is not clear where on Fig. 3b decarbonation occurs. What are the black lines and arrows in the subsolidus region supposed to signify?

We confirm that this point is no longer relevant since we now show a different model system.

Methods, melting calculations section: It is far from clear what the authors used the Katz et al. (2003) parameterization for (lines 928-952), and how that ties in with their phase diagram shown in Fig. 3b. It is puzzling that they jump from this calculation immediately to Perple_X modelling in an anhydrous model system (CMAS-CO₂).

We thank the reviewer for posing this good question.

Firstly, we chose the hydrous decompressional melting parameterisation of Katz et al. (2003) for these calculations as it is an existing, effective, data-driven parameterisation of volatile-rich (esp. hydrous) melting.

This is therefore relevant to the kimberlite mantle source, which most workers agree is enriched in volatiles such as water (references cited in the paper). The Katz et al. model is a well-known parameterisation of a large database of experimental petrology results. The updated Figure 3c (moved across from Extended Data) shows that our mechanism is sufficient to generate the volumes of volatile-rich magmas that are observed at the surface, exclusively via convective removal of the basal lithosphere, without the need for plumes.

As the reviewer correctly states, it is unlikely that CO₂ is the only volatile present during melting, which is why we originally chose the Katz parameterization (and again we agree that the phase equilibria should be internally consistent, as they now are). We require H₂O in the source to explain melting at normal mantle temperatures that can generate the observed kimberlite compositions. The Katz parameterisation predicts between 9 to 14% water in the melt following partial to full convective removal of the lithospheric keel. This range overlaps the estimated compositions of kimberlite melts in Table 1 of Sparks et al. (2009), but we would expect our calculation to provide an upper limit rather than an estimate for the water in the erupted melt. As Sparks et al. (2009) highlight, the composition of the primary melt is not expected to be the same as the composition of the erupted melt: H₂O will likely be lost from the melt by exsolution and by reaction with wall rocks and xenoliths as the melt ascends >150 km through the lithosphere.

We have amended our discussion of this modelling and its implications in the main text [lines 283-309]. We have also described and justified the use of this procedure in the Methods [lines 1,183-1,227].

The commentary in the “Phase diagrams and phase equilibria” section is contradictory. For example, the authors state (lines 965-970) that “The shaded area (Fig. 3b) represents the experimentally determined region of carbonate-silicate melts (Gudfinnsson and Presnall29), but phase relations in this region cannot be estimated using Perple_X because no adequate model for carbonate-silicate melts has been developed.” This is correct. But then they claim they can calculate the onset of carbonate melting using Perple_X (lines 970-973). This is not

possible without a thermodynamic model for the carbonate-silicate melt – something they just stated is lacking in Perple X.. It is very far from clear what the authors did but it is possible that they used a CaCO₃-MgCO₃ melt model of some flavour. They refer to the text document “solution_model.dat” and carbonate melt model “LIQ(EF)” on the Perple_X website. However, the reference in that section of the “solution_model.dat” document is to Franzolin, Schmidt, and Poli (2011, CMP 161:213-227) – which discusses solution models for solid carbonates, not liquid ones. It is therefore unclear where the carbonate melt model comes from as its source cannot be traced from the data provided here. This is critical for the model and must be clarified in any published version of his paper that contains the phase diagrams. In any event, it is hard to be sure that the authors “new” modelling did any better, or is any more applicable than the phase relationships shown in figure 1 of Gudfinnsson and Presnall (2005) (below). As such, it is unclear what exactly the purpose of the Perple_X modelling was, given the absence of (a) a solution model for the melt and (b) the absence of H₂O (and other constituents) in the modelled system.

The reviewer is right – we did not seek to reinvent the wheel with this aspect of our work. Again, we were purely concerned with whether it was feasible to generate melts, or better still, carbonated, and hydrous melts, within the *P-T* conditions of the thermal boundary layer (see Extended Data Fig. 8) that we propose is convectively removed (see Fig. 2 and 3b). We have simplified our arguments, removing some of the previous modelling that focused on the anhydrous model system (CMAS-CO₂).

Further examination of Figure 8d confirms that the modeled composition JADSCM-7 is completely solid (no melt) along path 1 – 2 – 3 in figure 3b. Yet the text refers to this as the path of the kimberlite magma. Please clarify.

In line with our previous responses, we have changed the phase equilibria plot and, in turn, removed this figure.

Finally, the phase proportions in Figure 8d are puzzling, in particular the presence of coesite. That composition, according to Gudfinnsson and Presnall, is 43.15% SiO₂, 4.02% Al₂O₃, 33.38% MgO, 10.95% CaO and 8.50 CO₂ (all wt.%), and all experiments GP05 ran had the assemblage ol + opx + cpx + gt + liq (Table 2 of GP05). But the Perple_X calculations have 5% coesite and no olivine... this could only make sense to me if the calculated stability of magnesite frees up enough SiO₂ to destabilize olivine in favour of orthopyroxene. However, if you recalculate JADSCM-7 as magnesite + diopside + enstatite + pyrope + forsterite, you get (mole fractions) 0.328 mag, 0.332 di, 0.067 py, 0.191 fo, 0.083 en. So the presence of coesite in the phase diagram remains a puzzle and the documentation provided is a long way off being sufficient to provide an explanation. Reviewers should not have to work this hard and neither should readers. Anyone who knows the slightest thing about kimberlites will have their attention drawn to the presence of a free silica phase here... Please explain more fully.

This point is no longer relevant as we have adopted the reviewer’s advice and focus on the melting of peridotite in the presence of CO₂ + H₂O. We’d like to note here that coesite and free quartz have recently been documented in xenoliths in diamondiferous kimberlites (from the Wajrakarur field, Dharwar craton, southern India; Chalapathi Rao et al., in press, *Geological Magazine*), so their presence in carbonate-rich silicate rocks is not entirely exceptional.

To summarise my issues with the phase equilibria modelling

- 1) The authors provide no real justification for the approach they have taken, using thermodynamic models that are greatly simplified compared to the natural systems represented by kimberlites. Just because certain models allow greater quantification of the parameters sought for further modelling, does not make them the best ones to use from a geological relevance point of view. This needs to be justified in detail.
- 2) The documentation of exactly how they have arrived at the plotted phase equilibria is inadequate for anyone to replicate their approach and requires a great deal of additional information

We are disappointed that the reviewer felt this way because we had outlined in the Methods which published model system we used, we provided a link to the openly available software, and we specified the melt compositions from a published compilation (as well as specifying the source of the phase boundaries). In our revised Methods section, we specify the source of all information shown on the phase equilibria plot. We cannot do any more than this.

- 3) Why are the chosen trajectories that are plotted in Fig 3 of the main MS sub-solidus?

We confirm that this issue has now been resolved.

Code availability and xenolith analysis

Finally, under “Code availability” (page 14; extended data), the authors state “More details on the computational methods and tools used for this study are available from the corresponding author...upon request.” In my view, this is unacceptable. What happens in ten years? Or if the corresponding author doesn't bother to answer an inquiry? This data has to be provided, in full, as supplemental data in a Nature-controlled archive.

To ensure complete transparency, we will adhere to the FAIR Guiding Principles for scientific data management and stewardship and make all analytical codes and databases available in the published Article, so that the study can be fully replicated by other groups and that the databases can be updated as more information becomes available.

What evidence do the authors have that their thermal boundary layer peridotite is sufficiently oxidized to contain carbonate (as well as diamond)? This oxidation state is not consistent with the xenolith record we have.

We thank the reviewer for posing this question.

There are two immediate issues here. First, the observation that few documented peridotite xenoliths contain carbonate phases is valid but may well reflect preservation bias: it would be surprising if these phases survived the rapid decompression involved in kimberlite eruption and ascent. Secondly, it must also be noted that diamonds can form by the oxidation of CH₄-bearing fluids, as well as by the reduction of CO₂-bearing fluids (leaving a more oxidised relict SCLM). Our model does not necessarily require a solid carbonate phase and is likely to involve CO₂-bearing fluids (which can be produced by partial oxidation of CH₄-bearing fluids).

The reviewer is right in thinking that the *primary* oxidation state of the cratonic SCLM does appear to have been progressively more reduced with depth (i.e., too reduced in parts for crystalline carbonate to be stable).

However, in the Kaapvaal Craton of South Africa (and others worldwide) it has been demonstrated that the lower lithosphere becomes progressively more melt-metasomatized with depth, a feature which intensifies between depths of 160–190 km, below the LAB and within the thermal boundary layer. The modification (via metasomatism) of harzburgite to lherzolite is accompanied by oxidation, as has been demonstrated by studies of zoned garnets found in xenoliths (Griffin et al., 1999; McCammon et al., 2001) and “by differences between the matrix minerals of diamond-bearing xenoliths and the corresponding phases included in the diamonds” (Stachel et al., 1998; Creighton et al., 2007) [quoted from Griffin et al., 2009].

The *PT* conditions of our thermal boundary layer encompass several key phase changes including the reduced solidus, with hydrous silicate melts stable at higher temperature and lower pressure, to carbonate-rich silicate melts at higher pressure. Within the latter field, ‘redox freezing’ is expected, particularly at higher temperatures (see revised Fig. 3b), whereby reduction of carbonatites to diamond is expected to occur (Rohrback and Schmidt, 2011; Pintér et al., 2022).

But is this representative, or even relevant for cases where the cratonic lithosphere is thinner? A recent study of garnet peridotite xenoliths from the central Slave Craton shows that the oxygen fugacity of cratonic mantle varies as a function of depth by several orders of magnitude (Yaxley et al., 2017). Specifically, Yaxley et al. (2017) found that the lowermost diamondiferous lithosphere (>130 km) is heavily oxidised (by ~4 log units in fO_2), possibly due to metasomatic enrichment. They propose that the lithospheric keel was infiltrated by carbonate-rich, hydrous melts in the presence of diamond. The diamonds represent the solid carbon phase that formed in equilibrium with a metasomatic, hydrous silicate melt in the kimberlite source region, which is consistent with what we are proposing.

In summary, the thermal boundary layer of keel is on average ~35 km thick and encapsulates several important phase changes (Fig. 3b). As such, the chemical conditions are sufficiently varied for both oxidised and reduced material to co-exist within this layer, and the geochemical evidence above would seem to support this view.

Other comments

Lines 225 – 228: While it is impressive how much data the authors have tried to co-opt into supporting their model, drawing in the high-T sheared peridotites into the argument here is a little opaque. Yes, the chemical zoning in these peridotites indicates a recent process, close to kimberlite sampling. But the sheared texture requires high strain rates, far higher than that available via convective mantle flow, and so while the statements used are not explicit it is not clear to me what is to be gained by referring to these samples.

We have removed the high-T sheared peridotites from the phase diagram. The reason we originally included them is that they are widely thought to signify some of the hottest and deepest available parts of the lithosphere, in many cases sampling the shallowest parts of the thermal boundary layer. Griffin et al. (2009) [ref.²¹] show that these high-T sheared lherzolite

xenoliths derive (on average) from depths of 170-180 km, below the LAB, but within the heavily melt-metasomatized section of (Kaalpvaal) lithospheric keel.

Please note we only refer to the sheared fabric as a physical attribute, rather than a dynamic process. We agree with the reviewer that the shearing requires higher strain rates (10^{-10} to 10^{-5} s⁻¹; see Skemer and Karato, *JGR*, 2008) than those available via convective mantle flow (10^{-18} to 10^{-13} ; see Fig. 2). Thus, we agree that the fabrics likely formed during eruptive ascent along deep conduits, or indeed within the mantle shortly prior to entrainment (see Arndt et al., *J. Pet.*, 2022).

Notwithstanding this, we take the reviewer's point and removed them from the phase diagram in the main paper. We have retained the compositional plots for context and completeness (Extended Data Fig. 7), especially considering that (1) these data provide evidence for changes in density that will augment convective removal processes; and (2) some high-T xenoliths overlap with the thermal boundary layer (based on our geotherm analysis; Extended Data Fig. 5) where we propose that melt generation occurs (Fig. 4b).

Extended data Figure 3. I find these correlations un-compelling. Perhaps they are statistically significant, because of the reasonably large number of data points, but the correlations seem to be dominated by the fewer data points to the left of the figures. Why are there no uncertainties apparent on the X-axis (the time-lag)? This parameter uses a kimberlite age, for which there will be (or should be!) an uncertainty) and a breakup age, which should also have an uncertainty. Maybe the uncertainties on the distance are small, but if the ages relating to the ages are included I would imagine that the error envelopes plotted will not look nearly as tight. If the lag period between say -10 and + 20 is regressed, one suspects that the correlation is very weak indeed, raising the question of whether all this data should be regressed as one population. Similarly, there is very significant scatter within kimberlites of certain ages. This gets back to my underlying sense that there are several processes that trigger kimberlite generation. This does not negate the process invoked here but it does weaken the case that this model is widely applicable.

The reviewer is correct in stating that the correlations [linear regressions] plotted in Extended Data Figure 3 are statistically significant: these plots showed the best fit regression line and the 95% confidence interval for this line. It is necessary to evaluate the data over appropriate time and distance scales, and to include as much of the kimberlite population as possible, to infer a global trend. Given the complexity of rift evolution and associated uncertainties, it would not be statistically sound to regress a much smaller subset of the data. For illustrative purposes, it is perhaps more instructive to visualise the distribution of migration rates globally, with a median value across all three continents of 20.6 km/Myr (Fig. 1e).

To further support our original findings, we have followed the reviewer's advice and now include radiometric age uncertainties for these kimberlites from Sebastian Tappe's database. We also explicitly quantify uncertainty in rift and breakup ages, estimates of distance, and additionally in our extended assessment, model (kimberlite-rift association) uncertainty. In terms of distance, the locations of the kimberlites and the continent-ocean boundary (COB) are well-established, the latter described in key syntheses of plate tectonic data (e.g., Müller et al., 2016). COBs are zones rather than precise linear boundaries and the global mean half-width of the COB 'transition' zone is ~90 km (Eagles et al., 2015). We therefore apply an uncertainty on the distance value of ± 90 km, using (conservatively) a uniform distribution.

We also apply a uniform random variation of 5% ($\pm 2.5\%$) to the rifting/breakup ages to capture uncertainties in tectonic reconstructions (Gernon et al., 2021).

Sampling from the above uncertainty distributions, we then performed 5,000 Monte Carlo simulations to test whether age uncertainty envelopes mean the regression “will not look nearly as tight”, as the reviewer suggests. In each run, we vary each kimberlite age within a (conservatively uniform) distribution of uncertainty, as outlined above. We generated a linear regression for each of these simulations, now shown in the scatter plots (e.g., Fig. 1c). Previously the slope of our regression was in the range 17–21 km/Myr. Accounting for age uncertainties, the slopes of the regressions range from 9.3–22.3 km/Myr (median: 16.1 km/Myr; see Fig. 1c). To interrogate this further, we applied an alternative approach, Theil Sen Regression, again considering all bounds of uncertainty (see above for an example of how this compares). This method yields a similar median rate of 15.5 or 17.6 km/Myr, for S America/Africa and Africa, respectively. For context, our thermo-mechanical simulations predict a migration rate for mantle convective instabilities in the range 15–20 km/Myr (Fig. 3a). Therefore, the kimberlite age uncertainties do not significantly affect our overall conclusions.

Extended Data Figure 8. After looking at this figure several times I am still unclear what the authors are trying to say/do with the Katz et al. model and why they have chosen it over other models – why is it particularly germane to the issue of kimberlite genesis rather than experiments?

We have justified the use of the Katz et al. (2003) parameterisation in the reviewer’s point above. For completeness, we have included justification in the main text and in the Methods.

Finally, we would like to again thank all three reviewers for their detailed and constructive comments, which we believe have helped to greatly clarify some key points in our paper. We believe that the resulting conceptual model of a strong mechanistic linkage between continental rifting and kimberlites is even more robust and compelling. Thank you.

Reviewer Reports on the Second Revision:

Referee #3:

This is a re-review of the paper submitted by Gernon et al to Nature outlining a model of the triggering of kimberlite eruptions.

It is very refreshing to see that the authors have considered my earlier comments diligently and with considerable thought. It is also pleasing to see that they have managed to find a way of rationalising the regions of “potentially problematic” kimberlites that I raised, within their model. This is impressive. They have also thoroughly revised the approach to using the phase equilibria data in a much more useful manner that now helps the paper.

Considering the huge amount of effort put in and the new results, I am now much more convinced that this is a more widely applicable model than I first thought. All credit to the authors.

I have only very minor quibbles with the re-submitted text, but overall now think that it is very suitable for publication in Nature.

Lines 278-280. I think the term “kimberlitic minerals” when applied to alkali amphibole, garnet and diopside is potentially mis-leading/ confusing as some may take this to mean that these phases crystallised on the liquidus of primitive kimberlite melts, even though this is not what is implied (I hope). In fact, I don't really see the need for this entire sentence.

The authors remain fixated on diamonds, including it in their very last sentence, but many kimberlites are essentially non-diamondiferous – do the authors mean to say that the model does not apply to such kimberlites? I doubt it. Why drag diamond into this? It has little to do with the main thrust. It is not as though the model can be used to predict whether a given kimberlite has diamonds or not, so continuing to sprinkle the mineral into sentences throughout the paper raises expectations that cannot be met.

I am confused (likely through my own short-comings) about the label on the x-axis of Fig 3 a. It could be taken as some to indicate that the entire lithospheric thickness is between 0 and 40 km, whereas I suspect that the authors are referring to the thickness of the lithosphere removed. Either way, the door is open to confusion.

In Fig 1, I remain a little confused as to why the authors only refer to the median values when the modes of all 3 populations referred to are quite different from the median values – the modes document to most abundant kimberlite eruptions – isn't that worth as much discussion as any measure (no matter how statistically robust) of central tendency?

Similarly, while the authors did a good job of exploring the effects of errors on the regression presented in Fig 1d (thank you) I remain troubled by the fact that (eyballing the data) only 16 of the > 80 data points actually fall within the “95% confidence limits”. There is a risk of any regression with enough data producing tight confidence intervals. Perhaps as significant an issue is whether this is a

single population to be regressed given the 2 processes identified by the authors that may operate and the complexity in the system (also readily acknowledged by the authors). What do the regression residuals and MSWD look like? Is the data strictly homoscedastic? It doesn't appear so. What is the real predictive value of such a regression when so many of the actual data points lie beyond the confidence limits? Irrespective of the "statistics" I am always troubled when significant clumps of the data have zero correlation. Given the total lack of systematic behaviour in large segments of the data, what is the justification for regressing all the data and making the assumptions implied by the fitting routines?

Beyond that I would be happy to see a slightly modified version of the paper published and congratulate the authors on an innovative model.

Author Rebuttals to Second Revision:

This is a re-review of the paper submitted by Gernon et al to Nature outlining a model of the triggering of kimberlite eruptions.

It is very refreshing to see that the authors have considered my earlier comments diligently and with considerable thought. It is also pleasing to see that they have managed to find a way of rationalising the regions of “potentially problematic” kimberlites that I raised, within their model. This is impressive. They have also thoroughly revised the approach to using the phase equilibria data in a much more useful manner that now helps the paper.

Considering the huge amount of effort put in and the new results, I am now much more convinced that this is a more widely applicable model than I first thought. All credit to the authors.

I have only very minor quibbles with the re-submitted text, but overall now think that it is very suitable for publication in Nature.

We are very pleased that the reviewer recognises the effort that has gone into our revisions, and that it is suitable for *Nature*. We appreciated their critical review, and we believe that this helped us refine key parts of our conceptual framework. Thank you.

Lines 278-280. I think the term “kimberlitic minerals” when applied to alkali amphibole, garnet and diopside is potentially mis-leading/ confusing as some may take this to mean that these phases crystallised on the liquidus of primitive kimberlite melts, even though this is not what is implied (I hope). In fact, I don’t really see the need for this entire sentence.

We agree with the reviewer and have removed this sentence to improve clarity.

The authors remain fixated on diamonds, including it in their very last sentence, but many kimberlites are essentially non-diamondiferous – do the authors mean to say that the model does not apply to such kimberlites? I doubt it. Why drag diamond into this? It has little to do with the main thrust. It is not as though the model can be used to predict whether a given kimberlite has diamonds or not, so continuing to sprinkle the mineral into sentences throughout the paper raises expectations that cannot be met.

While we appreciate the reviewer's feedback, we don’t agree that our paper is fixated on diamonds or that removing all mention of them would be appropriate. While our study does not focus exclusively on diamonds, we have mentioned them in a few places where they provide valuable insights, such as in Fig. 3b where we discuss the conditions under which kimberlite melts are typically formed. While it is true that not all kimberlites contain diamonds, the majority in our compilation broadly sample the diamond stability field within the mantle lithosphere (see Extended Data Figs. 3a,b), with a large proportion of these being diamondiferous (see Tappe et al., 2018).

Given the broad readership of *Nature*, we believe that mentioning diamonds is relevant and informative, especially since our findings carry important implications for understanding how, why, and when diamonds are transported to Earth's surface, not to mention for diamond exploration. That being said, we have made a reasonable compromise by removing two key statements that mentioned diamonds in the very last sentence (alluded to by the reviewer) and paragraph 9 (where we discussed diamond stability). We hope this addresses the reviewer's concerns while preserving the scientific rigour and impact of our study.

I am confused (likely through my own short-comings) about the label on the x-axis of Fig 3 a. It could be taken as some to indicate that the entire lithospheric thickness is between 0 and 40 km, whereas I suspect that the authors are referring to the thickness of the lithosphere removed. Either way, the door is open to confusion.

We agree that this label could give rise to confusion and have reworded this to state: “Average thickness of thermal boundary layer, b (km)”. We have also ensured that this is clear elsewhere in the text.

In Fig 1, I remain a little confused as to why the authors only refer to the median values when the modes of all 3 populations referred to are quite different from the median values – the modes document to most abundant kimberlite eruptions – isn't that worth as much discussion as any measure (no matter how statistically robust) of central tendency?

The plot alluded to by the reviewer, Fig. 1e, shows probability density plots for migration rate for all three continental regions. These densities are generated from multiple simulations incorporating age, distance, and model uncertainty. The median value (50 percentile) is a more appropriate summary than the mode, as it is not biased to any one model of association. Because there is no unambiguous way to associate an individual kimberlite eruption with a single initiating rifting event (as a point in time and space), we consider two alternative end-member cases: associating kimberlites with the rift section with either the minimum time lag or the minimum distance (see Methods; Extended Data Fig. 4). The minimum lag assumption leads to larger migration rate estimates (due to combinations of short lags and long distances) and a wider distributional spread. The minimum distance assumption yields shorter migration rates (due to short distances and longer lag times) and a narrower spread, with a higher peak in density compared to the minimum lag model (Extended Data Figs. 4a-b) – thus having a greater effect on the modal value. Fig. 1e shows the overall distribution combining both association models. It is likely that 'reality' is somewhere between these two end member cases – hence considering the median. For completeness, Extended Data Figs 4a and 4b show the mean and median values, as an indication of skewness.

Similarly, while the authors did a good job of exploring the effects of errors on the regression presented in Fig 1d (thank you) I remain troubled by the fact that (eyballing the data) only 16 of the > 80 data points actually fall within the “95% confidence limits”. There is a risk of any regression with enough data producing tight confidence intervals. Perhaps as significant an issue is whether this is a single population to be regressed given the 2 processes identified by the authors that may operate and the complexity in the system (also readily acknowledged by the authors). What do the regression residuals and MSWD look like? Is the data strictly homoscedastic? It doesn't appear so. What is the real predictive value of such a regression when so many of the actual data points lie beyond the confidence limits? Irrespective of the “statistics” I am always troubled when significant clumps of the data have zero correlation. Given the total lack of systematic behaviour in large segments of the data, what is the justification for regressing all the data and making the assumptions implied by the fitting routines?

In reference to the reviewer's comments on Fig. 1c, we would like to clarify some important points. Firstly, while we acknowledge that the relationship between distance and time is not perfect, we provided explanations for this in both revisions (notably, the peak in volcanism at breakup; Fig. 1d & Extended Data Fig. 5). Our analytical models suggest that the process we describe is likely to result in a wide range of rates (Fig. 3a), and we note that these are consistent with the regression slope estimates and uncertainties. We additionally (in Fig 1e and Extended Data Fig. 4) present an alternative approach to the estimation of migration rate. Our density plots for global kimberlites (Fig. 1e) incorporating known sources of uncertainty indicate that the median values (20.6 km/Myr) are again broadly consistent with the slope of the regression (16.1 km/Myr), providing further support for our findings. While two processes are occurring—namely rifting and breakup as the reviewer points out—we believe that they are not entirely independent and rather are related to the same overall geodynamic process.

Secondly, it appears that the reviewer may have confused the confidence interval with the prediction interval. The confidence interval (shaded grey in Fig. R1, below) represents the uncertainty in the regression coefficient, and it is expected that many points will fall outside the 95% confidence interval, particularly with a large sample size. The prediction interval on the other hand (dashed red and orange lines on Fig. R1) shows the spread of the individual data points/observations and is much wider than the confidence interval.

In Fig 1c, the band of blue lines shown in the figure does not represent the “95% confidence limits” but rather individual regressions for 5,000 simulations that account for the uncertainties shown.

Furthermore, the reviewer argues that regression may not be appropriate in this specific context, and therefore the slope of the regression is not valid. However, we respectfully disagree with this assessment, as linear regression is a standard approach for initially exploring the relationship between two such variables, and as outlined above, we provide additional support for these findings by the subsequent application of alternative analytical and modelling approaches. With regard to the regression, the population is not strictly normal and would not be expected to be so, given the spatial/temporal complexity and uncertainties associated with the initiating event (rift initiation or breakup). In this context, the residual plots are not unreasonable (see Fig. R2, below).

We understand the concerns raised by the reviewer regarding the appropriateness of regression, but we respectfully maintain that our overall approach and findings are robust and well-supported, and that regression is just a part of the exploratory picture. We hope that these clarifications address the reviewer's concerns and help to make our findings clearer.

Fig. R1: Relationship between distance and time lag for kimberlite clusters in (a) Africa, and (b) Africa and South America. The blue line shows the standard linear regression, and the grey band shows the 95% confidence limits. The dashed red line shows the 95% prediction bands for the original data set using standard linear regression. The dashed orange line shows the 95% prediction bands for the original data set using Theil Sen regression.

Fig. R2: (a) The residual quantile-quantile plot shows that although most of the points fall broadly on the line, there is a thin tail to the right (where the curve flattens out), and slight skewness. (b) In the residuals vs. fitted plot, the smoothed line shows a slight increase with distance. Variation is not constant, but as points are sparse at small distances it is difficult to draw any strong conclusions. The numbered points are the three largest residuals. As the aim here is to identify a broad relationship between distance and lag without overly manipulating the data, there is no strong case for building a more complex model. We also note that Theil-Sen regression gives broadly similar slope estimates to the standard linear regression (see Extended Data Fig 3c). In addition, we do not rely on confidence intervals, and instead present a range of regression lines from multiple realisations, incorporating very conservative estimates of uncertainty in the data.

Beyond that I would be happy to see a slightly modified version of the paper published and congratulate the authors on an innovative model.

Again, we would like to thank the reviewer for the efforts they have invested in reviewing our paper, which has helped clarify and strengthen key parts of our model.